# Cloud radiative effects significantly increase wintertime atmospheric blocking in the Euro-Atlantic sector

Sandro W. Lubis [1] ✉, Bryce E. Harrop[1], Jian Lu[1,2], L. Ruby Leung [1], Ziming Chen[1], Clare S. Y. Huang [3] & Nour-Eddine Omrani [4]

Reliable simulation, prediction, and complete theoretical understanding of atmospheric blocking remain challenging despite its significant socio-economic impacts. Generations of climate models have notoriously under-estimated blocking frequency, particularly over the Euro-Atlantic sector. Identifying factors controlling blocking frequency and dynamics is therefore essential for improving its simulation. Here, using a cloud-locking experiment, we show that cloud radiative effects (CREs) significantly increase the fre-quency of Euro-Atlantic blocking. CREs enhance upstream diabatic source of wave activity, both directly through longwave heating and indirectly through their feedback on latent heating, with the latter playing the dominant role. The resulting increase in the upstream diabatic source feeds into local wave activity downstream and promotes blocking formation. Qualitatively similar results are shown by multi-model experiments with radiatively inactive clouds to longwave radiation, albeit with a larger impact from mean-state changes. The results underscore the necessity of accurately representing cloud-radiation interactions in weather and climate models for improved prediction of blocking events.

One of the main characteristics of the Earth's mid-latitude troposphere is the jet stream, a fast, narrow band of winds blowing from west to east around the globe. The jet stream occasionally meanders over thou-sands of kilometers over a certain region, favoring the formation of a quasi-stationary high-pressure system that disrupts the mean westerly flow. This condition is known as blocking[1–5]. A blocking event can persist for an extended period of time, from a few days to more than a week[3–5]. Due to their persistence and size, blocks often bring about anomalous, sometimes extreme, weather in the mid-latitudes, includ-ing heat waves, cold spells, droughts, and heavy rainfalls, depending on the season and region[4,6–11]. For example, the unprecedented heat-wave in the Pacific Northwest during the summer of 2021 was due at least partially to strong anticyclonic blocking, which caused a sig-nificant number of deaths and billions of dollars in damage[12,13]. Despite

its importance, simulating and predicting blocking events is notor-iously challenging due to the lack of a comprehensive dynamical the-ory and because blocks are rare and localized[14]. Therefore, it is essential to understand the mechanisms and factors contributing to the formation of blocks.

Generations of climate models mostly underestimate the fre-quency of blocking, casting doubt on future projections of blocking's impacts on climate extremes[4,15,16]. This significant negative bias in blocking frequency is pronounced over the Euro-Atlantic sector and has continued across three generations of climate models (CMIP3 to CMIP6)[15,16]. Such underestimation has been known to be associated with the biases in the model's mean state and/or the wave source at the lower boundary, which are linked to the processes that drive or damp blocking[15–18]. Previous studies have suggested that there are several

[1]Pacific Northwest National Laboratory, Richland, WA, USA. [2]College of Oceanic and Atmospheric Sciences and State Key Laboratory of Physical Oceano-graphy, Ocean University of China, Qingdao, China. [3]Epsilon Data Management LLC, Irving, TX, USA. [4]Bjerknes Centre for Climate Research, University of Bergen, Bergen, Norway. ✉e-mail: sandro.lubis@pnnl.gov

factors contributing to these biases, which include (1) the near-surface temperature gradient[19,20], (2) the strength and pattern of the sea surface temperature (SST) and the oceanic front in the Gulf Stream[18,21–26], (3) the North Atlantic current positions[23,27], (4) the remote influence of tropical convection[22,28,29], (5) the intensity of the orographic drag[30], and (6) latent heating (LH) in ascending air masses upstream of the block[12,31–35]. While improving the representation of these factors in climate models has partially increased the accuracy of blocking frequency[15,16], fully solving the problem of atmospheric blocking biases remains challenging. This is partly attributed to the other unknown sources of bias, and to compensating errors from various factors, such as the effects of horizontal and vertical model grid resolutions[36].

Over the past decades, blocking dynamics were mostly presumed to be predominantly dry, and moist processes were thought to be only of secondary importance. However, by applying back trajectory analysis to observational data, recent studies showed that latent heating (LH) from cloud formation can make a leading order contribution to the anticyclonic potential vorticity (PV) through the strongly ascending warm conveyor belt (WCB) airstreams upstream of the blocks[31,32,34,35]. By simulating observed blocking events in the Integrated Forecast System, Steinfeld et al.[32] showed that artificially turning off LH tendencies upstream of blocking events can substantially reduce the occurrence, duration, and size of blocks. Neal et al.[12] also found that cloud LH upstream of the block feeds the wave activity downstream, contributing to block formation. More recently, Hauser et al.[34,35] corroborated the significance of moist processes in the formation and maintenance of blocking over the Euro-Atlantic sector. They suggested that accurately representing these moist processes could help reduce forecast errors related to blocking in weather prediction models. This highlights the crucial role of diabatic heating associated with moist processes in blocking formation.

Among diabatic processes, the role of Cloud Radiative Effects (CREs)−the radiative heating resulting from the presence of a cloud−in the dynamics and formation of atmospheric blocking is poorly understood. While much research has focused on the impact of diabatic heating associated with LH release on the blocking[31,32,34,35], the influence of CREs remains largely unexplored. CREs can be as important as other diabatic processes, as they significantly affect large-scale circulation and eddies through increased baroclinicity owing to changing meridional temperature gradients and static stability[37–42]. CREs can also cause a robust precipitation increase over the tropical and extratropical oceans across models[40,43], suggesting that cloud-radiative impact might operate via changes in condensation and LH. A recent study showed that CREs influence forecast error growth in the WCB and Rossby wave activity through modifications of latent heating, which in turn governs diabatic PV generation[44]. Given the significant influence of CREs on moist processes and their interaction with large-scale circulation, it is reasonable to hypothesize that they could also affect blocking formation.

To test this hypothesis, we utilize a "cloud locking" technique in the U.S. DOE's Energy Exascale Earth System Model (E3SM)[45,46], in which interactive cloud-radiation interactions are decoupled from the circulation with minimal disruption to the mean state. Isolating the role of CRE in this way provides greater clarity into its impact on blocking formation. While cloud locking has previously been applied to studies of climate variability[39,42,47], extratropical storms[38], and extreme precipitation[43], its application to atmospheric blocking is novel. We further compare the cloud-locking results with those from an experiment where clouds are transparent (radiatively inactive) to longwave radiation as well as with similar experiments from other models participating in the Cloud-Feedback Model Intercomparison Project (CFMIP). As will be shown later, cloud-radiative feedbacks substantially increase blocking frequency, indicating that CRE-related model biases can directly influence blocking formation. Finite-

amplitude local wave activity (LWA) budget analysis[5,12,48] further reveals how CRE-induced diabatic heating shapes blocking dynamics and formation, providing a distinct pathway by which diabatic processes modulate blocking. Overall, these findings underscore the necessity of accurately representing CREs to improve the simulation and projection of blocking events in weather and climate models.

## Results
### Impact of CREs on occurrence of atmospheric blocking
Figure 1 shows the spatial distribution of blocking frequency in the reanalysis and E3SM experiments. The control experiment with prescribed present-day SST forcing (CTL, hereafter; Fig. 1a) generally captures the observed Euro-Atlantic blocking frequency distribution, although the simulated blocking frequency is relatively lower and the maximum center in the mid-latitudes is shifted southward to ~55°N compared to the MERRA2 reanalysis climatology in winter (December to February). The relatively low blocking frequency in CTL may result from the absence of interactive lower-boundary SST forcing, which is known to modulate Euro-Atlantic blocking formation[22,23,25,26]. The evolution of Euro-Atlantic blocking events in the CTL run is also found to be comparable to that of blocking events in the reanalysis (see Supplementary Fig. S1). These results indicate that E3SM performs relatively well in capturing Euro-Atlantic blocking.

To investigate the impact of interactive CREs on blocking climatology, we perform a cloud-locking (CLOCK) experiment in which cloud-radiation feedback is disabled while maintaining mean state similar to that in the CTL. As such the cloud properties received by the radiative transfer scheme are taken from the CTL simulation that is independent of the flow field (see details in the Method section). Although this technique has been shown to effectively preserve the spatial, seasonal, and diurnal structure of CREs, it instantaneously removes the interaction between the atmospheric state and the clouds as seen by the radiative transfer[38,43,46].

Figure 1 b shows the effect of disabling the interactive CRE on the blocking frequency climatology. It is shown that disabling the interactive CRE decreases the blocking frequency over the Euro-Atlantic regions, including the North Atlantic Ocean and West/Central Europe (Fig. 1b,d). Notably, the CLOCK experiment simulates a much lower blocking frequency over the mid-latitude Euro-Atlantic (with the areal mean of $6.06 \pm 1.52\%$ blocked days per winter in CLOCK vs $7.73 \pm 1.50\%$ blocked days per winter in CTL). In other words, removing the interactive CRE reduces the mid-latitude Euro-Atlantic blocking frequency by up to $21.60 \pm 2.14\%$ (see Fig. 1b,d). This result suggests that the interactive CREs can increase the blocking occurrence beyond the mean state.

While the cloud-locking methodology essentially explains the impact of interactive CREs on the blocking formation (with minimal disruption to the mean state), one might wonder whether the changes in the mean state due to radiatively inactive clouds could have a direct or indirect impact on the mean blocking frequency. To investigate this, we conduct another experiment in which clouds do not interact with longwave radiation, i.e., longwave cloud radiative effects (LWCRE) are deactivated. This configuration is similar to the Clouds On/Off Klima Intercomparison Experiment phase 2 (COOKIE-2) methodology within CFMIP; however, here, the underlying SSTs and sea ice are held constant for each year based on climatological monthly varying SSTs from year 2000[46]. We refer to this as the LWOFF experiment. The key difference between the LWOFF and CLOCK simulations is that LWOFF modifies the mean climatic state by removing a major atmospheric heat source (longwave radiative heating from clouds), whereas CLOCK only removes interactive cloud radiative effects with minimal disruption to the mean state[38,43,46].

Figure 1 c shows the response of blocking frequency climatology in the LWOFF. By disabling LWCRE, blocking frequency is reduced

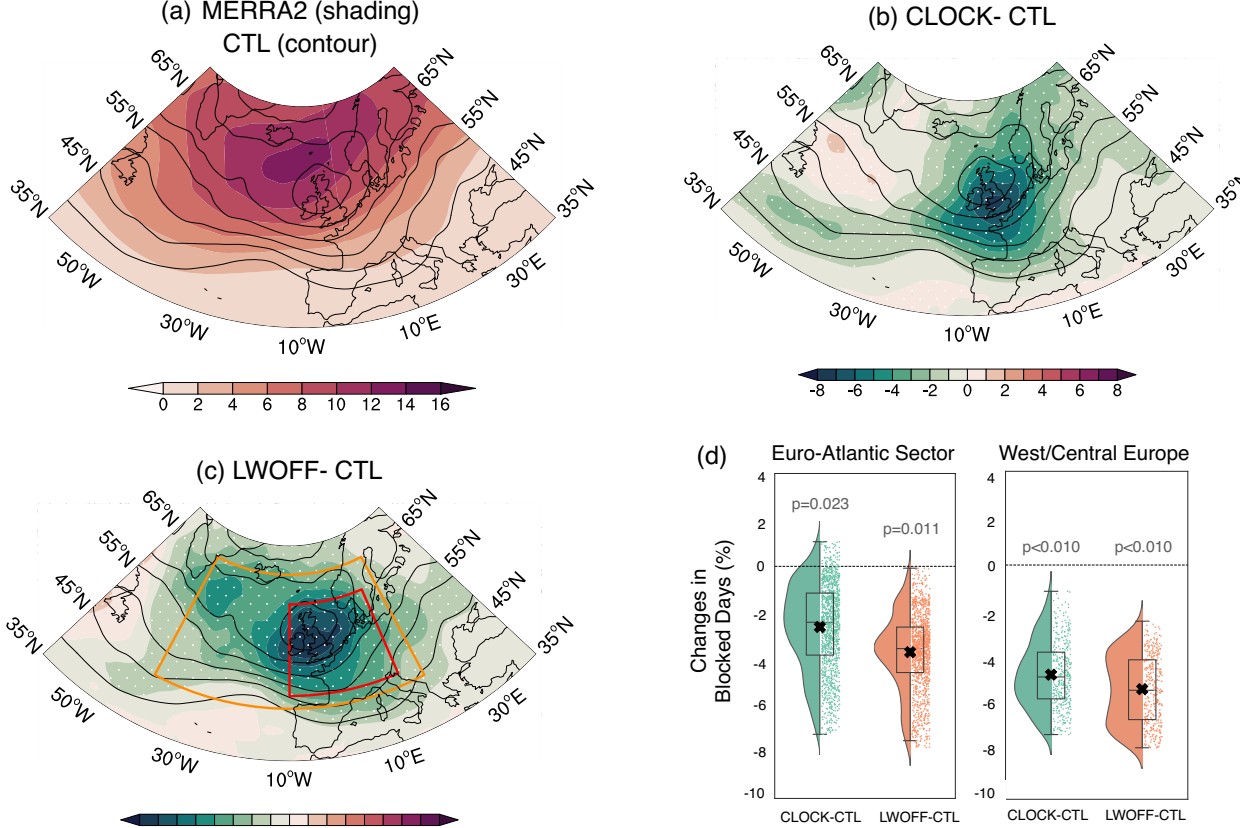

**Fig. 1 | Spatial distribution of Euro-Atlantic winter blocking frequency in the reanalysis and Energy Exascale Earth System Model (E3SM) experiments.**
**a** Wintertime (December–February) blocking frequency climatology in the MERRA-2 (shading) and the control (CTL) simulation (contour lines, with intervals of 2 starting at 2). The unit is percentage of blocked days in the season (i.e., with 2% corresponding approximately to two blocked days per winter). **b** Difference between the cloud-locking (CLOCK) experiment and CTL (shading) and the climatology in the CTL run (contour lines). **c** Similar to (**b**) but for the difference between the longwave cloud radiative effect–off (LWOFF) experiment and CTL

(shading). Dots in (**b**) and (**c**) indicate differences exceeding the 95% confidence interval based on the two-tailed Student's *t*-test. **d** Changes in blocked days averaged over the Euro-Atlantic sector (43°–65°N, 40°W–20°E; orange box, and West/Central Europe (45°–60°N, 10°W–15°E; red box in (**c**)) in half-violin-style box-whisker plot. The vertical widths of boxes represent the interquartile range, whiskers extend from 5% to 95%, and horizontal lines and **x** symbols indicate median and mean values, respectively. The dots represent individual data points. The *p*-values from the *t*-test of CLOCK (LWOFF) versus CTL differences are shown in each panel.

notably over the mid-latitude Euro-Atlantic, with the areal mean dropping to 4.85 ± 1.25% blocked days per winter in the LWOFF, compared to 7.73 ± 1.50% blocked days per winter in the CTL. This suggests that removing LWCRE reduces the mid-latitude Euro-Atlantic blocking frequency by roughly 37.26 ± 3.42%. Moreover, the decrease in blocking frequency affects not only central and western Europe but also a larger portion of the North Atlantic Ocean. The stronger response in the LWOFF simulation compared to CLOCK suggests that changes in mean LWCRE significantly influence the mean states, which, in turn, largely affects the blocking formation. Furthermore, we also compare these results with those from other models in the CFMIP that follow a similar experimental protocol (Table 1 and Fig. 2). The results indicate that disabling LWCRE generally reduces the frequency of blocking events over the mid-latitude Euro-Atlantic region, as demonstrated in most model simulations, similar to E3SM's LWOFF run (Fig. 2). This reduction is statistically significant in all models except one AMIP experiment (MRI), which shows only a slight decrease (Fig. 2b, c). On average, disabling LWCRE lowers the climatological value by about 15, 19% from the corresponding multi-model mean of AMIP runs (Fig. 2b–c and Tables S1–S2 in Supplementary Information). These results confirm that the impact of removing LWCRE on blocking formation is robust and independent of the specific models used. In the following section, we will explain how the CREs enhance Euro-Atlantic blocking.

## Influence of CREs on the dynamical processes of blocking formation

Previous studies have demonstrated that diabatic heating can act as a direct source of wave activity, which influences blocking formation[5,12]. Additionally, changes in the large-scale mean state, such as the location and structure of the jet and the strength of stationary waves, can influence the conditions that favor blocking formation and development[15,49,50]. Building upon these insights, we propose two hypotheses to explain how CREs influence the dynamical processes of blocking formation:

1. Cloud-radiation interactions promote blocking formation by enhancing upstream diabatic source of wave activity throughout the blocking lifecycle.
2. CREs alter the background mean state, which in turn affects the environment in which blocks form.

To test the first hypothesis, we analyze the composite evolution of Euro-Atlantic blocking events based on the finite-amplitude LWA budget. We choose to use the budget of LWA instead of PV because LWA is a positive-definite quantity, as being proportional to the square of (QG)PV anomaly, that directly measures wave amplitude and captures the growth and decay of blocking events. Decomposition of LWA budget and corresponding attribution to the driving physical

**Table 1 | A list of the COOKIE-2 (Clouds On/Off Klimate Intercomparison Experiment) experiments used in this study**

| Name | Research Center (Country) | Experiment | Member |
|---|---|---|---|
| CESM2-amip | NCAR (USA) | amip-1979–2014 | r1i1p1f1 |
| CESM2-amip-lwoff | NCAR (USA) | amip-1979–2014-lwoff | r1i1p1f1 |
| HadGEM-amip | MOHC (UK) | amip-1979–2014 | r1i1p1f1 |
| HadGEM-amip-lwoff | MOHC (UK) | amip-1979–2014-lwoff | r1i1p1f1 |
| IPSL-amip | IPSL (France) | amip-1979–2014 | r1i1p1f1 |
| IPSL-amip-lwoff | IPSL (France) | amip-1979–2014-lwoff | r1i1p1f1 |
| MRI-amip | MRI (Japan) | amip-1979–2014 | r1i1p1f1 |
| MRI-amip-lwoff | MRI (Japan) | amip-1979–2014-lwoff | r1i1p1f1 |
| E3SM-CTL | DOE (USA) | fixed SST-2000[a] | 1 |
| E3SM-LWOFF | DOE (USA) | fixed SST-2000[a]-lwoff | 1 |

The experimental protocol in COOKIE-2 is based on deactivating LWCRE only, similar to the E3SM-LWOFF simulation.

[a]The underlying SSTs and sea ice are held constant each year, based on the observed monthly SSTs in year 2000s condition.

processes are straightforward without ambiguity from change in the sign of metric, as demonstrated in previous relevant studies[5,12,51]. The column budget of LWA[5,12,48] reads:

$$\frac{\partial}{\partial t}\langle A\rangle\cos\phi = \underbrace{-\nabla_H\cdot\langle\mathbf{F}\rangle}_{I} + \underbrace{\frac{f\cos\phi}{H}\left(\frac{v_e\theta_e}{\partial\bar{\theta}/\partial z}\right)_{z=0}}_{II} + \underbrace{\langle\dot{A}\rangle\cos\phi}_{III}, \quad (1)$$

where the right-hand side terms are (I) horizontal convergence of the LWA flux representing the redistribution of wave activity (the sum of zonal and meridional convergences of the LWA flux), (II) meridional eddy heat flux at the lower boundary (i.e., low-level baroclinic source), and (III) nonconservative (diabatic) source or sink of LWA (see Methods for the definitions of variables). Term III is backed out as residual from Eq.(1). It includes the overall contributions from both the diabatic source/sink of wave activity due to diabatic heating and other dissipation mechanism such as mixing, radiative and Ekman damping.

Equation (1) partitions the contribution of dynamical processes to the growth of LWA $\langle A\rangle\cos\phi$ indicating the formation of a block (the occurrence of a "wave activity congestion"[5]) and allows us to quantify the impacts of diabatic processes via term III. The influence of CREs on blocking formation can be directly assessed by explicitly calculating the contributions of diabatic effects and other processes involved in the LWA budget (see Eq. 12 in Methods).

The development of the blocking events (centered at $45°$–$55°$N and $10°$W–$10°$E) in the CTL run during the onset period (days -3 to -1) basically captures the observed blocking evolution in the reanalysis (Fig. 3 and Figs. S1–2 in the Supplementary Information). The onset of blocking is marked by a successive increase in LWA (shading in Fig. 3a–d) over Europe. This rise in LWA signifies stronger deceleration of the zonal flow, which slows atmospheric circulation and promotes the development of blocked flow. This is consistent with previous studies showing that blocks are often associated with persistent, large-valued wave-activity anomalies[5,12,52,53].

The block-related change in LWA during the onset is primarily driven by the horizontal convergence of LWA flux (Term I) and by non-conservative (diabatic) source or sink of LWA (Term III), whereas contributions from the low-level baroclinic source (Term II) are relatively small and secondary (Fig. 3e–p). Specifically, the downstream growth of the LWA over Europe (positive values in Fig. 3a–d) is

predominantly associated with the horizontal convergence of wave activity flux (positive values in Fig. 3e–h), which is partially offset by negative values of Term III (Fig. 3m–p). The positive values in Term I are mainly dominated by the zonal convergence of LWA flux (Fig. S3), consistent with previous studies showing that the zonal LWA flux is the dominant driver of the LWA tendency and a predictor for atmospheric blocking[5,50,53]. Specifically, when the zonal LWA flux exceeds the carrying capacity of the jet stream, blocking would manifest like a traffic congestion[5]. On the other hand, the low-level eddy heat flux (Fig. 3i–l) makes only a minor contribution to the downstream enhancement of LWA.

The downstream growth of horizontal LWA convergence is closely linked to enhanced eastward wave-activity flux (zonal LWA flux) originating from the upstream region (see vectors in Fig. 3e–h and Fig. S3). As shown in Fig. 3m–p, positive values of low-level sources and non-conservative (diabatic) processes are collocated with the emergence and growth of wave activity fluxes, acting as LWA sources, with non-conservative processes being of primary importance. This suggests that the upstream $\langle\dot{A}\rangle\cos\phi$ acts as the main diabatic source of wave activity, which is then advected by the zonal flow downstream and converges to form blocks, as indicated by the LWA convergence downstream (cf.[12]).

Such a large positive values in the upstream nonconservative sources of LWA were also found in a previous study[12], likely arising from diabatic heating associated with moist ascent along the WCB. Looking closely at Fig. 3m–p, it is evident that the positive values are located between the low- (trough) and high-pressure (ridge) centers, where moist air converges in the WCB. This pattern is also observed in the reanalysis data (see Fig. S2m–p), where the positive values upstream are located between the surface low and high pressure centers (see Fig. S4). In addition, both the minimum outgoing long-wave radiation and maximum column water anomalies, characterized the position of the WCB, are collocated with the local maxima of nonconservative LWA sources (Fig. S4). These overall characteristics suggest that the positive $\langle\dot{A}\rangle\cos\phi$ anomaly upstream is predominantly driven by diabatic heating associated with cloud processes. This finding is confirmed by our further direct calculations of diabatic source of wave activity ($\Delta\Sigma$), showing that cloud diabatic heating (both LWCRE and LH) significantly contributes to the positive values of the nonconservative sources of LWA in the upstream region, with the most significant contribution coming from latent heat release (Figs. S5–S6 and later in Figs. 6–7). These results support the findings of Neal et al.[12] and other studies, highlighting the major role of diabatic heating in the onset of blocking formation[31,32,34,35].

Taken together, the results indicate that the formation of the Euro-Atlantic blocking event in both reanalysis and CTL is associated with persistent, large-amplitude wave activity, primarily driven by zonal convergence of LWA flux downstream. This enhanced convergence downstream is fueled by upstream wave activity sources from diabatic heating, with only a minor contribution from the upstream low-level baroclinic generation.

To investigate how the interactive CRE and changes in the mean LWCRE affect the dynamical processes of blocking formation discussed above, we quantify the column LWA and its budget across all three simulations during the onset period, averaged over day $-3$ to day $-1$. Figure 4 compares the LWA and its tendency during the onset period of blocking events (days -3 to -1) in the reanalysis and across the three experiments. CTL and MERRA2 exhibit similar LWA and LWA tendency values, with no significant differences between them (Fig. 4d). Among the model experiments, CTL shows the largest LWA tendency (-2.8 m s⁻¹ day⁻¹), followed by CLOCK (-1.9 m s⁻¹ day⁻¹) and LWOFF (-0.8 m s⁻¹ day⁻¹) (Fig. 4d), consistent with stronger wave activity and growth during the blocking onset period in CTL compared to CLOCK and LWOFF (Fig. 4a–d). Given that stronger LWA enhances blocking formation, these results confirm our earlier findings that

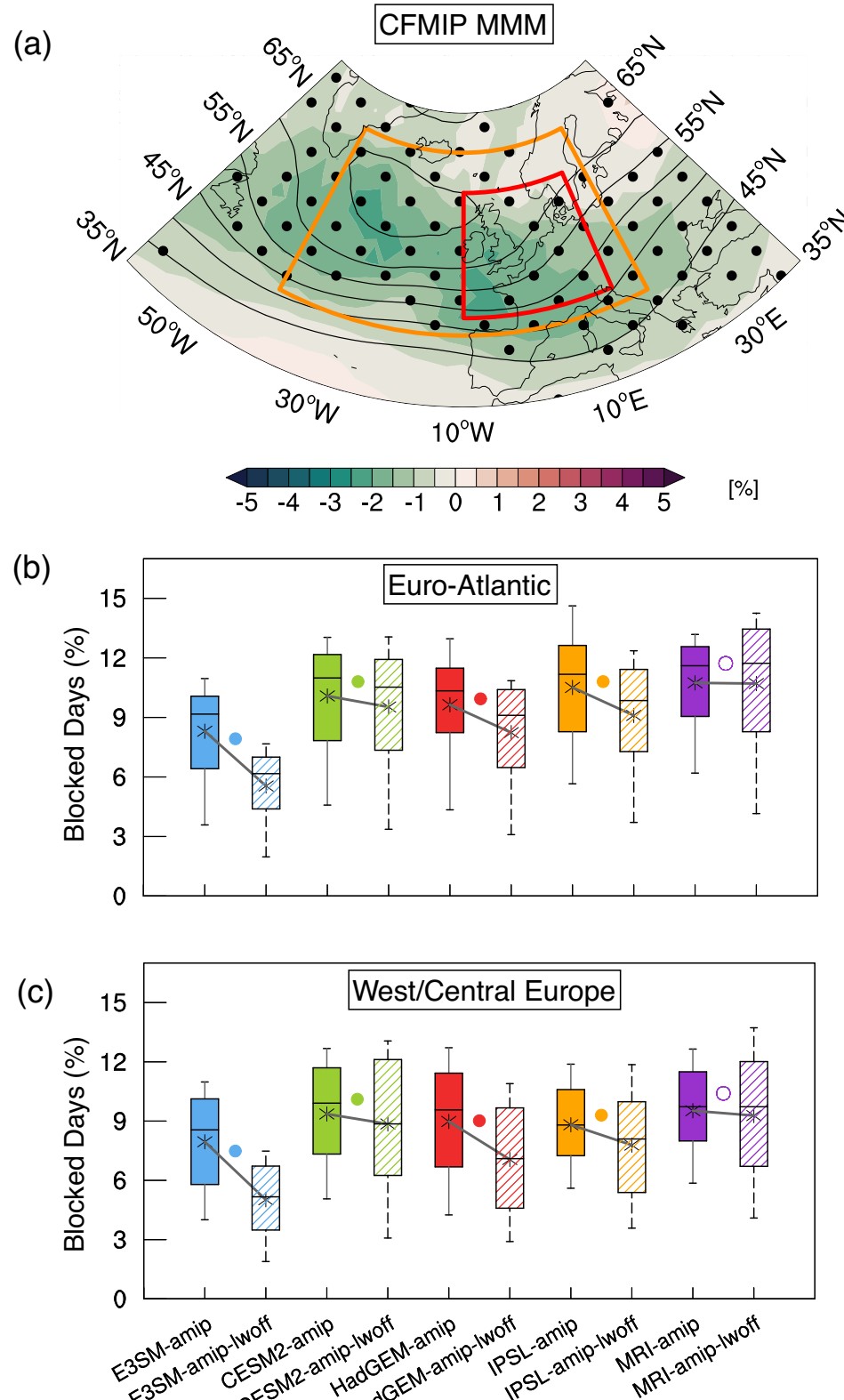

blocking events occur more frequently in CTL than in CLOCK and LWOFF.

To further understand which processes are responsible for the LWA increases in response to CREs, we compare the LWA budget terms during the onset of the blocks, as shown in Fig. 5. We define the upstream (downstream) region to the west (east) relative to the block to better quantify the contribution of different processes. CTL

captures the budget reasonably well compared to MERRA2 (Fig. 5d) (note that the results remain insensitive to the precise location of the polygon). In the upstream region (Fig. 5d), the contribution of $\langle \dot{A} \rangle \cos\phi$ to the increase in LWA is highest in the CTL (-7 m s$^{-1}$ day$^{-1}$), with a significant reduction in CLOCK (~ 4 m s$^{-1}$ day$^{-1}$) and in LWOFF (~ 2 m s$^{-1}$) (Fig. 5a–d). This indicates stronger nonconservative sources of LWA with interactive CRE. Additionally, the low-level source term

**Fig. 2 | Summary statistics from the Clouds On/Off Klimate Intercomparison Experiment, phase 2 (COOKIE-2). a** Differences in blocking frequency between the Atmospheric Model Intercomparison Project (AMIP) and AMIP longwave cloud radiative effect–off (AMIP–LWOFF) simulations from the multi-model mean (color shading). Contour lines show the ensemble mean of AMIP simulations, drawn with intervals of 2 starting at 2. Black dots mark regions where at least 75% of models agree on the sign of change. **b** Domain-averaged blocking frequency over the Euro-Atlantic sector (43°–65°N, 40°W–20°E; orange box in (**a**)) for each model. Lines connect paired AMIP and AMIP–LWOFF runs, with filled (open) circles indicating

statistically significant (not significant) differences at the 95% level. For comparison, the Energy Exascale Earth System Model (E3SM) control (CTL) and longwave cloud radiative effect–off (LWOFF) simulations are also shown, following a similar protocol but with prescribed sea surface temperatures (SSTs) in year-2000s conditions. The vertical widths of boxes represent the interquartile range, whiskers extend from 5% to 95%, and horizontal lines and * symbols indicate median and mean values, respectively. **c** As in (**b**) but averaged over West/Central Europe (45°–60°N, 10°W–15°E; red box in (**a**)). Colors denote models, with filled (raster) boxes indicating AMIP (AMIP–LWOFF).

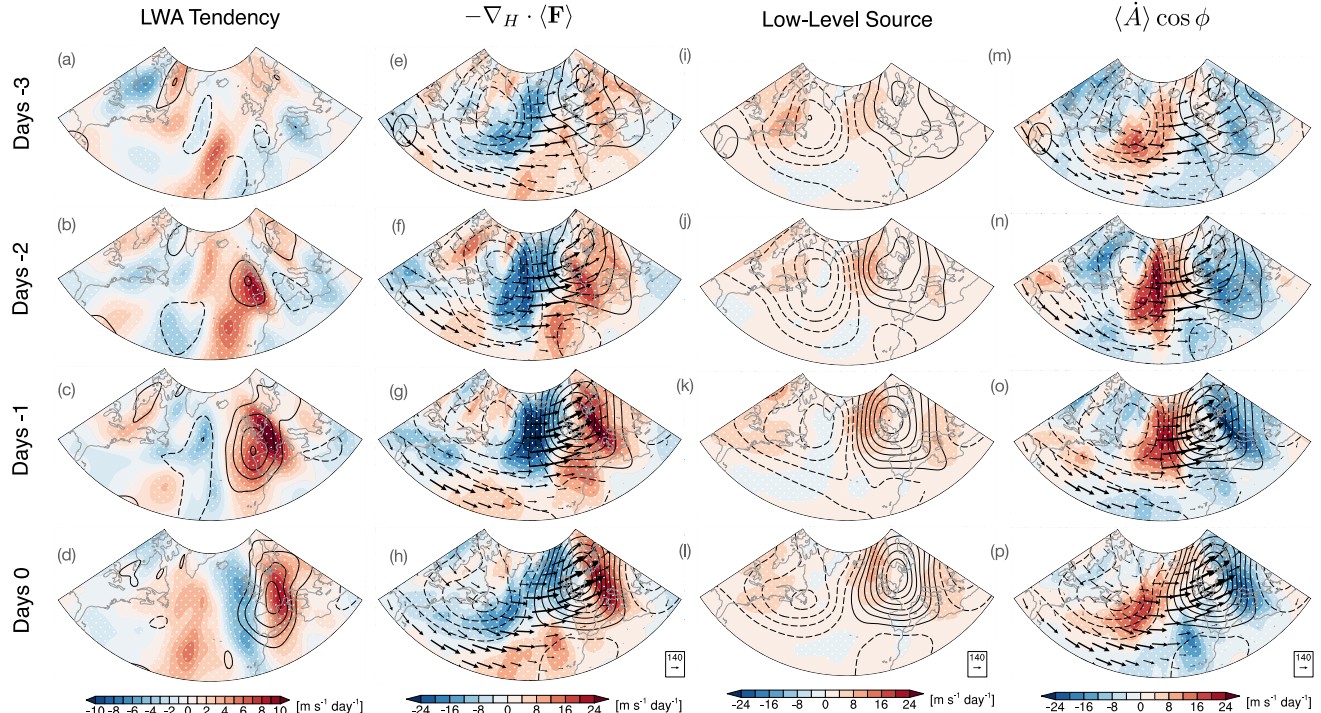

**Fig. 3 | Development of Euro-Atlantic blocking in the control (CTL) run based on column local wave activity (LWA) budget. a–d** Composite anomalies of column LWA budget terms (shading; units: m s$^{-1}$ day$^{-1}$) from the onset to the mature stage of the block (days −3 to 0). Each column from left to right corresponds to an LWA tendency term: **a–d** net tendency; **e–h** vertically averaged horizontal LWA flux (arrows; $F_\lambda$, $F_\phi$; units: m$^2$ s$^{-2}$) and its convergence (shading; $-\nabla_H \cdot \langle \mathbf{F} \rangle$); **i–l** meridional eddy heat flux at the base of the atmosphere (low-level source); and **m–p** nonconservative (diabatic) source or sink of LWA ($\langle \dot{A} \rangle \cos \phi$, residual). Contour lines in (**a–d**) denote LWA anomalies (units: m s$^{-1}$; interval: 5 m s$^{-1}$), while contour lines in (**e–p**) represent geopotential height anomalies (units: m; interval: 50 m). Stippling indicates regions where anomalies are statistically significant at the 95% level based on a bootstrap test.

remains weakly positive across all experiments, with the largest values found in CTL. The stronger low-level source upstream in the CTL is consistent with the stronger anticyclonic circulation (i.e., block amplitude) in this region (see the contour lines in Fig. 5a–c), resulting in stronger southerly flow ($v_e > 0$) and perturbation potential temperature ($\theta_e > 0$). In contrast, the contribution of the horizontal LWA flux convergence ($-\nabla_H \cdot \langle F \rangle$) is negative in all experiments, signifying that the divergence of LWA partially offsets the surplus $\langle \dot{A} \rangle \cos \phi$ from the upstream region.

In the downstream region (Fig. 5e), the horizontal LWA flux convergence exhibit the highest positive values in CTL (~ 10 m s$^{-1}$ day$^{-1}$), with CLOCK and LWOFF showing low values (~6 m s$^{-1}$ day$^{-1}$ and ~3 m s$^{-1}$ day$^{-1}$, respectively), indicating 40% and 70% reductions relative to CTL. The substantial increase in horizontal LWA flux convergence in the CTL is consistent with the significant increase of $\langle \dot{A} \rangle \cos \phi$ in the upstream region (Fig. 5d), which is mainly dominated by diabatic processes (Figs. S5–S6 and later in Figs. 6–7). This suggests that the upstream diabatic source increases horizontal convergence of LWA flux downstream, consistent with Neal et al.[12]. In contrast, the contribution of $\langle \dot{A} \rangle \cos \phi$ downstream is negative in all experiments, with CTL showing the greatest negative value (~ -6 m s$^{-1}$ day$^{-1}$). The large negative values are associated with the dissipation of wave activity due to radiative

damping, irreversible mixing, and surface friction downstream of the block (Fig. S6). Overall, the results suggest that CRE increases LWA primarily by enhancing the upstream diabatic wave activity source $\langle \dot{A} \rangle \cos \phi$. LWA is then advected by the zonal flow downstream and converges to form localized blocking structures.

Next, we examine which diabatic processes associated with changes in interactive CREs contribute to the enhanced $\langle \dot{A} \rangle \cos \phi$ upstream. To quantify these contributions, we explicitly calculate the diabatic source and sink of wave activity ($\Delta \Sigma$) due to different diabatic heating rates ($\dot{\theta}$) in the reanalysis and across the three experiments (Fig. 6). A comparison between CTL and MERRA2 shows that CTL captures the vertical structure and column-averaged $\Delta \Sigma$ reasonably well, albeit with a few differences (Fig. 6). The LH contribution to $\Delta \Sigma$ in CTL is maximized upstream (i.e., to the west) of the block center between 500–400 hPa (Fig. 6b, c), while the LWCRE contribution to the diabatic wave source is centered between 600–300 hPa upstream of the block (Fig. 6g, h). This region of enhanced diabatic forcing corresponds to the ascending branch of the warm conveyor belt (WCB), which is characterized by strong vertical and poleward transport of moist air (see later in Fig. 7a, e, b, f). However, the upstream diabatic (LH) forcing at lower levels (950–800 hPa) is not adequately captured by CTL (Fig. 6b, c), which may stem from the model's

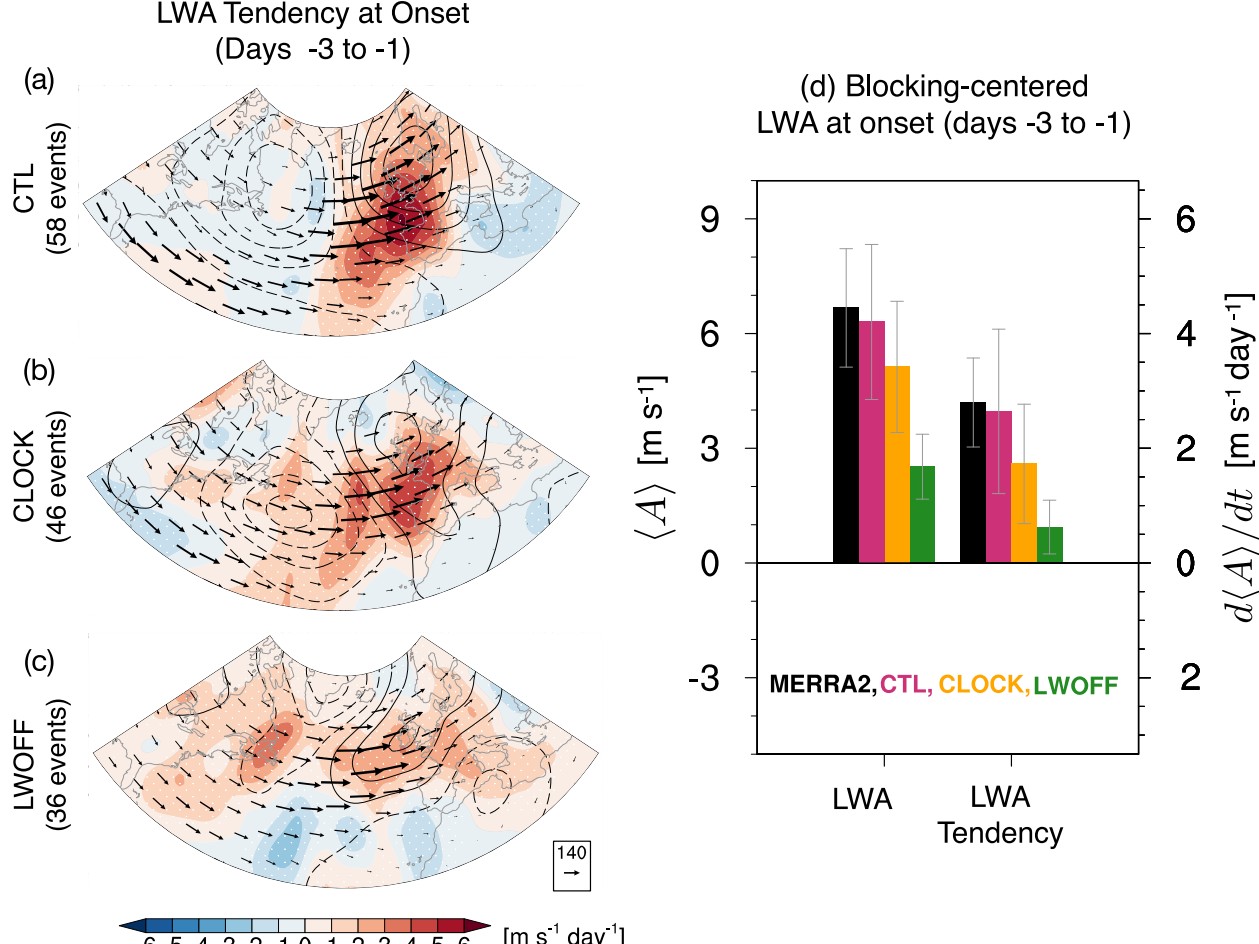

**Fig. 4 | Column local wave activity (LWA) and its tendency during the onset of Euro-Atlantic blocking. a–c** Composite of column LWA tendency (shading, unit: m s⁻¹ day⁻¹) and 500-hPa geopotential height anomalies (contours, unit: m) averaged during the onset period (days − 3 to −1) for (**a**) control (CTL), **b** cloud-locking (CLOCK), and **c** longwave cloud radiative effect−off (LWOFF). Vectors indicate vertically averaged horizontal LWA flux (arrows; $F_\lambda$, $F_\phi$; units: m² s⁻²). Stippling

marks regions statistically significant at the 95% level from a bootstrap test. **d** Blocking-centered averages of the composite LWA anomaly (units: m s⁻¹) and its tendency during the onset period from the MERRA-2 and model simulations. The average is computed over a 10° × 10° area centered on the blocking high (i.e., the local maximum of geopotential height anomaly) from each map in (**a–c**). Error bars in (**d**) indicate ± 1 standard deviation.

limitation in representing low-level clouds and the associated LH upstream during the blocking onset (see later in Fig. 7a, b).

By comparing the ΔΣ across all model experiments, we find that the contributions of longwave cloud radiative heating and LH to ⟨Ȧ⟩ cos φ are largest in CTL, with LH making the dominant contribution (Fig. 6). In CLOCK, the LWCRE contribution is significantly reduced by ~ 57% (−0.15 m s⁻¹ day⁻¹ compared to 0.35 m s⁻¹ day⁻¹ in CTL), and the diabatic wave source upstream between 500-300 hPa is inactive (Fig. 6i). This reduction in LWCRE contribution to ΔΣ is expected because by design, the covariation between CRE and individual weather systems are decoupled. As a result, the modulation of LWCRE by blocking in CLOCK is suppressed. Similarly, the LH contribution to ΔΣ in CLOCK is reduced by ~ 46% when the CRE is locked (1.3 m s⁻¹ day⁻¹ in CLOCK compared to ~ 2.4 m s⁻¹ day⁻¹ in CTL) (Fig. 6c, d), further highlighting the role of interactive CRE in amplifying wave activity during blocking onset. Making clouds transparent to longwave radiation eliminates the impact of LWCRE on ΔΣ (Fig. 6a, j) and significantly weakens the LH contribution (Fig. 6a, f). This is consistent with the reduced LWA in LWOFF (Fig. 4), indicating the importance of LWCRE in amplifying wave activity during blocking onset.

The variations in the strength of upstream diabatic wave sources discussed above are closely tied to the amplitude and vertical extent of anomalous diabatic heating during the development of the block (Fig. 7a–h). In CTL, similar to MERRA2, enhanced ΔΣ (Fig. 6c,h) is

associated with intensified LH and CRE within the WCB region (Fig. 7a–b, e, f; Figs. S7–S8), consistent with stronger vertical aloft and poleward moisture transport in this region. Comparison of diabatic heating terms across model experiments reveals a stronger positive LH anomaly upstream (west of the block center) in CTL relative to CLOCK and LWOFF (Fig. 7b–d, S9). This is consistent with stronger LH-induced ΔΣ upstream in CTL compared to the other experiments (Fig. 6b). Similarly, the longwave cloud radiative heating anomaly is stronger in CTL than in other experiments, with positive heating between 850-550 hPa concentrated west of the block center (Fig. 7e–h and Fig. S9). The stronger positive anomalies in LH and LWCRE in the experiment with interactive CRE align with our findings in Fig. 6, which show that LH and LWCRE contribute most to the diabatic source of wave activity upstream, leading to stronger zonal LWA convergence downstream (Fig. 5e and Figs. S3, S5).

The reduction of LH in CLOCK and LWOFF in the absence of CREs can be explained by the lack of LWCRE impacts on large-scale ascent and moisture transport. During the onset period, LWCRE significantly warms the mid-troposphere (850-450 hPa) in the WCB region upstream in CTL (Fig. 7f), whereas this warming is weaker in CLOCK (Fig. 7g) and absent in LWOFF (Fig. 7h). In the absence of LWCRE-induced warming, cooler mid-tropospheric temperatures steepen the vertical temperature gradient, enhancing static stability and weakening large-scale ascent. This weaker ascent is consistent with reduced

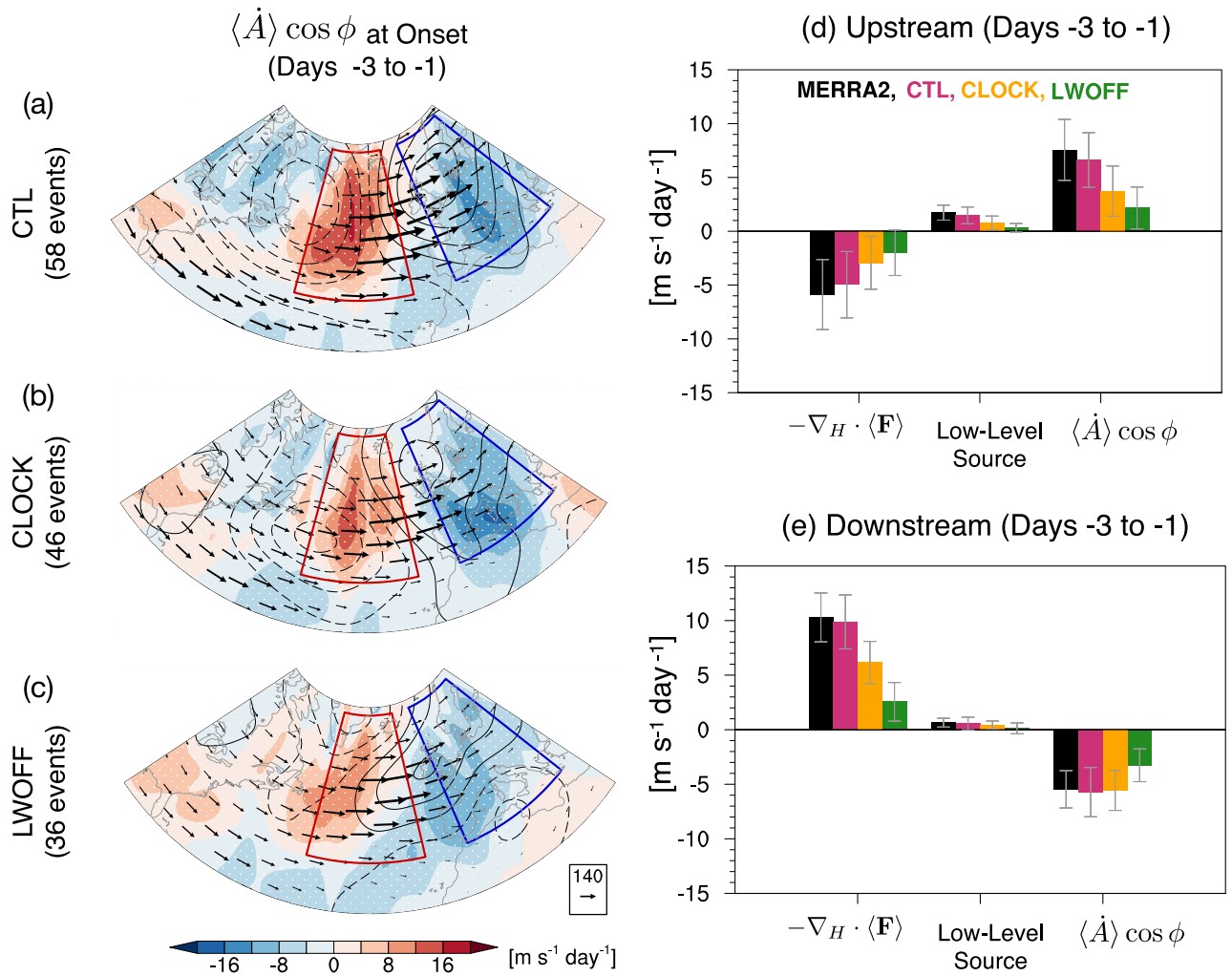

**Fig. 5 | Column local wave activity (LWA) budget terms during the onset of Euro-Atlantic blocking. a–c** Composite of the nonconservative source/sink of wave activity ($\langle \dot{A} \rangle \cos \phi$; shading, units: m s$^{-1}$ day$^{-1}$) and 500-hPa geopotential height anomalies (contours, units: m) averaged during the onset period (days $-3$ to $-1$) for (**a**) control (CTL), **b** cloud-locking (CLOCK), and (**c**) longwave cloud radiative effect–off (LWOFF). Vectors indicate vertically averaged horizontal LWA flux (arrows; $F_\lambda$, $F_\phi$; units: m$^2$ s$^{-2}$). Stippling marks regions statistically significant at the 95% level from a bootstrap test. **d** Domain-averaged composite LWA budget terms in the upstream region (red polygon: 35°–68°N, 46°W–16°W in panels a-c) from the MERRA-2 and model runs. **e** As in (**d**) but for the downstream region (blue polygon: 35°–68°N, 5°W–25°E in (**a–c**). Error bars indicate $\pm 1$ standard deviation.

negative vertical velocity ($\omega$) anomalies in CLOCK (Fig. 7i) and LWOFF (Fig. 7j) in the WCB region with enhanced LH (see green box), indicating less efficient lifting of moist air into the mid-troposphere. This reduction in ascent is accompanied by weaker vertically integrated moisture transport (IVT) into the WCB (Figs. 7k–l, S10), confirming that less moisture is being advected poleward upstream of the block. The weakening of moisture transport in both CLOCK and LWOFF is consistent with the reduced mean moisture flux upstream (over the eastern North Atlantic) (see later in Fig. 8g, h). As a result, condensation is suppressed, leading to a decline in LH. This is consistent with previous studies showing that LWCRE plays an important role in intensifying vertical motion and LH[37,40,41,54].

Another possible pathway is that LWCRE enhances LH through an evaporation feedback. Using observational-based MERRA2 data, we show that during the onset period, positive anomalies in longwave cloud radiative heating, LH, and surface evaporation are partially co-located in the upstream region of the block (green box, Fig. S11). This positive LWCRE anomaly upstream in the WCB is primarily associated with increased high cloud cover, with a secondary contribution from low-to-mid-level clouds (Fig. S12). An increase in high clouds can effectively trap downward LW radiation, which could lead to enhanced

surface temperature and latent heat flux (LHF). Positive LHF indicates increased moisture transfer from the surface to the atmosphere, supporting enhanced LH aloft. This increase in LHF (Fig. S11e) is primarily driven by stronger surface wind and enhanced near-surface vapor gradient ($q_s - q_a$) upstream of the block region (Figs. S11g, h). The latter can be associated with surface warming linked to CRE during the blocking onset (Fig. S11f), which increases the saturated specific humidity ($q_s$) and hence the moisture gradient (Fig. S11g). Removing this effect would suppress LHF and consequently weaken LH, consistent with previous studies[40]. While these reanalysis-based diagnostics support the plausibility of this pathway, they cannot fully isolate the role of LWCRE from other contributing processes. Currently, our model simulations were conducted with prescribed SSTs, which limits this surface feedback, and the necessary surface diagnostic variables were not archived to directly evaluate this pathway. Therefore, future investigation in a separate modeling study, particularly those using interactive SSTs, is needed to robustly evaluate this mechanism.

In summary, interactive CREs promote blocking formation mainly by enhancing upstream diabatic wave source, both through its direct influence through LWCRE and indirectly via its impact on LH in the

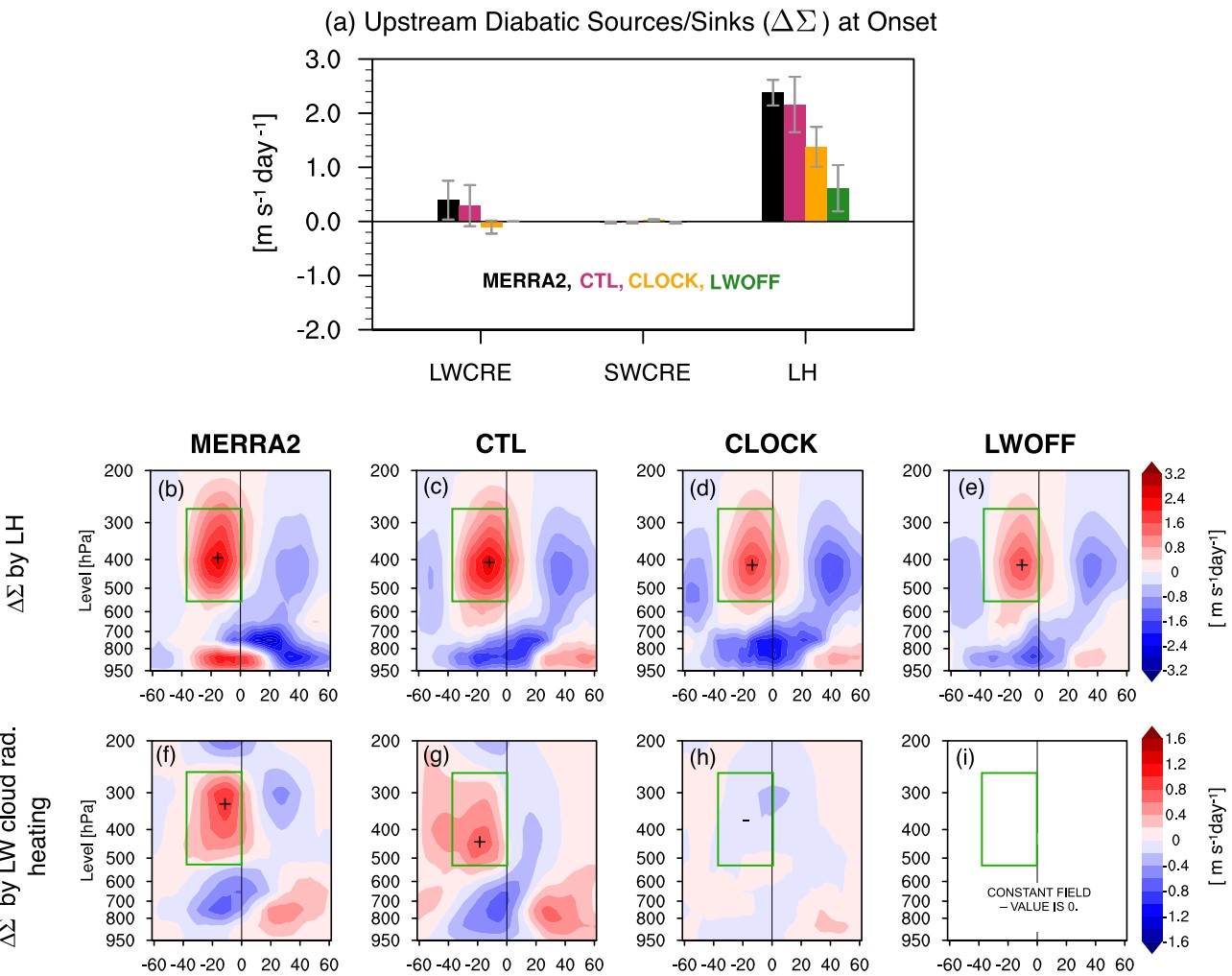

**Fig. 6 | Contribution of diabatic heating to nonconservative sources and sinks of Local Wave Activity (LWA) during blocking onset. a** Anomalous diabatic sources and sinks of wave activity ($\Delta\Sigma$; m s⁻¹ day⁻¹) from longwave cloud radiative effect (LWCRE), shortwave cloud radiative effect (SWCRE), and latent heating (LH), averaged during the onset period (days −3 to −1) over the upstream region (red polygon in Fig. 5a–c). Error bars show ±1 standard deviation. **b–i** Zonal cross-sections of blocking-centered composites, averaged over 10° latitude relative to the block center. Shading (m s⁻¹ day⁻¹) shows diabatic anomalies due to (**b–e**) LH and (**f–i**) LWCRE. SWCRE is omitted due to its negligible magnitude. The color bar for LH is scaled to twice that of LWCRE. Vertical lines mark the block center, and green boxes in (**b–j**) highlight the upstream domain of strong diabatic sources. Positive (negative) values denote sources (sinks). Plus and minus signs denote maxima of diabatic heating and cooling upstream of the block.

WCB region upstream of the block, with the latter playing a dominant role. The LWA from the upstream diabatic source is then advected by the zonal flow and converges downstream to form blocks. The significant reduction in blocking frequency in the CLOCK and LWOFF experiments underscores the importance of CREs in these processes.

**Influence of CREs on blocking through background circulation**
One might question whether changes in CREs could induce changes in background mean state that affects blocking formation. Previous studies have found that blocking prevalence is influenced by the strength and latitude of the jet[5,49], as well as the intensity of background stationary waves[5,22,50]. In particular, reduced blocking frequency is often linked to a poleward jet shift and a weakened quasi-stationary ridge[5,22,49,50]. Figure 8a–d shows the responses of the mean jet (Fig. 8a–b) and background stationary wave (Fig. 8c–d) in CLOCK and LWOFF relative to CTL. The CLOCK experiment exhibits a weak poleward jet shift (Fig. 8a) and a partial weakening of the planetary-scale anticyclonic ridge associated with stationary waves over Europe (Fig. 8c). In contrast, LWOFF exhibits a more pronounced poleward jet shift (Fig. 8b) and a stronger weakening of the quasi-stationary ridge

over Europe. These background circulation changes in response to CREs are consistent with previous studies showing that CREs play an important role in shifting the jet poleward by modifying meridional temperature gradients and static stability[37–39,42]. The poleward shift of the jet and weaker quasi-stationary ridge provide unfavorable conditions for blocking to form over Europe[22,49]. This finding broadly supports previous studies showing that the weaker quasi-stationary ridge and stronger PV gradients, associated with a poleward-shifted jet, increase the jet's carrying capacity for wave activity, making a block more difficult to form for a given level of incident transient wave-activity flux[53,55]. As a result, downstream LWA flux convergence is reduced, consistent with weaker wave activity and lower blocking frequency in these two experiments relative to CTL (Figs. 1, 5e).

The reduction in blocking formation in the CLOCK and LWOFF experiments is consistent with weaker climatological low-level baroclinicity over the storm-track region (eastern North Atlantic), as indicated by the Eady growth rate (EGR) (Fig. 8e–f). Although both experiments show reduced EGR, the decrease over this region is much larger in CLOCK than in LWOFF. This suggests that disabling LWCRE causes a pronounced weakening of low-level baroclinicity, thereby

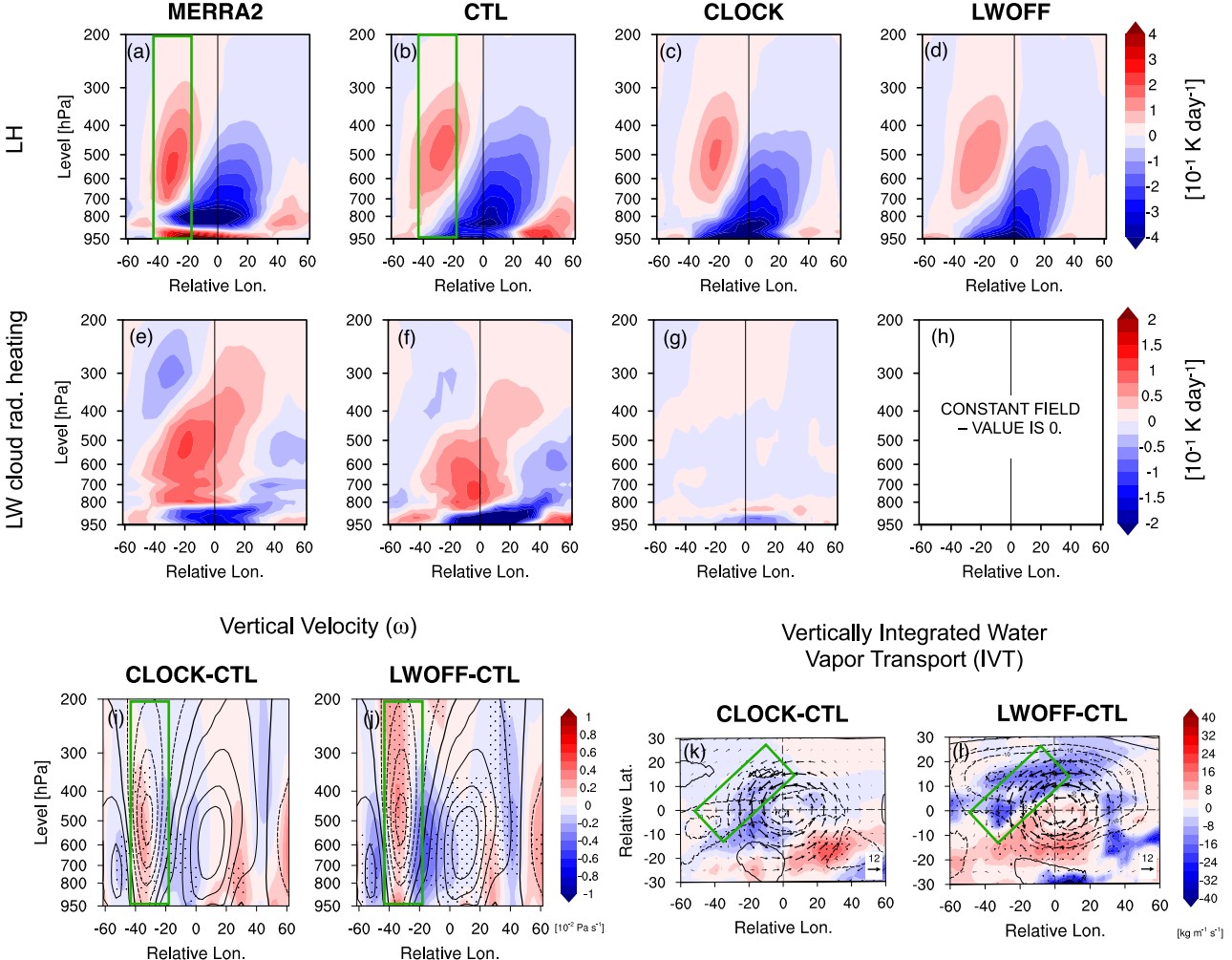

**Fig. 7 | Blocking-centered composites of diabatic heating, vertical wind velocity, and moisture transport anomalies during blocking onset. a–d** Zonal cross-sections of latent heating (LH) anomalies (shading; units: 10⁻¹ K day⁻¹) from (**a**) the MERRA-2, **b** the control (CTL) run, **c** the cloud-locking (CLOCK) run, and (**d**) the longwave cloud radiative effect–off (LWOFF) run. **e–h** As in (**a–d**) but for longwave cloud radiative effect (LWCRE) anomalies. Anomalies are averaged during the onset period (days −3 to −1) over a 10° latitude band relative to the block center. **i–j** Composite differences of vertical wind anomalies (shading; units: 10⁻² Pa s⁻¹) for (**i**) CLOCK-CTL and (**j**) LWOFF-CTL, with contours showing the control (CTL). **k–l** As in (**i–j**), but showing latitude–longitude composites of integrated vapor transport (IVT) anomalies (shading; units: kg m⁻¹ s⁻¹). Contours indicate geopotential height differences, and vectors show water vapor flux differences. The dashed lines mark the block center. Green boxes in (**a, b, i, j, k, l**) highlight the upstream LH region. Stippling in (**i–l**) marks differences from CTL significant at the 95% level based on a *t*-test.

reducing the atmosphere's capacity to generate and sustain transient eddy activity, which in turn diminishes blocking formation. The reduction in the climatological mean of low-level baroclinicity is also consistent with weaker low-level baroclinic source (i.e., eddy heat flux divergence) in CLOCK and LWOFF relative to CTL during blocking onset (Fig. 5d, e).

In addition to the baroclinicity changes, removing CREs also significantly weakens climatological moisture transport upstream (eastern North Atlantic) in CLOCK and LWOFF relative to CTL (Fig. 8g, h). This reduced moisture transport is likely related to weaker stationary wave ridge that modulates poleward moisture transport from low latitudes (Fig. 8c, d). This weakening of climatological moisture transport, in turn, accounts for the reduced poleward moisture transport anomaly upstream during the blocking lifecycle (Fig. 7k, l), leading to reduced LH upstream and ultimately less blocking formation in CLOCK and LWOFF.

Taken together, changes in CREs can influence blocking formation indirectly by modifying the background mean state, including the jet, quasi-stationary ridges, mean baroclinicity, and moisture transport. These mean-state modifications explain the stronger reduction in

LWA and blocking frequency in LWOFF, where the background circulation is substantially modified, compared to CLOCK, where it is only minimally altered.

## Discussion

This study demonstrates that CREs significantly increase the occurrence of atmospheric blocking events in the Euro-Atlantic sector. This conclusion is supported by different two techniques used in CRE experiments: (1) cloud locking, which removes cloud-radiation-circulation interactions with minimal disruption to the mean state, and (2) disabling LWCRE, which substantially alters the mean CRE and consequently, the mean state. Specifically, the results show that interactive CREs significantly increase the Euro-Atlantic blocking frequency by up to 1.67% blocked days per winter. This suggests that high-frequency covariance between CRE and dynamics can increase the formation of the block, even with minimal changes in the mean state. A qualitatively similar result was also found by disabling LWCRE, albeit with a larger impact on blocking formation (by up to 2.88% blocked days per winter) due to substantial changes in the mean state. This result is further corroborated by our analysis from multi-model

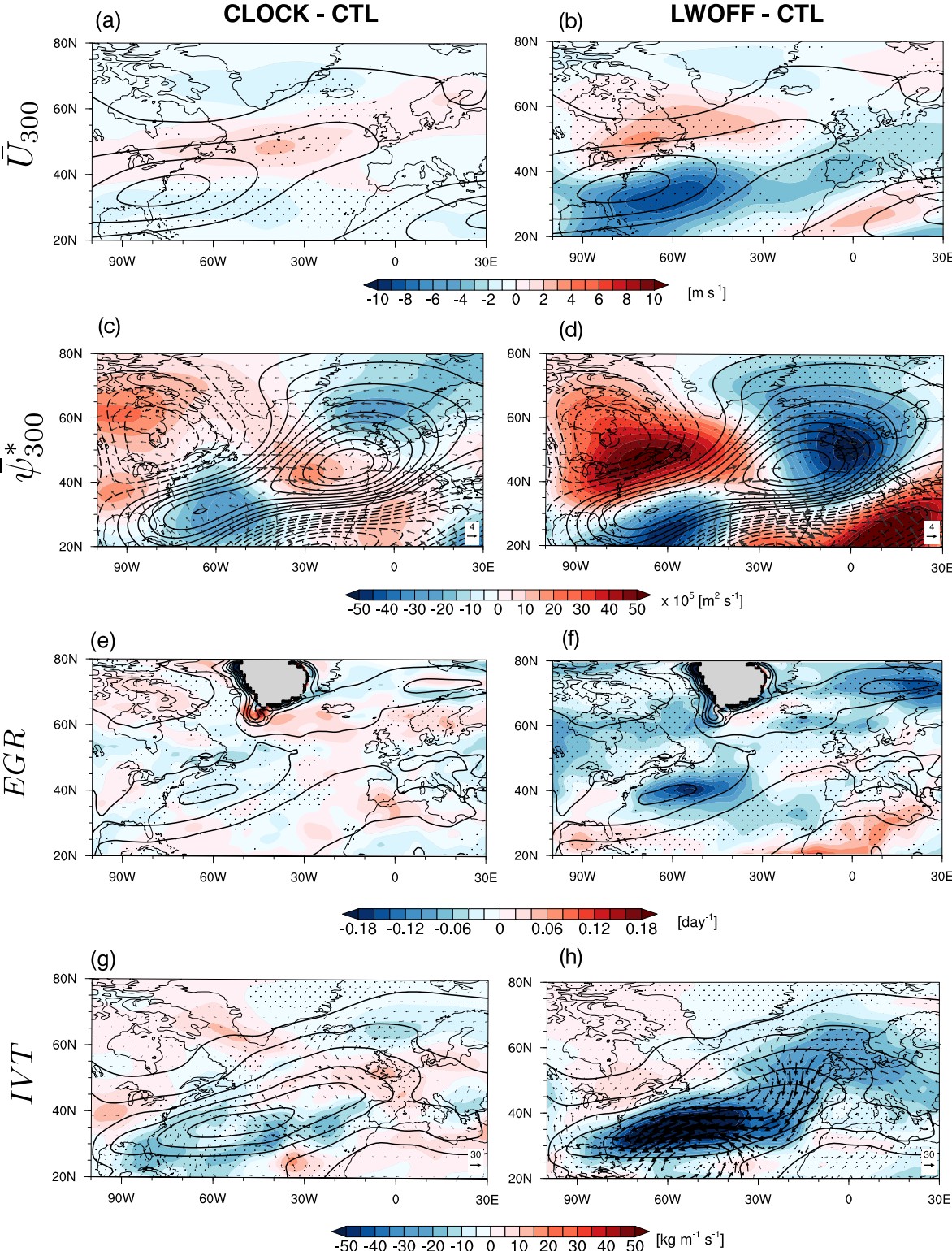

**Fig. 8 | Impacts of cloud radiative effects (CREs) on the climatological mean of North Atlantic zonal winds, stationary waves, low-level baroclinicity, and moisture transport. a, b** Differences in the wintertime 300-hPa zonal wind (shading; m s⁻¹) for (**a**) cloud-locking (CLOCK) and (**b**) longwave cloud radiative effect−off (LWOFF), relative to the control (CTL) run. **c, d** As in (**a, b**), but for differences in the 300-hPa stationary wave streamfunction ($\psi^*_{300}$; shading; $10^5$ m² s⁻¹) with Plumb flux vectors (m² s⁻²). **e, f** As in (**a, b**), but for differences in the maximum Eady growth rate (EGR) at 850 hPa (shading; day⁻¹). **g, h** As in (**a, b**), but for differences in vertically integrated vapor transport (IVT; shading; kg m⁻¹ s⁻¹), with vectors showing water vapor flux. Stippling denotes regions where differences from CTL are significant at the 95% confidence level based on a $t$-test. Bold contours in all panels show the CTL climatology of each field.

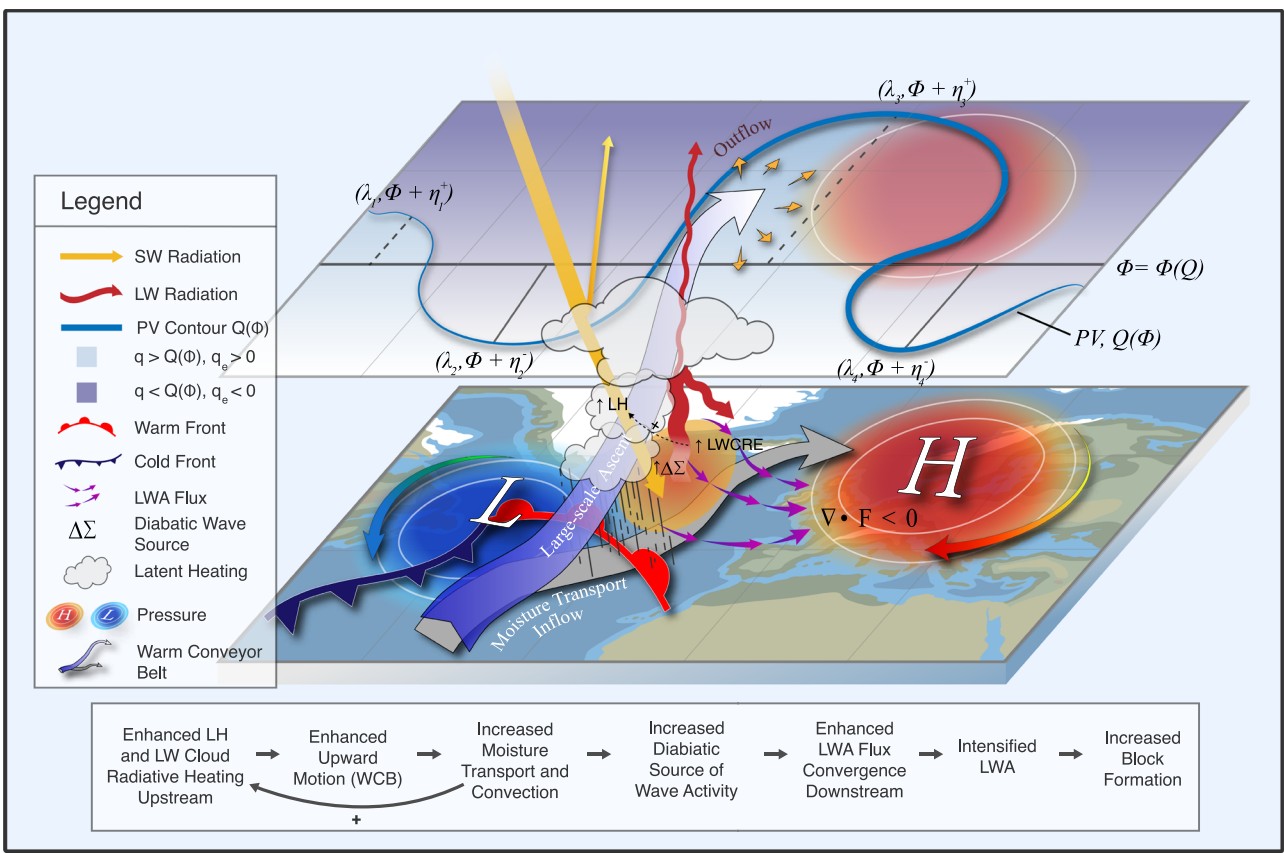

**Fig. 9 | Schematic diagram illustrating how cloud radiative effects (CREs) enhance upstream diabatic heating and promote block formation.** In the warm conveyor belt (WCB) upstream of the block, enhanced diabatic heating arises from latent heating (LH), driven by moisture transport inflow, and from longwave (LW) cloud radiative effect (LWCRE), as high clouds absorb outgoing LW (infrared) radiation. LWCRE further amplifies LH through a positive feedback by intensifying large-scale ascent and moisture transport within the WCB. This CRE-LH coupling strengthens the upstream diabatic source of wave activity ($\Delta\Sigma$), which increases downstream convergence of horizontal local wave activity flux and hence, favors blocking formation. In the absence of LWCRE, this feedback is weakened, leading to reduced ascent, moisture transport, and LH, and thereby suppressing the diabatic wave source and its downstream impact.

experiments under the CFMIP project (COOKIE2), where clouds were made transparent to LWCRE, leading to a significant reduction in blocking events. Taken together, these results highlight that LWCRE substantially impacts blocking formation, independent of the specific model used.

Using the finite-amplitude LWA diagnostic, we further examine how CREs influence blocking formation throughout its lifecycle. The results reveal that cloud-radiation interactions enhance diabatic wave forcing in the upstream block region, where clouds dominate within ascending WCB airstreams—both directly through longwave heating and indirectly through feedback on latent heating, with the latter playing the dominant role. The enhanced LH in the presence of CREs arises because LW cloud radiative heating in the mid-troposphere amplifies large-scale ascent in the WCB, which strengthens poleward moisture transport and thereby further increases LH upstream of the block (Fig. 7). The resulting increase in the upstream wave activity source is then redistributed by horizontal LWA convergence (Figs. 5, 6), leading to LWA accumulation downstream that promotes blocking formation (Figs. 1, 4). Recent work by Neal et al.[12] also highlighted the importance of diabatic wave activity generation within the WCB region of an upstream cyclone in developing a downstream blocking anticyclone. Our results advance this understanding by showing that, in addition to cloud LH, the LW cloud radiative heating and its feedback on LH play an important role in amplifying the diabatic source of wave activity upstream. The sequence of processes by which CREs enhance upstream diabatic sources during blocking onset is summarized schematically in Fig. 9.

In addition to the sizable indirect impact of CREs on blocking through LH, the results also reveal that the indirect effect of CREs— through the shift of the mean jet and the strength of stationary waves, mean baroclinicity, and mean poleward moisture transport—influences blocking formation (Fig. 8). Disabling LWCRE causes a significant poleward shift of the jet, weakens stationary high-pressure ridges over Europe, reduces low-level baroclinicity, and weakens poleward moisture transport upstream[5,22,49,50], which altogether create conditions unfavorable for blocking formation. This is consistent with previous studies showing that the weaker quasi-stationary ridges and stronger PV gradients (associated with a poleward-shifted jet) increase the carrying capacity of the jet stream to carry wave activity, thereby limiting the pile-up of LWA that signifies blocking formation[53,55]. This mechanism is consistent with the reduced downstream LWA convergence during the onset period in both simulations relative to CTL (Fig. 5).

The findings of this study not only corroborate previous research emphasizing the role of upstream diabatic heating associated with LH in the WCB on blocking formation[12,31,32,34,35], but also elucidate the important role of cloud-radiation interactions in the WCB. From a PV-based perspective, the enhanced upstream diabatic LWA source, associated with CREs and their feedback on LH, can be interpreted as enhanced diabatic PV source in the WCB region upstream of the block, which in turn amplifies negative PV anomalies aloft (i.e., the ridge) in the upper troposphere[12,56]. Such ridge amplification in our results can be explained by the enhanced downstream horizontal LWA convergence due to the CREs impacts

on the background wind and quasi-stationary ridge (Fig. 8), which together decreases the carrying capacity of the jet stream to carry wave activity. This convergence, in turn, leads to a pile-up of LWA downstream, corresponding dynamically to the amplification of the upper-level ridge (negative PV anomaly) (see Fig. S13). Thus, this proposed mechanism based on the LWA perspective is consistent with the indirect diabatic impact from PV-based thinking, while emphasizing the critical role of CREs and their feedback on LH in enhancing the upstream diabatic wave source essential for blocking formation.

Fully explaining the jet-storm track and stationary wave responses to cloud locking or disabling CREs remains a challenge in the community, as research on the diabatic dynamical effects of clouds on midlatitude circulation is still in its early stage[39,41,42]. For example, Lu et al.[42] recently demonstrated that cloud locking can lead the combined effect of two wave sources (diabatic and baroclinic) for the upper-level wave propagation to shift poleward. Nonetheless, their results mainly presented a dynamical consistency, rather than a clean causality. In our results, although CRE-induced changes in the mean wind, baroclinicity, and stationary waves are robust and favor the blocking formation, the focus of the mechanistic inquiry is more on the direct diabatic effect of the CREs. A comprehensive explanation of the broader jet and stationary wave adjustments to CREs lies beyond the scope of the current manuscript, which underscores the importance of this line of research.

Recent studies highlight that moist dynamic processes in WCBs significantly impact medium-range predictability in the extratropics, particularly due to their role in developing downstream ridges and initiating blocking over the Euro-Atlantic sector[35,44,57,58]. Our findings emphasize that, in addition to accurately simulating WCBs, capturing cloud-radiation-circulation interactions within the WCB is also crucial, as CREs can significantly affect the moist dynamic processes and the formation of blocking. This suggests that improving the representation of CREs, particularly within WCBs, could potentially improve the simulation and prediction of blocking in medium-range weather forecasts.

Many climate models continue to struggle to accurately capture complex cloud processes, leading to difficulties in representing CREs. Instead of explicitly resolving cloud microphysics and radiative processes, models typically rely on parameterizations[39,59]. Consequently, inaccuracies in representing these processes can obscure the role of cloud-radiation-circulation interactions, which may potentially contribute to biases in the spatial and temporal characteristics of blocking. Improving the representation of CREs, particularly through better simulating cloud processes, could therefore help reduce such biases and enhance the simulation of blocking.

Recall that our experiments use fixed lower-boundary SST, which enables a more direct attribution of blocking responses to CREs by minimizing confounding effects from SST-related forcing. In the real world, however, CREs can interact with SSTs, and they can jointly influence blocking formation through coupled air-sea feedbacks[25,26]. Therefore, future efforts should focus on inter-model comparisons of cloud-locking experiments, including those with interactive SSTs and accounting for CRE uncertainties, to establish consensus on the effect of CREs' dynamical feedback on blocking.

Atmospheric blocking often causes high-impact weather extremes such as heat waves, droughts, cold outbreaks, and floods in the mid-latitudes[4,6–11]. Therefore, improving their simulation and prediction by minimizing CRE-related model biases could potentially enhance early-warning systems to mitigate the damage caused by such events.

## Methods
### Numerical experiments
We use the U.S. DOE's Energy Exascale Earth System Model v1 (E3SM[45]) to investigate the contribution of cloud radiative effects (CREs) on the midlatitude atmospheric blocking. All experiments are run in the "AMIP" configuration with realistic land-sea geography, an active land model, and prescribed annually repeating sea surface temperatures (SSTs) and sea-ice concentrations (based on a 20-year monthly climatology centered on year 2000 of the real world)[46]. The SST pattern is repeated annually, so that interannual variability (e.g., El Niño−Southern Oscillation) is not included. Decoupling cloud radiative forcing from SST-driven changes allows for a clearer isolation of the impact of CREs on blocking dynamics, with minimal interference from lower-boundary SST-related forcing. The control run (CTL) uses the atmospheric forcing representative of the present-day climate conditions (using the initial condition of the year 2000 of the free-running E3SM v1 DECK AMIP simulation)[46]. The CTL run lasts 20 years following a 30-year spin-up to ensure global soil moisture equilibrium.

The cloud-locking experiment (CLOCK) is identical to CTL, except the cloud properties are "locked" to isolate the interactive CRE. Specifically, we prescribe the cloud optical properties from the first three years of CTL simulation in the CLOCK simulation[46]. This choice follows the common practice in cloud-locking experiments[38,47] and is intended to retain a realistic and internally consistent CRE climatology while reducing computational burden[46]. Prior studies have shown that even a single year of cloud properties can be sufficient for a cloud-locking configuration (e.g., Middlemas et al.[47]), so we do not anticipate issues related to undersampling when using three years. In this way, the CLOCK experiment possesses a nearly identical CRE climatology to CTL, but the CREs are decoupled from the dynamics and other physical processes. This means no high-frequency covariance between CREs and circulation affects the blocking. This cloud-locking technique is similar to that used in previous studies[38,47]. The CLOCK run is also integrated for 20 years.

Because the cloud-locking experiment fundamentally differs from the complete CRE denial experiments, such as those conducted in the COOKIE project[60,61], we also perform one CRE denial experiment in the E3SM where LWCRE is disabled (referred to as LWOFF hereafter). In this setup, SSTs and sea ice conditions are kept the same as in the CTL experiment. However, unlike the CLOCK experiment, the LWCRE is deactivated so that the average CRE heating in the atmosphere changes, which then alters the mean circulation and the climate regimes[46].

Finally, we also compare the LWOFF experiment to the COOKIE-2 simulations of the Cloud-Feedback Model Intercomparison Project (CFMIP) Tier 2 in two different configurations: standard Earth-like configurations (AMIP) with interactive CREs and AMIP with disabled LWCRE (AMIP-LWOFF). The COOKIE-2 experimental protocol specifies that only the LWCRE is deactivated, while SSTs and sea-ice concentrations are prescribed and held identical between the control run and LWCRE-off simulations[61]. We analyze daily Z500 fields from these simulations at a spatial resolution of 2.5° × 2.5°. These simulations have been conducted by four modeling groups so far and are listed in Table 1.

### Reanalysis datasets
Daily gridded horizontal velocity, geopotential height, temperature, surface pressure, and temperature tendencies of the NASA MERRA-2 dataset[62], covering the period from 1980 to 2020 with a spatial resolution of 0.5° × 0.625° and 29 vertical levels, were obtained to assess the ability of the CTL run to simulate the large-scale circulation features, heating rates, and blocking activity. MERRA-2 was chosen here because it provides different diabatic heating rates (a fully gridded 3D heating derived from the assimilation of satellite observations and the GEOS-5 model), making it particularly well-suited for process-level diagnostics involving diabatic influences on wave activity[42]. Cloud radiative heating rates are diagnosed as all-sky radiative heating rates minus clearsky radiative heating rates.

## Detection of blocking

We employ a Dole and Gordon's blocking index[63], which is based on finding strong, stationary, and persistent positive daily 500-hPa geopotential height (Z500) anomalies. This index is also known as an anomaly-based blocking index[4,64,65] and has outperformed a few other Z500-based indices in identifying heat wave-causing weather patterns[10]. For this index, the Z500 anomalies are calculated for each grid point as the difference with respect to the climatological mean daily values of the analyzed period. Blocks are identified daily as the two-dimensional areas of at least $2 \times 10^6$ km$^2$ extensions with Z500 anomalies exceeding the threshold. The threshold is defined based on the 90th percentile of the Z500 anomaly distribution between 50° and 80°N[4]. This threshold is applied to all grid points, and can be varying with the calendar month based on three-month centered distributions. To ensure the quasi-stationarity and persistence, the blocking event must persist for at least five days, with a minimum spatial overlap of 50% between the blocked areas on consecutive days. Note that any two blocking episodes that are separated by only one non-blocked day are counted as one blocking event, and any blocking events with less than 5 days are ignored. Here, we focus on the blocking events in winter (December–February, DJF) over the Euro-Atlantic sector.

## Finite-amplitude local wave activity budget

Following the formalism of Huang and Nakamura[51], the local finite-amplitude wave activity (LWA) is defined as the line integral of the eddy component of quasi-geostrophic potential vorticity $q_e$ along the meridional displacement of contour $q$ from its equivalent latitude $\phi$ as:

$$A(\lambda, \phi, z, t) = - a \int_0^\eta q_e \cos(\phi + \phi')\, d\phi', \quad \phi' \in [0, \eta], \quad (2)$$

where

$$q(\lambda, \phi, z, t) = f + \frac{1}{a \cos\phi} \frac{\partial v}{\partial \lambda} - \frac{1}{a \cos\phi} \frac{\partial (u \cos\phi)}{\partial \phi} + f e^{z/H} \frac{\partial}{\partial z}\left( \frac{e^{-z/H}(\theta - \tilde\theta)}{\partial \tilde\theta / \partial z} \right) \quad (3)$$

is quasi-geostrophic potential vorticity, and

$$q_e(\lambda, \phi, \phi', z, t) = q(\lambda, \phi + \phi', z, t) - Q_{REF}(\phi, z, t), \quad (4)$$

is the eddy component of the $q$ contour relative to a zonally symmetric reference state at the latitude where $q = Q_{REF}$[51]. $\lambda$ is longitude, $\phi$ is latitude, and $z = - H \ln(p/p_0)$, with $p_0 = 1000$ hPa and $H$ is the reference scale height (7 km), $\theta$ is potential temperature, and $u$ and $v$ are the zonal and meridional component of velocity field, respectively. $\tilde\theta$ is the area-mean of potential temperature over the hemisphere, and $f$ is the Coriolis parameter. The $Q_{REF}$ is calculated by zonalizing the wavy potential vorticity contour on the $z$-surface with an area-preserving map[5]. The integral bounds denote a domain of integration from the given latitude $\phi' = 0$ to the meridional displacement $\phi' \in [0, \eta]$. By definition, the local wave activity $A$ quantifies the amplitude of finite-amplitude wave disturbance in the atmosphere[51,66,67]. The density-weighted column mean of LWA budget reads as[5,48,51]:

$$\frac{\partial}{\partial t}\langle A \rangle \cos\phi = \underbrace{-\nabla_H \cdot \langle \mathbf{F} \rangle}_{I} + \underbrace{\frac{f \cos\phi}{H}\left( \frac{v_e \theta_e}{\partial \tilde\theta / \partial z} \right)_{z=0}}_{II} + \underbrace{\langle \dot{A} \rangle \cos\phi}_{III} \quad (5)$$

where,

$$\nabla_H \cdot \langle \mathbf{F} \rangle = \frac{1}{a \cos\phi} \frac{\partial}{\partial \lambda} \langle F_\lambda \rangle + \frac{1}{a \cos\phi} \frac{\partial}{\partial \phi}(\langle F_\phi \rangle \cos\phi), \quad (6)$$

$$\langle F_\lambda \rangle = \langle u_{REF} A \rangle \cos\phi - a \left\langle \int_0^\eta u_e q_e \cos(\phi + \phi')\, d\phi' \right\rangle + \frac{1}{2}\left\langle v_e^2 - u_e^2 - \frac{R}{H}\frac{e^{-kz/H}\theta_e^2}{\partial \tilde\theta / \partial z} \right\rangle \cos\phi, \quad (7)$$

$$\langle F_\phi \rangle = - \langle u_e v_e \rangle \cos\phi, \quad (8)$$

where $R$ is the gas constant for dry air (287 J K$^{-1}$ kg$^{-1}$). The angle bracket denotes the density-weighted vertical average of LWA $\langle (\cdot) \rangle \equiv \frac{1}{H}\int_0^\infty e^{-z/H}(\cdot)\, dz$, and

$$u_e(\lambda, \phi, \phi', z, t) = \frac{u(\lambda, \phi + \phi', z, t) \cos\phi}{\cos(\phi + \phi')} - u_{REF}(\phi, z, t), \quad (9)$$

$$v_e(\lambda, \phi, \phi', z, t) = v(\lambda, \phi + \phi', z, t), \quad (10)$$

$$\theta_e(\lambda, \phi, \phi', z, t) = \theta(\lambda, \phi + \phi', z, t) - \theta_{REF}(\phi, z, t). \quad (11)$$

The data is interpolated onto a regular pseudo-height ($z$) grid with $dz = 1$ km before the computation of vertical integral. The upper limit of the integral is taken to be $z = 48$ km (~ 1 hPa). Details of the calculation for reference states are discussed in previous studies[5,12,48,51,66].

Terms I in Eq. (5) represent the sum of the zonal and meridional convergences of the column-mean horizontal fluxes of wave activity. Term II is the upward LWA flux at the lower boundary (i.e., the vertical component of the generalized Eliassen-Palm flux). Term III represents diabatic sources and sinks of wave activity associated with non-conservative processes and is often evaluated as the residual of the budget. This term contains both the diabatic effect and the mixing effect due to wave dissipation. In this study, we calculate directly the diabatic wave sources/sinks of wave activity using the diabatic heating rates ($\dot\theta$) output from the model simulation as:

$$\Delta\Sigma_x = - a \left\langle \int_0^\eta f e^{z/H} \frac{\partial}{\partial z}\left( e^{-z/H} \frac{\dot\theta / c_p e^{\kappa z/H}}{\partial \tilde\theta / \partial z} \right) \cos(\phi + \phi')\, d\phi' \right\rangle, \quad (12)$$

so that the total diabatic sources/sinks of wave activity due to diabatic heating ($\Delta\Sigma$) can be decomposed as,

$$\Delta\Sigma = \Delta\Sigma_{LWCRE} + \Delta\Sigma_{SWCRE} + \Delta\Sigma_{LH} + \Delta\Sigma_{CLR}, \quad (13)$$

where LWCRE is longwave CRE, SWCRE is shortwave CRE, LH is latent heating, and CLR is clear-sky radiative heating. The longwave and shortwave cloud heating rates are defined as the difference between all-sky and clear-sky rates. In this framework, the blocked flow forms as a result of the saturation in the zonal distribution of LWA[5]. Thus, quantifying each term in Eq. (5) helps to elucidate which force drives changes in LWA and, thus, the formation of the block.

The computation of Eq.(1)-(12) is implemented with v2.2.0 of the Python package falwa[68,69], which computes finite-amplitude Rossby wave activity and fluxes with snapshots of weather variables as input. The class falwa.oopinterface.QGFieldNHN22, which solves reference states with the boundary conditions as in Neal et al.[12] (see Eqs. 14–16 in their SI), encapsulates the LWA and flux calculations presented in this study.

## Eady growth rate

To understand how rapidly baroclinic eddies can grow in a baroclinic environment, the maximum Eady growth rate (EGR)[70] is computed as:

$$\sigma = 0.31 g \, N^{-1} T^{-1} \left| \frac{\partial T}{\partial y} \right|, \tag{14}$$

Here, $\sigma$ is the maximum Eady growth rate, the Brunt-Väisälä frequency, denoted as $N = \sqrt{\frac{g}{\theta} \frac{\partial \theta}{\partial z}}$, and $g$ is the acceleration due to gravity. Higher Eady growth rates indicate faster growth of baroclinic eddies and, therefore, more seeding to the blocking formation.

## Diagnostics for quasi-stationary waves

The propagation of quasi-stationary planetary-wave activity is analyzed through the horizontal component of the Plumb flux[71] in a spherical coordinate system as follows:

$$F_S = p \cos(\phi) \begin{pmatrix} \frac{1}{2a^2 \cos^2(\phi)} \left[ \left( \frac{\partial \psi'}{\partial \lambda} \right)^2 - \psi' \frac{\partial^2 \psi'}{\partial \lambda^2} \right] \\ \frac{1}{2a^2 \cos^2(\phi)} \left[ \frac{\partial \psi'}{\partial \lambda} \frac{\partial \psi'}{\partial \phi} - \psi' \frac{\partial^2 \psi'}{\partial \lambda \partial \phi} \right] \end{pmatrix} \tag{15}$$

here, streamfunction $\psi = Z/2\Omega \sin \phi$, where $Z$ is geopotential, and $\Omega$ denotes Earth's rotation rate. The stationary wave in the streamfunction is calculated as $\psi = \overline{\psi} - [\overline{\psi}]$, where overbars and square brackets denote the monthly and zonal means, respectively.

## Data availability

MERRA-2 data are publicly available from NASA GES DISC (https://doi.org/10.5067/A7S6XP56VZWS; https://doi.org/10.5067/9NCR9DDDOPFI). The simulation output from this study is made available at https://portal.nersc.gov/archive/home/b/beharrop/www/e3sm_cre_denial_overview_data/Data_Overview_CRE_denial_in_E3SM.tar. The COOKIE2 data are publicly available via the Earth System Grid Federation, which can be accessed at https://esgf-node.llnl.gov/search/cmip6/. The raw data underlying the figures are available at https://doi.org/10.5281/zenodo.17138125.

## Code availability

The source code needed to run the E3SM experiments is available at https://github.com/beharrop/E3SM. Template bash scripts for setting up and running cloud-locking E3SM are included in the supplementary material of Harrop et al.[46]. The data in this study are analyzed with NCAR Command Language (NCL). To detect a blocking event, the Contour Tracking (ConTrack) algorithm is employed. The ConTrack is a Python package designed to simplify the process of automatically tracking and analyzing synoptic weather features in weather and climate datasets. The code is publicly available online at https://github.com/steidani/ConTrack. Code to compute finite-amplitude local wave activity, fluxes and non-conservative force contribution via calling falwa[68,69] is publicly available at https://github.com/csyhuang/hn2016_falwa/tree/master/scripts/lubis_et_al_2025. The scripts for analyses and generating figures are available at https://doi.org/10.5281/zenodo.17138125.

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

## Acknowledgements

This work is supported by U.S. Department of Energy Office of Science Biological and Environmental Research as part of Global and Regional Model Analysis program area. The Pacific Northwest National Laboratory (PNNL) is operated by Battelle for the U.S. Department of Energy under Contract DE-AC05-76RL01830. This study utilized computing resources at the National Energy Research Scientific Computing Center (NERSC), a U.S. Department of Energy Office of Science User Facility located at Lawrence Berkeley National Laboratory, operated under Contract No. DE-AC02-05CH11231. N.-E.O. was supported by the Impetus4Change (I4C; grant 101081555), NextGEM (grant 101057527), and Nor-ESM4CMIP7 (RCN grant 352204) projects, funded by the Research Council of Norway (RCN) under the European Union's Horizon Europe research and innovation programme.

## Author contributions

S.L. conceived the original idea, designed the research, and conducted the analysis. B.H. designed and ran CRE experiments. B.H., J.L., R.L., Z.C., C.H., and N.O. contributed to improving the analysis and interpretation. S.L. wrote the first draft, and all the authors edited the paper.

## Competing interests

The authors declare no competing interests.
