## [Transparent Peer Review file · Nature Communications]

Cloud-Radiative Effects Significantly Increase Wintertime Atmospheric Blocking in the Euro-Atlantic Sector

Corresponding Author: Dr Sandro Lubis

Version 0:

Reviewer comments:

Reviewer #1

(Remarks to the Author)

This study presents results that estimate the cloud-radiative effects (CREs) on the frequency of Euro-Atlantic blocking during boreal winter by using climate model simulations with a technique that isolates the interactive CRE. While it is increasingly accepted within the community that the role of upstream latent heating (or moist processes in general) accompanied by synoptic-scale disturbance is pivotal in triggering and maintaining atmospheric blocking systems, the authors stress that it has yet been largely unexplored how the CREs, specifically, affect the genesis of atmospheric blocking. This missing piece of the mechanism becomes a motivation for the study to further investigate the linkage between CREs and atmospheric blocking.

The authors aim to evaluate the quantitative contribution of CREs in the column-integrated local wave activity (LWA) budget diagnostics. This objective is addressed by comparing the cloud locking (CLOCK) experiment, where interactive CREs have been disabled, to the control simulation, and by explicitly computing the diabatic sources/sinks of LWA from the residual term. As a result, the authors argued that the CREs contribute to the increase of atmospheric blocking frequency by 1) directly increasing the diabatic wave source, as well as by 2) enhancing the amount of latent heat release through longwave CRE (LWCRE) and its feedback on surface evaporation and the subsequent moisture content. They also noted that 3) changes in mean CREs indirectly are capable of substantially modulating blocking frequency by changing the background mean states.

From my reading, the topic of this study – causally linking CREs to atmospheric blocking frequency – is quite novel. To my knowledge, no modeling study has used CLOCK experiments (or even a subset of CFMIP experiments) to investigate this detailed aspect. The key argument of the study that CREs increase the blocking frequency is clearly presented, and it is relevant for many fields of atmospheric sciences and weather community.

Nevertheless, regarding their physical interpretation of the results, I have some serious concerns. For instance, there is a distinct discrepancy between the location of LWCRE and condensational heating relative to the center of the block, which makes the suggested feedback of LWCRE on condensational heating questionable. I also found the argument that the CLOCK experiment does not significantly alter the mean state to be not very convincing. The credibility of the statement is further weakened by the barely visible white stippling that denotes statistical significance. In addition, even if the methodology of LWA budget analysis has been quite established now, estimating the diabatic process contribution from the residual of the budget entails some key assumptions and thus needs caution. Specifically, it does not provide any information on the vertical structure of CREs, which is obviously a very important component for wave source (not necessarily with respect to 'column-integrated' LWA diagnostics). Another concern that counteracts the strengths of the study is some of the conclusions drawn from the results sound too strong, despite some uncertainties in the proposed mechanism (e.g., Lines 272-273). With that said, I hope the authors address these concerns through major revisions.

Specific major comments

1) Figure 6, LWCRE within blocking: There has now been much evidence that upstream diabatic heating is critical for downstream blocking development, as seen from the references. A novel finding of this study is that longwave CRE (LWCRE) regulates the amount of condensational heating at the upstream of blocking center, based on the composited results in Fig. 6. Although not described clearly, the authors mentioned that such feedback by LWCRE may be realized

through surface evaporation (caption in Fig. 8; note that the description in Lines 274-278 is to explain the LWOFF response). However, the composited LWCRE heating anomaly is situated within atmospheric blocking, whereas the condensational heating anomaly occurs mostly upstream of the blocking center, or along the northwestern boundary of the geopotential height isolines. Given that these heating anomalies are not collocated, it seems unclear how LWCRE could influence the farther upstream latent heat release remotely through the suggested feedback. The fact that the mechanism of the suggested feedback itself remains uncertain (e.g., Lines 272-273) further complicates understanding this causal linkage. Can the authors add more supporting evidence?

2) Figure 6, physical meaning of positive LWCRE: It is quite surprising (and interesting) for me that the contribution of long-wave cloud-radiative effect is positive, not negative, despite that low-level clouds are dominant in the midlatitudes. I wonder if this physically makes sense and provides an insight. As noted in the introduction, the warm conveyor belt airstreams that trigger this life cycle of blocking occur mostly at mid-to-low troposphere, and there is radiative cooling from low-level clouds tops and heating at cloud bottoms (e.g., Cesana et al. 2019 and many others). Does it mean that radiative warming at the cloud bottom (peaks at the surface) outweighs its atmospheric cooling effect and acts as a positive diabatic source of wave activity? Providing a physical explanation of positive LWCRE in the discussion of Fig. 6 would help readers understand its importance more comprehensively.

3) Lines 472-484, Methodology-the validity of estimating radiative heating contribution of wave activity from equation (12): Related to the comment above, I am a bit confused of how the contribution of cloud radiative heating can be accurately estimated from the current methodology. If the results in Fig. 6a are taken at face values, it basically suggests that near-surface warming by upstream longwave radiative heating drives mid-tropospheric circulation anomaly. However, as we all know, change in atmospheric circulation is not simply driven by diabatic heating at near surface or certain height only. The vertical structure of radiative heating is complex and can sometimes induce counteracting responses in midlatitude circulation (e.g., Voigt et al. 2023). From my understanding, because of the height scaling factor in Eq. (12), it puts more (perhaps too much) weight to the near surface radiative heating values by construction, and limitedly reflects the rich vertical structure of cloud-radiative effects. I think this is conceivable a potential caveat of the residual approach of the LWA diagnostics. Can the authors show that this estimated LWCRE from Eq. (12) well represents the LWCRE in a more general condition (not particularly associated with blocking onset)? Otherwise, adding a discussion on some methodological limitations may provide further insights into the study.

4) Lines 472-484, Methodology-implicit assumption on model representation of longwave heating: This study calculated the diabatic sources/sinks of wave activity using the simulated diabatic heating rates, which is different from previous studies (e.g., Neal et al. 2022) who estimated the total diabatic heating effects by suppressing positive diabatic forcing term in the LWA budget integration. While this explicit calculation thus adds a uniqueness to the study, it also hinges on an implicit assumption that the model simulates the diabatic heating rates in a reasonable quality, which is not shown or mentioned at all in the manuscript. To put it another way, it seems important to ascertain whether the control experiment well simulates the horizontal and vertical structure of climatological diabatic heating and does not exhibit obvious errors.

5) Figure 7, significance of changes in mean states: It is repeatedly emphasized in the manuscript that the CLOCK experiment does not significantly change the mean states at all (particularly in the conclusion section), and thus any notable difference between CLOCK and control simulations is attributable to interactive CREs. However, I am not sure if this is entirely true, considering that there are still some hints of poleward displacement of the North Atlantic jet, weakening of the southwest-northeast tilt of stationary eddies, and reduced maximum Eady growth rate over the Gulf Stream and the entrance of the North Atlantic storm track. I feel these features will be more pronounced if colored stippling is used for statistical significance. Can the authors quantitatively show these mean state changes in CLOCK are negligible compared to LWOFF? In any case, the authors need to tone down statements that overstate the absence of significant change in the mean state for CLOCK throughout the manuscript, as there are actually some significant changes.

6) Figures 5, 6, 7, differences in moisture transport: Changes in the climatological mean states from the CLOCK experiment are seemingly small (Fig. 7), but it does not necessarily mean that such changes are trivial in modulating atmospheric blocking frequency. One possible pathway of their impacts is a reduction of poleward moisture transport over the upstream of Euro-Atlantic blocking. If the climatological moisture transport is reduced in CLOCK, compared to the control run, it indicates that the moist processes driven by stationary eddies are weaker and therefore contribute to the reduction of condensational heating (Fig. 6a), in addition to the contribution from interactive CREs.

7) Figure 8, schematic diagram and figure caption: The diagram is reminiscent of Figure 1 in Steinfeld et al. (2022) and seems to build upon previous findings. However, I am somewhat skeptical about whether this schematic illustration accurately reflects the results presented in this study.

The first step, 'moist air injection', is not explicitly shown in this study but used without any reference being provided in the caption. Regarding the third step, as noted in the earlier major comments, LWCRE is not located upstream of the blocking center, but rather within it. This differs from illustration. Moreover, the process described in the figure caption (i.e., 'The enhanced LWCRE in turn amplifies LH by boosting evaporation') is not clearly demonstrated in this study, resulting in an unsubstantiated statement. The term 'divergent outflow' at the upper troposphere also abruptly appears without any context. Lastly, it feels a bit strange to see the vertical structure of the proposed mechanism for the first time in this study as a schematic diagram. I recommend the authors either replace it with a new diagram based on the results of the study within the framework of LWA diagnostics, or provide a list describing the chain of processes in order.

Minor comments

1. Line 36: 'there is no comprehensive dynamical theory'
2. Lines 35-36: 'simulating and predicting blocking events ... are rare and localized'. This sentence could flow better. I recommend rephrasing it.
3. Line 10, 39: 'notoriously underestimate'. I understand that the authors want to strongly motivate the study by pointing out the systematic model biases, but it sounds too strong. Also, in terms of the anomaly method for blocking detection, note that some regions show overestimating biases (Woollings et al. 2018). I recommend an expression like 'mostly underestimate'.
4. Line 46: 'SST'. This acronym is not defined beforehand.
5. Line 53: ref. 34. Information of ref. 34 is missing in the reference.
6. Line 101: 'climatology in winter (December to January)'. There seems a mismatch between this expression and the figure caption. I guess '(December to February)'?
7. Figure 1 caption: It would be helpful to expand all acronyms at their first usage in captions.
8. Hatches in Fig. 2a: What does this black hatch mean?
9. Line 242: 'stronger southerly flow ($V_e < 0$)'. Is this sign error? I think $V_e > 0$ represents southerly flow, not $V_e < 0$.
10. Line 244: ') in the upstream' -> 'in the upstream'
11. Lines 275-276: 'Reduction in latent heat release ... evaporation'. Alternatively, this could be simply due to a weaker warm conveyor belt airstream and the associated moisture transport. See major comment 6.
12. Line 300: 'is minimal' -> 'are minimal'.
13. Lines 308-311: This sentence states that there is a 'partial weakening' of the climatological ridge over Europe, but then it is followed by an expression 'Despite no change in the mean state'. Based on the results in Fig. 7, further investigation seems necessary. See major comment 5.
14. Line 321: 'the observed reduction ... over the region'. It sounds like the authors are describing observational results. I recommend rephrasing it.
15. Figure 7 caption: Is the unit of the streamfunction correct? It seems a factor of 10 is missing.
16. Color of stippling: I strongly recommend changing the color of the stippling. It is very difficult to identify.
17. Lines 377-378: 'the distinctive role of diabatic wave activity injection ... anticyclone'. I feel this sentence could flow better, and its structure also looks somewhat too similar to the original sentence in Neal et al. I recommend rephrasing it.
18. Line 442: 'a blocking-based anomalies index'. I think the authors meant 'an anomaly-based blocking index'.
19. Line 480, Equation 13: Is total diabatic heating composed of only these four terms? I thought there is also a heating term of vertical diffusion (or sometimes referred to as diabatic heating due to turbulent mixing). Also, I guess longwave CRE (LWCRE) is derived from the difference between net longwave heating and clear-sky radiative heating, correct (same for SWCRE)? If so, please add this detail.

Reference

- Cesana, G., Waliser, D. E., Henderson, D., L'Ecuyer, T. S., Jiang, X., & Li, J. L. F. (2019). The Vertical Structure of Radiative Heating Rates: A Multimodel Evaluation Using A-Train Satellite Observations. *Journal of Climate*, 32(5), 1573-1590, <https://doi.org/10.1175/jcli-d-17-0136.1>
- Neal, E., Huang, C. S. Y., & Nakamura, N. (2022). The 2021 Pacific Northwest Heat Wave and Associated Blocking: Meteorology and the Role of an Upstream Cyclone as a Diabatic Source of Wave Activity. *Geophysical Research Letters*, 49(8), <https://doi.org/10.1029/2021gl097699>
- Steinfeld, D., Sprenger, M., Beyerle, U., & Pfahl, S. (2022). Response of moist and dry processes in atmospheric blocking to climate change. *Environmental Research Letters*, 17(8), <https://doi.org/10.1088/1748-9326/ac81af>
- Voigt, A., Keshtgar, B., & Butz, K. (2023). Tug-Of-War on Idealized Midlatitude Cyclones Between Radiative Heating From Low-Level and High-Level Clouds. *Geophysical Research Letters*, 50(14), <https://doi.org/10.1029/2023gl103188>
- Woollings, T., Barriopedro, D., Methven, J., Son, S. W., Martius, O., Harvey, B., Sillmann, J., Lupo, A. R., & Seneviratne, S. (2018). Blocking and its Response to Climate Change. *Curr Clim Change Rep*, 4(3), 287-300, <https://doi.org/10.1007/s40641-018-0108-z>

Reviewer #2

(Remarks to the Author)

Cloud-Radiative Effects Significantly Increase Wintertime Atmospheric Blocking in the Euro-Atlantic Sector (Lubis et al., 2024)

This study investigates the radiative impact of clouds on the dynamics and frequency of midlatitude atmospheric blocking during winter. Using two modeling techniques, the authors assess the impact of cloud radiative effects (CREs) on the blocking and show that CREs significantly increase the frequency of blocking over the Euro-Atlantic sector. To gain a process-based understanding of the radiative impact of clouds on the blocking dynamics, a budget analysis of the local wave activity was performed for different simulations. The results indicate that CREs directly affect the onset of blocking by enhancing the diabatic source of wave activity upstream of the blocks. Furthermore, the results indicate that CREs can also indirectly affect the dynamics of blocking through changes in the mean atmospheric state, such as changes in the position of the jet stream and the eddy growth rate.

Overall, this is a well-structured study that highlights the importance of cloud-radiation-circulation coupling for the dynamics of atmospheric blockings that are associated with high impact weather extremes. Over the last decade, the importance of cloud-radiative effects on midlatitude dynamics, either on climate or weather time scales, has gained attention in climate studies, and with it the need to improve the simulation of CREs in models. A better understanding of blocking dynamics and a better simulation of CREs in models are both important aspects of modeling, and therefore I find this study interesting and useful as it provides new insights into midlatitude dynamics.

The methods used in the study are well established and documented in previous studies. The authors used both well-known cloud locking and COOKIE methods to assess the impact of CREs on blocking. The results were also validated by comparing the impact of CREs from other model simulations, and overall, the authors find a robust result regardless of the model used. In addition, the suitability of the diagnostics used was assessed using reanalysis data. Overall, I consider the study suitable for publication, although I have some remarks, questions and minor revisions.

Remarks:

1- As mentioned in the preprint, many factors contribute to the formation of blocking. I am interested to know how the impact of CREs compares to the other factors. If possible, some information in the discussion can also help to put the impact of CREs into perspective. Also, since CRE uncertainties are smaller than the CRE itself, can we expect to see a significant impact of CRE uncertainties on blocking frequency? For example, the study of Zhao et al., 2018 (<https://www.sciencedirect.com/science/article/pii/S0169809517309122>) shows that uncertainties in CREs could affect the mean circulation.

This also makes me wonder about the time scale of the impact of CREs. It may be the case that uncertainties in the simulation of CREs affect the blocking frequency on longer climate time scales of years, but do we expect uncertainties in CREs to affect blocking events on time scales of weeks or during the blocking onset period? (This relates to the argument in lines 401-402: "minimizing CRE-related model biases, could therefore potentially enhance early-warning systems to mitigate damages caused by such events.")

Finally, can the authors elaborate on which method is better to quantify the impact of CRE on blocking for medium-range weather forecasts?

2- Considering that SST and sea ice are fixed in the simulations performed in this study (also preserving the mean state in cloud locking), it seems better to use the term cloud-radiative heating (CRH) rather than cloud-radiative effects (CRE)? CRE is more general and includes the impact of clouds on surface radiative effects. The term Atmospheric Cloud Radiative Effect (ACRE, (K/s)) has also been used before (e.g., Voigt et al., 2020, <https://wires.onlinelibrary.wiley.com/doi/full/10.1002/wcc.694>), which separates the radiative effect of clouds in the atmosphere from that of the surface. Or perhaps it should be mentioned in the paper that the radiative heating of clouds in the atmosphere is actually being studied? Following on from this, I was wondering if the results might change if the interactive ocean was coupled to the model simulations rather than the SST being prescribed. Can the surface CRE also affect the blocking?

3- Are all temperature tendencies from the physical parameterization considered in the analysis of the diabatic source of LWA? In Fig. S6 the "Res" is attributed to dissipation processes, but in Eq. 13 only the heating rates from radiation and condensation are considered. What about heating from turbulence or convection? I am also confused by the changing notion of latent heating and condensation throughout the text. In your analysis, does condensation heating (COND) include heating from all microphysical processes and heating from convection? If so, I suggest calling this just latent heating.

4- Regarding the repeated argument of LWCRE feedback on latent heating by enhanced evaporation. Can the authors elaborate on this? The preprint refers to studies that actually show that CRE warming in the tropics is balanced by less latent heating.

There are studies that indicate the mechanism of the impact of longwave radiation on clouds and latent heating. For example, Fu et al., 1995 ([https://doi.org/10.1175/1520-0469\(1995\)052<1310:IORACI<2.0.CO;2](https://doi.org/10.1175/1520-0469(1995)052<1310:IORACI<2.0.CO;2)), showed that LW radiation

enhance condensation and droplet growth. Or by affecting the eddies that can promote stronger vertical motion and hence latent heating (Li et al., 2015).

Minor revisions:

Line 15: ... (CREs significantly increase the formation of) - "CREs significantly enhance the formation of blocking or increase the frequency of blocking"

Line 46: ... sea surface temperature (SST)

Lines 43-44: ... (models mean state) - "model representation of mean atmospheric state and/or the waves at ..."

Line 53: reference 34 is missing.

Line 54: ... (blocking dynamics were mostly presumed to be predominantly dry) - "the dynamics of atmospheric blocking is predominantly driven by dry dynamics"

Line 68: ... (resulting from the presence of clouds) - "resulting from cloud-radiation interaction (absorb, emission, ...)"

Line 74: ... (over the extratropical ocean across models), I think reference 41 shows the impact of CRE on tropical ocean rather than extratropical ocean, reference 37 talks about the impact of CRE on extratropical ocean

Line 77: reference 42 do not directly show the impact of CRE on forecast error growth, rather the impact of all-sky radiation on error growth.

Line 83: ... (greater clarity into), greater than? I believe compared to COOKIE methods? In this section I believe there is a need to shortly explain the difference between the two methods to help smoothly follow the story (even though it is explained later and in the method).

Line 84: ... "used cloud locking to study the impact of CRE on"
Also, in reference 39 the cloud locking method is not used rather than just disabling CRH and applying COOKIE method.

Line 87: ... (to longwave radiation), I understand that the impact of shortwave CRE is negligible but would be good to mention it.

Line 90: might be helpful to shortly mention here or in the discussion why CRE is biased in weather and climate models.

Figure 1a: can the contour labels be added to ease the comparison.

Figure S1: caption: shading (tendency) unit is m day^{-1}

Line 118: The reference to Fig. 1d with the result of 21% decrease in blocking frequency seemed at first confusing.

Line 126: "Clouds On/Off Climate Intercomparison Experiment (COOKIE)"

Line 129, as only the cloud radiative heating is disabled in these simulations, I suggest the term clouds-off LW or CLWOFF or something similar instead of LWOFF

Line 142: also, please refer to Table 1 here

Figure 2: caption: remove ((b) As in (a))?

Line 160: density-weighted vertical average, what is the interval of the vertical integral in this analysis?

Line 162: remove (RHS), only used once

Line 169: Is A "wave activity" or local wave activity (LWA)?

Figure 3:

- (a-d) do these panels show the left-hand side term of Eq. 1 or sum of right-hand side terms?
- Caption: meridional eddy heat flux at the base of the atmosphere or at the lower boundary? And which level is that?
- Caption: ((m-p) sources and sinks of wave activity) - "(m-p) nonadiabatic sources and sinks of the wave activity"

Figure S3: ... blocking -in- the CTL ..., also change Lag to days, consistent with Fig 3

Line 209: OLR change to "outgoing longwave radiation", here would be good to also mention that minimum OLR and max water are associated with the position of WCB.

Line 212-220: nonadvective or nonadiabatic?

Figure 4: is the "blocking high center" refer to the position of maximum geopotential anomaly?

Line 274: ... (eliminates the LWCRE to LWA) - "eliminates the impact of LWCRE on LWA"

Line 282: ... from LWCRE and COND, (introduced before)

Figure 6: caption: ... Euro-Atlantic blocking ((remove shading), unit: $m s^{-1} day^{-1}$), In (remove all) panels (b-g)

Lines 288-289: please refer to Fig. 6a and Fig. S6

Figure 7 caption: I suggest rewriting the caption, as using phrases such as 'as in (a-b)' does not imply what contours are shown in other panels.

Line 310: From Figure 7 I tend to say that the mean state is in fact changing in the CLOCK?

Line 358: ... where clouds are fully transparent to longwave radiation.

Table 1: Clouds On/Off Klimate Intercomparison Experiment

Eq. 3: maybe better to bring equation 4 before 3 as "q" is not in Eq.2

Line 462: Qref change to QREF, consistent with other terms

Line 470: revise to "The density-weighted vertical average of LWA ("column LWA") tendency reads as"

Line 475: Non-advective or nonadiabatic?

Ref 44: There is newer reference to Harrop et al., 2024 (<https://gmd.copernicus.org/articles/17/3111/2024/>):

(Harrop, B. E., Lu, J., Leung, L. R., Lau, W. K. M., Kim, K.-M., Medeiros, B., Soden, B. J., Vecchi, G. A., Zhang, B., and Singh, B.: An overview of cloud–radiation denial experiments for the Energy Exascale Earth System Model version 1, *Geosci. Model Dev.*, 17, 3111–3135, <https://doi.org/10.5194/gmd-17-3111-2024>, 2024.)

Reviewer #3

(Remarks to the Author)

Review comments on Lubis et al.

In this study, the authors present compelling evidence on the critical role of the cloud radiative effects (CREs) in enhancing blocking frequency. By using a suite of idealized simulations and applying local wave activity diagnostic, they show that CREs amplify blocking frequency by strengthening upstream wave activity. Investigating the impact of CREs on blocking dynamics and frequency is novel, and their idealized simulations offer convincing support for the significant influence of CREs on wintertime blocking frequency, employing a solid dynamical framework.

However, the manuscript, in its current form, requires substantial refinement. I have several questions and concerns regarding the methodology of their idealized simulation, as well as the interpretation of their results. Particularly, the experimental setup contains inconsistencies that hinder reproducibility, and the interpretation of results lacks sufficient explanations of underlying physical mechanisms.

My detailed comments are as follows.

Major Comments:

1. Methodology:

From my understanding, all the idealized simulations run in this work use the same SSTs and SICs computed from a 20-year monthly climatology from 1990 to 2010 (see my second minor comment below). With this choice of the boundary condition, I am concerned that the authors may be underestimating moisture sources from the ocean to the atmosphere, particularly over the Gulf Stream and North Atlantic Current. By averaging out mesoscale ocean eddies, which are prevalent in these regions, the simulations may suppress surface latent heat flux, potentially reducing moisture availability to the atmosphere. Previous studies (e.g., Yamamoto et al. (2021, WCD), Matthews et al. (2024, GRL)) have shown these regions to be critical moisture sources for latent heating associated with Euro-Atlantic blocking events. I recommend that the authors consider using non-averaged boundary conditions. If that is not feasible, please provide a clear justification for the choice of this SST/SIC boundary condition and acknowledge this potential limitation of this work in the main text.

For CLOCK experiment, why do authors prescribe the cloud optical properties only from the first three years of the CTL simulation and not the entire 20-year period of the CTL simulation? The authors describe how in this manner they can ensure the similar CRE climatology to CTL, but I am not convinced. Furthermore, the authors write how they loop over the three-year cloud optical property files for the eleven-year cloud locking experiment, which sounds inconsistent with the description that the experiment is run for 20 years. Please rewrite the paragraph to clarify the above points if it is my mere misunderstanding. If not, I recommend that the authors undertake the simulation with the use of the CRE file with the entire 20-year period of the CTL simulation to truly ensure that both simulations have the same CRE climatology.

The usage of 1979-2020 ERA5 data is not quite consistent with the experimental design where the authors chose to use 20-year integration based on the 20-year climatological SSTs and SICs. In order to make a fair comparison with the model results, please use 1990-2010. Also, please add the results from ERA5 on Figs 4 and 5 as well.

The decomposition of the total diabatic sources/sinks of wave activity is a great attempt, and I found the results quite interesting. I recommend that the authors use JRA55 reanalysis data, which I believe provides the necessary diabatic terms, to compare and validate their results.

2. Interpretation of the physical significance of CRE

From the model results that are presented in this manuscript, I agree with the authors that CRE does seem to significantly -- although secondarily to latent heating -- contribute to increase the wintertime Euro-Atlantic blocking frequency. However, the specific role of CRE in atmospheric blocking remain unclear. In the introduction, the authors review prior studies on the impact of CRE on large-scale atmospheric circulation, yet they do not discuss the physical processes directly relevant to blocking. Since CRE can induce both cooling and heating effects depending on cloud type, and blocking involves high-pressure systems (clear skies) alongside warm conveyor belts (precipitating clouds), the authors should clarify their hypothesis on how CRE may influence blocking. A more detailed physical explanation would strengthen their argument, especially regarding Figures S5 and 6(b).

Minor Comments:

1. The authors write that for CTL they use initial condition of the year 2000s, but which exact years do they use precisely? Or do they mean they used the atmospheric condition of year 2000?
2. In l.127 the authors write that they use SST from year 2000, while in l.425 they describe that they use the same SSTs and SICs as CTL. Please amend this inconsistency accordingly.
3. Detailed description of the methodology of the COOKIE-2 simulations is completely missing, and the authors need describe it in detail in the Method section. Also, I found it very confusing that the authors use COOKIE-2 and CFMIP interchangeably.
4. How many blocks are detected in each experiment, and thus included in each composite maps?
5. The authors attribute the LWA sources almost solely to the diabatic heating term, which is not in line with the previous studies (e.g., Pfahl et al. 2015, Steinfeld and Pfahl 2019, and Yamamoto et al. 2021 all found ~50% associated with diabatic heating, and the rest adiabatic). How can you explain this discrepancy?
6. I suggest that the authors swap around Sections 3.1 and 3.2, to first show the mean states of the idealized simulations before discussing LWA results.
7. I would appreciate it if the authors can list limitations of the current study, including the point that I raised in my first comment, in the conclusion.
8. Fig 1b: interesting tail of negative signature extending all the way to off the east coast of North America. If you extend the area to a broader area, would it extend to the proximity of Gulf Stream? (related to my first major comment)

Other small comments:

- L80 "climate feedback modes" -> "climate modes"?
- Fig S1: Is this a case study, or composite?
- L.130: Which major atmospheric heat source are the authors speaking about? Please be more specific.
- L.154-156: Again, please elaborate more, on what bases the authors come up with these two hypotheses, and what they exactly mean by them.
- Fig 2 caption: Please properly describe what the box-whisker plots in (b) and (c) show. Are they showing interannual variability, or ensemble spread? Please explain. They are clearly different from (a), so please remove "(b) As in (a)". Also, the circles over the box-whisker plots are misleading, because they are normally used to indicate the presence of anomalies.
- L.174 "closely resembles": I would lessen the tone, as there are visible differences, including that there is no clear low-pressure system on the west of blocking in reanalysis.

- L.176 “stronger deceleration of the zonal flow”: Is this shown anywhere?
- Fig S3: It reads lag-3 – Lag0 but are they also supposed to read “days -3” – “days 0”?
- L. 192 “congestion of a traffic jam”: Tautology. Please rewrite.
- Fig S4: Please change the contours of all the panels to geopotential height anomalies, so it is easier to compare against Fig S2, to see what the authors are trying to show the readers in l.210.
- Fig S5: “DTCOND” does not match with the description in the caption. Please make them consistent.
- L.222 “the interactive CRE (CLOCK) and changes in the mean LWCRE (LWOFF)”: The descriptions of what you want to illustrate and the corresponding simulation designs are not quite matching, or more like the opposite. Please rewrite.
- L.224: its budget “of all three simulations”
- Figs 4, 5: Stippling is not quite visible. I suggest that the authors change the color range to something like [-5 to 5], [-20, 20] respectively.
- Fig 4: Please indicate the 10deg x 10deg area used to construct (d) on each map.
- Figs 4 & 5: please adequately explain what exactly the error bars indicate.
- Fig 5: Are the encircled regions also 10deg x 10deg?
- Fig S6: “ $\Delta\Sigma_{tot}$ ” does not match with the description in the caption.
- L.271 “cloud-radiative impact might operate”: please elaborate more on what you mean by this.
- L.274: I don’t understand the sentence. Is it “from LWA”? Are you speaking about LWOFF simulation? Please clarify.
- L.276 “increased nighttime cooling”: Do you mean radiative cooling of the surface?
- L.279 “discused” -> “discussed”
- L.284 “larger longwave radiative heating”: By which physical mechanisms? Please explain.
- L.310 “no change in the mean state”: Just above this sentence, the authors described that there are some differences in the mean state.
- L.312 “This is consistent with our previous analysis”: please specify which paper/results the authors are referring to.
- L.317 Please explain how exactly turning off LWCRE change the meridional temperature gradient
- Fig 7: Indicate what contours of (c)-(f) are, and also the contour intervals for all the panels
- L.323 “Eady’s growth rate” -> “Eady growth rate”
- L.442 “blocking-based anomalies index” -> “anomaly-based blocking index”
- L.463: the definition of q is already written
- L.465: H=7km is already written
- L.468: “the meridional displacement of the wavy potential vorticity contour relative to ϕ ”
- L.485: reference for falwa is missing?
- L.488 “boundary conditions listed in Neal et al.”: please specify.
- L.489 “Eddy Growth Rate” -> Eady Growth Rate

References:

- Yamamoto, A., Nonaka, M., Martineau, P., Yamazaki, A., Kwon, Y.-O., Nakamura, H., and Taguchi, B.: Oceanic moisture sources contributing to wintertime Euro-Atlantic blocking, *Weather Clim. Dynam.*, 2, 819–840, <https://doi.org/10.5194/wcd-2-819-2021>, 2021.
- Mathews, J. P., Czaja, A., Vitart, F., & Roberts, C.: Gulf Stream moisture fluxes impact atmospheric blocks throughout the Northern Hemisphere. *Geophysical Research Letters*, 51, e2024GL108826. <https://doi.org/10.1029/2024GL108826>, 2024.

Version 1:

Reviewer comments:

Reviewer #1

(Remarks to the Author)

I appreciate the authors’ effort in addressing my previous comments and revising the manuscript accordingly, which I believe has substantially improved the manuscript. In their revision, the authors now propose two hypotheses to causally link CREs to dynamical processes of blocking formation: 1) increasing diabatic heating sources and 2) modifying the background mean state (L155-163). To test these hypotheses, partly in response to the reviewers’ previous comments, they have included new results on the vertical structure of diabatic heating, cloud-related variables, and moisture transport. I find all of them relevant and supportive of their proposed mechanisms. The authors also newly streamlined these mechanisms at the end of the section (L308-330), which adds clarity to understanding how interactive CREs influence blocking frequency. While I enjoyed reading the authors’ detailed response to my previous comments, I have one follow-up comment, which is simply to share my thoughts, along with a minor suggestion on the updated results. I have listed a few other minor suggestions for clarifications.

- Lines 308-330: The revised manuscript now suggests two (at least) possible pathways based on their original and new findings. They are clearly described and sound convincing for the most part. In particular, I find that the first mechanism provides a clear explanation dynamically linking LWCRE to changes in LH. Meanwhile, I would also like to point out that the possibility of the second mechanism, related to high cloud cover and evaporation feedback, has been explored only by MERRA2, not by model experiments. I assume the reason model results are not shown for the same diagnostics in the study is either that the variables relevant to the diagnostics are not available from the current model configuration, or that model results may differ substantially from the MERRA2 results, given that cloud fraction and surface properties are quite challenging to represent accurately in models. In any case, it is less clear whether

the second mechanism similarly operates in the CTL simulation.

The results from Figs. 7e and 7f also raise the possibility that the second mechanism may not be well captured by the model. Specifically, the positive LWCRE anomaly in MERRA2 is located vertically from 800 hPa to 400 hPa with its peak at 500 hPa, whereas that in CTL is found between 850-550hPa (L304) with its peak at ~800 hPa. This vertical difference may indicate that cloud cover fraction at different altitudes in the model deviates from MERRA2 quite noticeably. If this surface warming through high cloud cover anomaly is not explicitly shown by model experiments, then I believe the suggested causality from MERRA2 is not fully demonstrated as well (as surface warming itself can be induced by many other factors). With that said, I feel this would be a minor point relative to the key arguments of the study. The authors have already provided sufficient new and interesting results, and I do not think it is essential for this study to ascertain whether the proposed second mechanism holds in model experiments. Nevertheless, it would be appreciated if the authors could clarify that the second 'possible' pathway proposed in this study is based on reanalysis results, and that further investigation in a separate modeling study would be helpful.

- Magnitude of vectors: Information on the magnitude of vectors is missing throughout the manuscript. I recommend including reference vectors in the figures or providing their magnitude information in the figure captions.

- Lines 156-157 'changes in the large-scale mean state, ..., the strength of stationary waves, influence'-> 'and the strength of stationary waves'? I recommend revisiting this sentence.

- Lines 445-446 'Therefore, improving ... could therefore potentially ...'. I suggest removing one of 'therefore' in the sentence.

- Vectors in the second row of Fig. 8: I found the black vectors denoting the Plumb flux appear somewhat smaller compared to the vectors in other figures. Please consider thickening them or changing their color.

Reviewer #2

(Remarks to the Author)

Cloud-Radiative Effects Significantly Increase Wintertime Atmospheric Blocking in the Euro-Atlantic Sector (Lubis et al., 2025)

This is my second review of the study. I appreciate the authors' efforts in addressing the previous comments, and the manuscript has really improved following the major revisions. I have only a few minor comments. Addressing these comments, I believe the paper is in good shape to proceed.

<Minor comments>

Lines 18-19 (also Lines 393-394): The sentence suggests that the main way in which CRE affects the diabatic source of wave activity is through direct impact ('primarily', CRE heating), while the indirect pathway through changes in latent heating is secondary. In fact, this is the opposite. It would be better to say, for example, that CRE affects the diabatic source of wave activity both directly and indirectly, with the indirect pathway playing a significant role.

Line 86: '... its impact on simulated variability,'

The term simulated variability here is unclear, -> changing to 'its impact on blocking formation'

Line 107: Here, it would be better to refer to the limitations of the experiments discussed later in the method, as Fig. S1 illustrates that notably fewer blocking events are simulated with the E3SM CTL. Is this related to the missing lower-boundary SST-related forcing in E3SM simulations?

Figures 5, 6, and S6: as the diabatic terms shown in Fig. 6a do not add up to the total diabatic term shown in Fig. 5d, it seems a large contribution comes from clear-sky radiative heating (according to equation 13), right? And if yes, this term does not change a lot between CTL, LWOFF and CLOCK? In other words, the major changes in diabatic source wave activity are due to changes in diabatic heating and specifically the latent heating?

Line 279: From Fig. 7 (cf. panels a and b), it seems that the limitation mentioned is due to differences in latent heating between MERRA and CTL at lower levels rather than to low-cloud radiative processes. At least, the spatial variability of cloud radiative heating is comparable between MERRA and CTL (panels e and f), but not latent heating. This makes me wonder about latent heating at lower levels by MERRA that is absent in CTL. What could be the reason?

Line 320: I believe that reference 41 fits better here, as it also shows that the absence of CRE cools the atmosphere and increases static stability within ascending regions of cyclones. However, it also highlights the significant role of LW cooling by low-level clouds in reducing static stability and weakening cyclones, similar to reference 38.

Line 328 References 37 and 54 show that CRE (warming) results in reduced latent heating (and precipitation) within the tropics. Reference 37 shows that CRE in the mid-latitudes increases precipitation (therefore LH), but this was attributed to enhanced eddies and vertical motion (similar to the first proposed pathway).

From the ref 37:

'the heating of the tropical atmosphere by ACRE (Fig. 4c) is primarily balanced by a reduction in the latent heating,

consistent with the reduction in tropical precipitation evident in the clouds-on simulation (Fig. 8).’

Therefore, referencing these studies does not seem correct. A similar reference might be 40, which shows that CRE enhances cloud microphysical heating, although it does not show changes in surface evaporation.

Lines 414-416: I don’t think that the identified and proposed mechanism of CRE impact is ‘fundamentally’ different. Results show that CRE (directly and indirectly) affect upstream diabatic heating (already later in line 418, the importance of upstream diabatic heating for blocking is mentioned), and changes in LH will translate into changes in PV at upper-levels. (like ref 40 that shows changes in LH by CRE lead to changes in large-scale winds and PV at upper levels and dominate the changes in the tropopause wave that, as also said in the preprint here, is important for blocking formation, e.g., ref 31). The results presented are additive and complementary to some established ideas. The point is that the indirect impact of CRE has a higher weight than the direct impact of CRE heating.

<Typos>

Line 480: Table S1 -> Table 1

Line 487: Table S2 -> Table 1

Line 496: cloud heating rates, all-sky rates -> cloud radiative heating, all-sky radiative heating, clear-sky radiative heating

Reviewer #3

(Remarks to the Author)

I appreciate the authors’ extensive additional analyses and their point-by-point response to my previous comments. I think the manuscript has improved since the previous round, particularly with the inclusion of diabatic decomposition and cloud cover diagnostics. However, I find the physical interpretation of the diagnosed effects still lacks dynamical depth and clarity, and important diagnostics that support the proposed mechanisms remain underutilized.

Below I outline my major concerns in detail:

Major Comments

1. Dynamical explanation

While the manuscript clearly establishes that CREs (especially LWCRE) enhance blocking by increasing upstream wave activity, I find the explanation of the physical mechanisms still remains largely descriptive. The text alludes to CRE-induced enhancements in latent heating and high cloud cover, but does not fully explain the dynamical pathway from cloud radiative forcing to blocking onset. The authors refer to Supplementary Figure S12 showing increased high cloud cover upstream of blocks. However, this key diagnostic is not meaningfully integrated into the main text or used to support the proposed feedback between LWCRE and latent heating. This weakens the causal argument. To improve clarity and support the proposed mechanism, I recommend:

- Bringing the high-cloud fraction results into the main text
- Interpreting results within the warm conveyor belt (WCB) framework, which provides a well-established basis for linking moist ascent, diabatic PV generation, and upper-level ridge formation (e.g., Grams et al., 2011; Methven, 2015)

In addition, the introduction (lines 70–82) would benefit from a more detailed discussion of prior work on how physically CREs influence large-scale dynamics, to better motivate the study and help interpret later figures, the former of which currently reads rather opportunistic.

Finally, the discussion around lines 297–307 would be significantly strengthened by explicitly incorporating:

1. PV-based reasoning — how diabatic heating generates negative PV anomalies aloft and strengthens ridges (Haynes and McIntyre, 1987)
2. Low-PV air transport via WCBs, which links latent heating with anticyclonic upper-level anomalies (e.g., Pfahl et al., 2015; Steinfeld and Pfahl, 2019)

Bringing these concepts into the discussion would ground the proposed mechanism in current dynamical theory and reinforce the CRE-blocking connection.

2. Mischaracterization of LWA sources vs. redistribution

The manuscript attributes blocking formation primarily to the convergence of horizontal LWA flux. However, this interpretation risks conflating LWA redistribution with LWA generation.

According to previous works (e.g., Nakamura and Huang, 2018; Yamamoto and Martineau, 2024), horizontal fluxes only redistribute LWA—they do not generate or destroy it. The true sources of wave activity are:

- The non-conservative term $(A) \cos\phi$, which includes diabatic heating, PV mixing, and numerical diffusion
- The low-level eddy heat flux, reflecting baroclinic generation.

Your Fig. 3 clearly shows both terms contributing positively upstream of the block, consistent with the findings of Yamamoto and Martineau (2024) for the Atlantic region. However, the manuscript text does not clearly distinguish these processes from flux convergence. I recommend reframing the interpretation to emphasize that CREs enhance wave activity generation via diabatic and baroclinic processes, which is then redistributed downstream by horizontal fluxes in lines 188–210.

3. Evaporation Feedback

In lines 327–331, the manuscript states that LWCRE-induced surface warming increases evaporation, which then amplifies latent heating. While plausible, I am afraid that this is oversimplified. According to the bulk formula, latent heat flux depends on the moisture gradient between surface and near-surface air, wind speed, and air density. Thus, an increase in surface temperature does not guarantee increased evaporation if near-surface air also becomes more humid. I suggest the authors

revise this explanation to clarify under what thermodynamic conditions the proposed feedback would hold.

Minor Comments:

- I have mentioned this for the first round, but I still think that the description of COOKIE/COOKIE2 are insufficient. For the sake of reproducibility, please clarify what they are in the main text.
- L.96 underscores -> underscore

References:

- Grams, C. M., Wernli, H., Böttcher, M., Campa, J., Corsmeier, U., Jones, S. C., Keller, J. H., Lenz, C. J., and Wiegand, L.: The key role of diabatic processes in modifying the upper-tropospheric wave guide: A North Atlantic case-study, *Q. J. Roy. Meteor. Soc.*, 137, 2174–2193, 2011.
- Methven, J.: Potential vorticity in warm conveyor belt outflow, *Q. J. Roy. Meteor. Soc.*, 141, 1065–1071, 2015.
- Haynes, P. H. and McIntyre, M. E.: On the evolution of vorticity and potential vorticity in the presence of diabatic heating and frictional or other forces, *J. Atmos. Sci.*, 44, 828-841, 1987.
- Pfahl, S., Schwierz, C., Grams, C. M., Wernli, H., Croci-Maspoli, M., Grams, C. M., and Wernli, H.: Importance of latent heat release in ascending air streams for atmospheric blocking, *Nat. Geosci.*, 8, 610–615, 2015.
- Steinfeld, D. and Pfahl, S.: The role of latent heating in atmospheric blocking dynamics: a global climatology, *Clim. Dynam.*, 53, 6159–6180, 2019
- Nakamura, N. and Huang, C.: Atmospheric blocking as a traffic jam in the jet stream, *Science*, 361, 42-47, 2018.
- Yamamoto, A. and Martineau, P.: On the driving factors of the future changes in the wintertime Northern-Hemisphere atmospheric waviness, *Geophys. Res. Lett.*, 51, e2024GL108793, 2024

Version 2:

Reviewer comments:

Reviewer #3

(Remarks to the Author)

I appreciate the authors' effort in expanding the analysis and responding to earlier comments. The manuscript is technically improved, but two major issues of physical interpretation remain unresolved.

This is the third time I raise this, and I will be more explicit here. I fully understand that this study uses the LWA framework to diagnose the impact of CREs on blocking. I am also aware of the formal dynamical basis of that framework, and its usefulness. My concern is not about methodology—but about the lack of physical explanation consistent with established dynamical understanding. The authors use LWA to quantify the contribution of diabatic processes to midlatitude wave activity and blocking. However, the manuscript never explains why those processes—particularly latent heating and LWCRE—lead to enhanced wave activity.

For example, to me Figure 7 clearly shows the signature of WCB-type ascent—vertical and poleward transport of low-PV air—which amplifies the upper-level ridge. This is a well-established mechanism, yet the manuscript avoids acknowledging it (l.277-288). Similarly, the authors attribute increased waviness to LWCRE but offer no physical mechanism—only a diagnostic association. If the claim is that CREs dynamically influence blocking, then the physical pathway must be explained in terms of PV-based thinking (e.g., via diabatic PV generation, modification of the jet, ridge amplification, etc.). It is not sufficient to show the signal in the LWA budget without interpreting the underlying dynamics. The authors already offered some dynamical explanations in the response to my previous comment, so I am merely asking to add these in the main text where relevant.

This omission is especially problematic given that the Introduction (l.75-84) frames the study as addressing the dynamical influence of LWCRE on the jet and storm track, but never follows through with a coherent explanation. To be absolutely clear, I am asking the authors to articulate the physical mechanism by which cloud radiative effects—particularly LWCRE—alter the atmospheric circulation, in terms of known dynamical processes such as diabatic PV generation, ascent in WCBs, or changes in jet structure. This includes explaining the role of cloud height and vertical heating profiles, since LWCRE is not a single uniform effect. Without this physical context, the claim that LWCRE enhances blocking lacks credibility, regardless of what appears in the LWA diagnostics.

The LWA formulation, after all, is based on QGPV dynamics—you cannot claim it's a different framework and therefore exclude PV-based interpretation. At this point, I find it disappointing that these core interpretative issues remain unaddressed.

Second, the description of LWA sources and redistribution still remains unclear and fragmented, although I see much improvement compared to the previous version. I understand the authors do not want to label zonal LWA convergence as a “sink,” but the current wording does not help the reader clearly understand what is happening dynamically. The mechanism should be presented cleanly: non-conservative forcings (e.g., latent heating and CREs) and low-level eddy meridional heat flux generate LWA upstream of blocking events, which is then redistributed by zonal LWA convergence, forming localized blocking structures. This sequence is physically grounded and fully consistent with Yamamoto and Martineau (2024), which the authors should cite. Right now, the manuscript reads more like a descriptive catalog of LWA differences than a coherent

dynamical explanation.

Version 3:

Reviewer comments:

Reviewer #3

(Remarks to the Author)

I appreciate the authors' effort to address my previous concerns. The manuscript now provides much greater clarity regarding the dynamical context of CREs and diabatic heating. I have a few additional minor comments regarding the newly added paragraphs.

I.211–I.216:

There is some redundancy in the discussion of Figs. 3m–p. You might consider simplifying the whole thing to something like:

“As shown in Figs. 3m–p, positive values of low-level sources and non-conservative (diabatic) processes are collocated with the emergence and growth of wave activity fluxes, acting as LWA sources, with non-conservative processes being of primary importance.”

I.375, 439, 451:

What do you mean by “quasi-stationary wave ridge”? Is this different from the “large-scale stationary ridges” mentioned at I.367? I found the expression confusing, since it is unclear whether you are referring to the climatological ridge or to a ridge associated with the blocking itself. Please clarify or rephrase.

I.377:

Did you mean “incipient” rather than “incident”?

I.446–457:

Here, the authors describe only the “indirect” diabatic impact (Hoskins et al. 1985; Steinfeld & Pfahl 2019). However, the “direct” diabatic impact — in which WCBs transport low-PV air from lower latitudes and altitudes into higher latitudes and the upper troposphere (e.g., Pfahl et al. 2015; Steinfeld & Pfahl 2019; Yamamoto et al. 2021) — should also be mentioned, as it likely plays a role.

We thank the editor and reviewers for their constructive comments and positive feedback. Before providing a point-by-point response to the reviewers' suggestions, we would like to highlight the major revisions made to the manuscript, as summarized below:

1. Additional supporting evidence has been incorporated, based on observation-based reanalysis data (MERRA-2) and model experiments, to strengthen our conclusion that cloud radiative effects (CREs) significantly increase the blocking frequency over the Euro-Atlantic sector:
 - By analyzing the vertical structure of cloud radiative heating, the diabatic wave source, cloud fraction, evaporation, and moisture transport during the lifecycle of blocking events, using both MERRA-2 reanalysis and model experiments, we confirm that interactive CREs enhance local wave activity (LWA) by strengthening the diabatic source of wave activity upstream of the blocking region. This intensification occurs primarily through the direct effect of longwave CREs and indirectly via their feedback on latent heating (LH). Although the direct contribution of longwave CREs to the diabatic wave source is secondary to their LH-indirect influence, both mechanisms contribute to the overall increase in the upstream diabatic wave source (see updated **Figs. 6 and 7**).
 - The increase in LH under LWCRE results from strengthened poleward moisture transport, and potentially to increased surface evaporation upstream of the Euro-Atlantic blocking region (see updated **Fig. 7 and Figs. S7–S12**).
 - In addition to the sizable indirect impact of CREs on blocking via LH, we emphasize that the indirect effects of CRE on the mean state, through the shift of the mean jet and weakening of the stationary waves, mean baroclinicity, and mean poleward moisture transport, influence the blocking formation (**updated Fig. 8**).
2. We have now compared the CTL results with reanalysis-based observations from MERRA-2 to validate our findings (see **updated Figs. 4-7**).
3. We have included additional details in the model experiment section, particularly the rationale for using a fixed SST lower boundary (see “*Numerical Experiments*”). A new discussion has also been added to address the limitations of the results in light of this choice (see “*Discussion*”).

We have also addressed all other reviewers' comments, and finally, we hope the revised version satisfactorily addresses your concerns.

Sincerely,
On behalf of all authors,
Sandro W. Lubis

Reviewer #1 (Comments to the Author):

This study presents results that estimate the cloud-radiative effects (CREs) on the frequency of Euro-Atlantic blocking during boreal winter by using climate model simulations with a technique that isolates the interactive CRE. While it is increasingly accepted within the community that the role of upstream latent heating (or moist processes in general) accompanied by synoptic-scale disturbance is pivotal in triggering and maintaining atmospheric blocking systems, the authors stress that it has yet been largely unexplored how the CREs, specifically, affect the genesis of atmospheric blocking. This missing piece of the mechanism becomes a motivation for the study to further investigate the linkage between CREs and atmospheric blocking. The authors aim to evaluate the quantitative contribution of CREs in the column-integrated local wave activity (LWA) budget diagnostics. This objective is addressed by comparing the cloud locking (CLOCK) experiment, where interactive CREs have been disabled, to the control simulation, and by explicitly computing the diabatic sources/sinks of LWA from the residual term. As a result, the authors argued that the CREs contribute to the increase of atmospheric blocking frequency by 1) directly increasing the diabatic wave source, as well as by 2) enhancing the amount of latent heat release through longwave CRE (LWCRE) and its feedback on surface evaporation and the subsequent moisture content. They also noted that 3) changes in mean CREs indirectly are capable of substantially modulating blocking frequency by changing the background mean states. From my reading, the topic of this study – causally linking CREs to atmospheric blocking frequency – is quite novel. To my knowledge, no modeling study has used CLOCK experiments (or even a subset of CFMIP experiments) to investigate this detailed aspect. The key argument of the study that CREs increase the blocking frequency is clearly presented, and it is relevant for many fields of atmospheric sciences and weather community.

Nevertheless, regarding their physical interpretation of the results, I have some serious concerns. For instance, there is a distinct discrepancy between the location of LWCRE and condensational heating relative to the center of the block, which makes the suggested feedback of LWCRE on condensational heating questionable. I also found the argument that the CLOCK experiment does not significantly alter the mean state to be not very convincing. The credibility of the statement is further weakened by the barely visible white stippling that denotes statistical significance. In addition, even if the methodology of LWA budget analysis has been quite established now, estimating the diabatic process contribution from the residual of the budget entails some key assumptions and thus needs caution. Specifically, it does not provide any information on the vertical structure of CREs, which is obviously a very important component for wave source (not necessarily with respect to ‘column-integrated’ LWA diagnostics). Another concern that counteracts the strengths of the study is some of the conclusions drawn from the results sound too strong, despite some uncertainties in the proposed mechanism (e.g., Lines 272-273). With that said, I hope the authors address these concerns through major revisions.

Authors:

We appreciate the reviewer's positive feedback and detailed, insightful comments on improving the manuscript. Below, we provide a point-by-point response to each comment, highlighted in blue.

Major Comments:

- 1) Figure 6, LWCRE within blocking: There has now been much evidence that upstream diabatic heating is critical for downstream blocking development, as seen from the references. A novel finding of this study is that longwave CRE (LWCRE) regulates the amount of condensational heating at the upstream of blocking center, based on the composited results in Fig. 6. Although not described clearly, the authors mentioned that such feedback by LWCRE may be realized through surface evaporation (caption in Fig. 8; note that the description in Lines 274-278 is to explain the LWOFF response). However, the composited LWCRE heating anomaly is situated within atmospheric blocking, whereas the condensational heating anomaly occurs mostly upstream of the blocking center, or along the northwestern boundary of the geopotential height isolines. Given that these heating anomalies are not collocated, it seems unclear how LWCRE could influence the farther upstream latent heat release remotely through the suggested feedback. The fact that the mechanism of the suggested feedback itself remains uncertain (e.g., Lines 272-273) further complicates understanding this causal linkage. Can the authors add more supporting evidence?

Authors:

We thank the reviewer for this thoughtful comment. To address the concern regarding the spatial relationship between LWCRE and latent heating (LH), and to clarify the mechanism underlying the proposed feedback, we have conducted a more comprehensive analysis of the three-dimensional structure and intensity of diabatic heating and the associated diabatic wave activity source, using both MERRA-2 reanalysis and model experiments. We now chose MERRA-2 over ERA5 because it provides comprehensive diabatic heating terms, including latent heating and cloud radiative heating components derived from satellite data assimilation within the GEOS-5 modeling framework. This makes MERRA-2 particularly suitable for investigating the diabatic influences on wave activity (cf. Lu et al., 2024; Smith et al., 2024). Importantly, blocking statistics from ERA5 and MERRA-2 show no significant differences, ensuring consistency in the representation of blocking events.

In the revised analysis, we examine the composite life cycle of column-averaged, pressure-weighted LH, LWCRE, and SWCRE anomalies from the onset to the mature stage of the blocking events in MERRA-2. We find that during the onset phase (e.g., days -6 and -3), positive anomalies of both LH and LWCRE consistently appear in the upstream region, west of the blocking center (**Fig. R1**). This contrasts with our original (old) Figure 6, which presented LWCRE only at 500 hPa and thus did not adequately capture the full vertical extent of the radiative heating. A similar, though weaker, evolution is found in the CTL simulation (**Fig. R2**). Therefore, our argument that LH and LWCRE are co-located in the upstream region during the onset phase remains robust. In the revised manuscript, we now present both the vertical structure of diabatic heating and the associated diabatic wave source (**Figs. 6 and 7**), as well as their column-integrated profiles (**Figs. S7–S12**), to avoid potential misinterpretations arising from only single-level diagnostics.

Fig R1. Blocking-relative composites of column-averaged pressure-weighted diabatic heating anomalies (LH, LWCRE, and SWCRE) during the onset of the block from MERRA2 (see Fig. S7 in the SI for details).

Fig R2. As in Fig. R1 but for CTL (see Fig. S8 in the SI for details).

Regarding the mechanisms, our revised manuscript discusses at least two possible pathways through which LWCRE feedback influences LH:

- **First mechanism:** LWCRE indirectly influences LH by affecting large-scale ascent and moisture transport. During the onset period, LWCRE significantly warms the mid-troposphere (850-450 hPa) in the upstream region in CTL (**Fig. R3f below**), whereas this warming is weaker in CLOCK (**Fig. R3g below**) and absent in LWOFF (**Fig. R3h below**). Without LWCRE-induced warming, the mid-tropospheric temperature remains cooler, leading to a stronger vertical temperature gradient. This condition enhances static stability and in turn, suppresses large-scale ascent. This weaker ascent is evident in the reduced negative vertical velocity (ω) anomalies in CLOCK (**Fig. R3i**) and LWOFF (**Fig. R3j below**) in regions of enhanced LH (see green box in **Fig. R3a**), indicating less efficient lifting of moist air into the mid-troposphere. The reduced ascent is accompanied by weaker vertically integrated moisture transport (IVT) into the WCB (**Figs. R3k-l** and **Fig.R4**), suggesting that less moisture is advected poleward upstream of the block. As a result, condensation is suppressed, leading to a decline in LH. This is consistent with previous studies showing that LWCRE enhances vertical motion and LH (Fu et al., 1995; Li et al., 2015).
- **Second mechanism:** LWCRE could also enhance LH through evaporation feedback. During the onset period, positive LW cloud radiative heating, LH, and evaporation anomalies are co-located in the upstream region of the block (see green box in **Fig. S9** and **Fig. R5 below**). This positive **LWCRE anomaly upstream** is primarily associated with increased **high cloud cover** (altitude >6 km, e.g., cirrus, cirrostratus), with a secondary contribution from low-to-mid clouds (see **Fig. R6 below**). An increase in high clouds can effectively trap downward LW radiation, which could lead to higher surface temperature (together with increased LH) and, potentially, to enhanced evaporation. Removing this effect reduces surface warming, leading to weaker evaporation and moisture supply. As a result, condensation is suppressed, leading to a decline in LH, consistent with previous studies (e.g., Fu et al., 1995).

We have now provided this supporting evidence and discussed them (see **L309-332**, in **Fig. 7** in the revised manuscript, and in **Fig. S7-S12** in the SI). We also emphasize that while all these pathways offer a plausible explanation for the interaction between LWCRE and LH, additional mechanisms may also contribute, and their relative significance remains an open question for future investigation (see **L333-337**).

Fig R3. Blocking-centered composites of diabatic heating, vertical wind velocity, and moisture transport anomalies during the onset of Euro-Atlantic blocking (see new Fig.7 in the manuscript for details).

Fig R4. A comparison of blocking-relative composites of IVT in MERRA and model experiments, averaged over the onset period (see new Fig. S10 in the SI for details).

Fig R5. Blocking-relative composites of (a-c) total cloud cover (TCC), (d-f) column-averaged pressure-weighted LWCRE, and (g-h) evaporation anomalies during the onset of the block in **MERRA2** (see new Fig. S11 in the SI for details).

Fig R6. Blocking-relative composites of cloud cover fraction at different altitudes from the onset to the mature stage of the block in **MERRA-2** (see new Fig. S12 in the SI for details).

2. Figure 6, physical meaning of positive LWCRE: It is quite surprising (and interesting) for me that the contribution of long-wave cloud-radiative effect is positive, not negative, despite that low-level clouds are dominant in the midlatitudes. I wonder if this physically makes sense and provides an insight. As noted in the introduction, the warm conveyor belt airstreams that trigger this life cycle of blocking occur mostly at mid-to-low troposphere, and there is radiative cooling from low-level clouds tops and heating at cloud bottoms (e.g., Cesana et al. 2019 and many others). Does it mean that radiative warming at the cloud bottom (peaks at the surface) outweighs its atmospheric cooling effect and acts as a positive diabatic source of wave activity? Providing a physical explanation of positive LWCRE in the discussion of Fig. 6 would help readers understand its importance more comprehensively.

Authors:

As addressed in our responses to general comment #1, the positive LWCRE anomalies upstream of the block during the onset period (lag -6 to -3 days) can be physically related to an increase in high cloud cover (HCC) upstream (see the observed cloud covers in **Fig. R6** or **Fig. S12** in the SI), with a secondary contribution from mid- and low-level clouds. The composite analysis indicates that positive HCC anomalies peak in the upstream region during the onset, coinciding spatially with enhanced LWCRE (see **Fig. R5** and **Fig. R6** above). The key mechanism at play here is that high clouds can contribute to radiative heating in the mid-to-upper troposphere due to their strong absorption and re-emission of longwave radiation. This heating alters the atmospheric thermal structure, enhances vertical ascent, and consequently increases LH. This relationship is evident in the vertical profiles of LH and LWCRE shown in **Figs. R3e,f**. In particular, the enhanced LH is vertically aligned with the mid-to-upper tropospheric LWCRE anomalies, suggesting a possible dynamical link between cloud radiative feedback and convective processes (see **L322-332**).

Although low-level clouds are dominant in the midlatitudes, their radiative effects are primarily concentrated in the lower troposphere, where they induce cooling at cloud tops and heating at cloud bases. However, the large-scale diabatic forcing relevant for wave activity in the upstream region of the block is found to be more sensitive to the high clouds rather than the near-surface warming from low clouds (although both could contribute to an increase in LWCRE see **Fig. R6**). Therefore, the net effect of LWCRE is positive due to the dominant contribution of high clouds upstream of the block, reinforcing the diabatic source of LWA that supports blocking formation. This physical connection strengthens the idea that cloud-radiative feedback plays an active role in modulating the warm conveyor belt ascent and its downstream impacts (on blocking).

We have included all these figures in the revised manuscript (**Fig. 7**) and the Supplementary Information (**Figs. S7–S12**) and have expanded the discussion accordingly (see **L309-335**).

3) Lines 472-484, Methodology-the validity of estimating radiative heating contribution of wave activity from equation (12): Related to the comment above, I am a bit confused of how the contribution of cloud radiative heating can be accurately estimated from the current methodology. If the results in Fig. 6a are taken at face values, it basically suggests that near-surface warming by upstream longwave radiative heating drives mid-tropospheric circulation anomaly. However, as we all know, change in atmospheric circulation is not simply driven by diabatic heating at near surface or certain height only.

Authors:

As addressed in our responses to general comments #1 and #2, we have revisited our proposed mechanisms and provided clearer pathways for how LWCRE may influence LWA (either directly or indirectly through LH). Our revised analysis clarifies that the relationship is not simply driven by near-surface radiative warming but rather by the broader interaction between cloud radiative effects, moisture transport, latent heating, and wave activity.

The vertical structure of radiative heating is complex and can sometimes induce counteracting responses in midlatitude circulation (e.g., Voigt et al. 2023). From my understanding, because of the height scaling factor in Eq. (12), it puts more (perhaps too much) weight to the near surface radiative heating values by construction and limitedly reflects the rich vertical structure of cloud-radiative effects. I think this is conceivable a potential caveat of the residual approach of the LWA diagnostics.

Authors:

We agree that the vertical structure of radiative heating is complex. However, the column-integrated LWA and LWA budgets (e.g., Eq. 12) are primarily influenced by mid-to-upper-level PV structures, as the wave amplitude is largest in these layers (see Zhang et al. 2025 and Huang & Nakamura 2017 (Fig. 2)). Consequently, the weighting is determined more by the distribution of wave-related PV structure (displacement of PV relative to the equivalent latitude) rather than near-surface radiative heating.

Reference:

- Zhang, R., E. K. M. Chang, and N. Nakamura, 2025: Wave packets and life cycles of troughs in the framework of local finite-amplitude wave activity. *J. Atmos. Sci.*, in press.
- Huang, C. S. Y., and N. Nakamura, 2017: Local wave activity budgets of the wintertime Northern Hemisphere: Implication for the Pacific and Atlantic storm tracks. *GRL*, 44, 5673–5682.

Can the authors show that this estimated LWCRE from Eq. (12) well represents the LWCRE in a more general condition (particularly associated with blocking onset)? Otherwise, adding a discussion on some methodological limitations may provide further insights into the study.

Authors:

We have now analyzed the vertical structure of diabatic heating and the associated wave activity source during the onset period using model simulations, and compared these results with MERRA-2, to better

identify the altitudes at which wave activity is generated or dissipated. Overall, our control experiment (CTL) reasonably captures the upstream structures of LWCRE and LH compared to MERRA-2 (see **Figs. R3a, b, e, f** or **Fig. 7** in the manuscript), despite some differences. The associated diabatic wave activity source due to LH and LWCRE is also well represented in CTL, showing broad agreement with MERRA-2 (see **Fig. 6** or **Fig. R7**).

In the revised manuscript, we now present the vertical structure of diabatic heating (**Fig. R3** or **Fig. 7** in the manuscript) and the associated diabatic wave activity source (**Fig. R7** or **Fig. 6** in the manuscript) to improve clarity. We also compare MERRA-2 and CTL results and discuss their differences, along with potential source of biases (see **L272-280** and **300-308**).

Fig R7. Contribution of diabatic heating terms to diabatic sources/sinks of wave activity during the onset of Euro-Atlantic blocking. (see new **Fig. 6** in the manuscript for details).

4) Lines 472-484, Methodology-implicit assumption on model representation of longwave heating: This study calculated the diabatic sources/sinks of wave activity using the simulated diabatic heating rates, which is different from previous studies (e.g., Neal et al. 2022) who estimated the total diabatic heating effects by suppressing positive diabatic forcing term in the LWA budget integration. While this explicit calculation thus adds a uniqueness to the study, it also hinges on an implicit assumption

that the model simulates the diabatic heating rates in a reasonable quality, which is not shown or mentioned at all in the manuscript. To put it another way, it seems important to ascertain whether the control experiment well simulates the horizontal and vertical structure of climatological diabatic heating and does not exhibit obvious errors.

Authors:

In the revised manuscript, we have now included a more detailed evaluation of the model’s ability to simulate diabatic heating and the associated diabatic source of wave activity. Specifically, our updated analysis (**Figs. 6 and 7**) shows that the CTL captures the vertical structure of diabatic heating, both LH and longwave cloud radiative heating (LWCRE), in the upstream region of the block reasonably well. A few differences are still evident in the magnitude and structure of the diabatic wave source compared to MERRA-2, which may stem from limitations in the representation of cloud radiative processes or from the complex interactions between eddies and cloud radiative heating during the onset phase of blocking in the model. Nonetheless, the CTL simulation reproduces the key diabatic heating patterns and associated wave activity sources with sufficient fidelity to support our process-level analysis. These comparisons are now discussed in **L272-280 and 298-300** of the revised manuscript.

Additionally, we note that the model’s climatological structure of CRE-related heating is broadly consistent with satellite-derived observational estimates (Harrop et al., 2024). In our separate work, led by Harrop et al., 2024, we evaluated the CTL CREs with observational data and found that, despite negative biases, the model reasonably reproduces CRE heating climatology at the surface, interior of the atmosphere, as well as the top of the atmosphere (see **Fig. R8 below** from Harrop et al., 2024).

We also now explicitly acknowledge and discuss the potential implications of CRE biases on blocking representation, as such biases are common across many climate models. In the revised manuscript (**L441-444**), we emphasize the need for future inter-model comparison studies -- particularly using cloud-locking experiments -- to more systematically assess the role of interactive CREs in blocking dynamics and to build consensus across models. In response to Reviewer #3, we have also added a new paragraph discussing the limitations of our current approach based on fixed SST lower boundary conditions (see **L438-444**).

Fig R8. Annual mean and zonal mean cloud radiative effect (CRE) at the (a) top of atmosphere (TOA), (b) surface, and (c) in the atmosphere. The solid lines use CERES-EBAFv4.1 data (NASA/LARC/SD/ASDC, 2019) and the dashed lines are from the control simulation. From **Harrop et al., (2024)**.

References:

Harrop, et al., An overview of cloud–radiation denial experiments for the Energy Exascale Earth System Model version 1, *Geosci. Model Dev.*, 17, 3111–3135, <https://doi.org/10.5194/gmd-17-3111-2024>, 2024.

5) Figure 7, significance of changes in mean states: It is repeatedly emphasized in the manuscript that the CLOCK experiment does not significantly change the mean states at all (particularly in the conclusion section), and thus any notable difference between CLOCK and control simulations is attributable to interactive CREs. However, I am not sure if this is entirely true, considering that there are still some hints of poleward displacement of the North Atlantic jet, weakening of the southwest-northeast tilt of stationary eddies, and reduced maximum Eady growth rate over the Gulf Stream and the entrance of the North Atlantic storm track. I feel these features will be more pronounced if colored stippling is used for statistical significance. Can the authors quantitatively show these mean state changes in CLOCK are negligible compared to LWOFF? In any case, the authors need to tone down statements that overstate the absence of significant change in the mean state for CLOCK throughout the manuscript, as there are actually some significant changes.

Authors:

We have now clarified in the manuscript that the impact of CLOCK on the background mean state is less disrupted compared to the LWOFF experiment. This is because, by design, the simulation with locked CREs introduces only minor changes to the climatological CRE fields, while effectively decoupling the high-frequency covariance between CREs and individual weather systems (cf. Grise et al., 2019; Benedict et al., 2020; Harrop et al., 2024). See our revised text (**L338-375**, **L379-380**, and **L408-L409**).

We have also corrected an error in our statistical significance test in our old Fig. 7, in which all regions were previously marked with white dots. In the revised analysis, significance is now correctly evaluated using a two-sided *t*-test and indicated with black stippling instead. Our analysis confirms that while modifications in jet position, stationary wave amplitude, Eady growth rate (EGR), and moisture transport are evident in CLOCK, their magnitude remains much weaker than in LWOFF (see new **Fig. 8** in the manuscript).

6) Figures 5, 6, 7, differences in moisture transport: Changes in the climatological mean states from the CLOCK experiment are seemingly small (Fig. 7), but it does not necessarily mean that such changes are trivial in modulating atmospheric blocking frequency. One possible pathway of their impacts is a reduction of poleward moisture transport over the upstream of Euro-Atlantic blocking. If the climatological moisture transport is reduced in CLOCK, compared to the control run, it indicates that the moist processes driven by stationary eddies are weaker and therefore contribute to the reduction of condensational heating (Fig. 6a), in addition to the contribution from interactive CREs.

Authors:

Thank you for the suggestions. After examining the climatological mean response of moisture transport in CLOCK and LWOFF relative to CTL, we found a significant reduction in mean poleward moisture

transport (IVT) in both experiments, with the largest reduction found in LWOFF. This reduced moisture transport without interactive or mean CRE is associated with a weaker WCB, resulting from the cloud radiative impact on large-scale ascent (**Fig 7**), as well as CRE-induced weakening stationary waves that modulate poleward moisture transport (**Fig. 8c-d**). This finding supports our new proposed pathway in which LWCRE enhances LH by strengthening poleward moisture transport upstream of Euro-Atlantic blocking. These results are now presented in the updated **Fig. 8** and discussed in the manuscript (see **L309-321 and L366-371**).

7) Figure 8, schematic diagram and figure caption: The diagram is reminiscent of Figure 1 in Steinfeld et al. (2022) and seems to build upon previous findings. However, I am somewhat skeptical about whether this schematic illustration accurately reflects the results presented in this study. The first step, ‘moist air injection’, is not explicitly shown in this study but used without any reference being provided in the caption. Regarding the third step, as noted in the earlier major comments, LWCRE is not located upstream of the blocking center, but rather within it. This differs from illustration. Moreover, the process described in the figure caption (i.e., ‘The enhanced LWCRE in turn amplifies LH by boosting evaporation’) is not clearly demonstrated in this study, resulting in an unsubstantiated statement. The term ‘divergent outflow’ at the upper troposphere also abruptly appears without any context. Lastly, it feels a bit strange to see the vertical structure of the proposed mechanism for the first time in this study as a schematic diagram. I recommend the authors either replace it with a new diagram based on the results of the study within the framework of LWA diagnostics or provide a list describing the chain of processes in order.

Authors:

We have improved the schematic diagram by adding an upper-level LWA illustration to highlight the connection between the diabatic wave source, large waviness in the PV contours, and blocking formation. A large PV displacement explains the strong LWA downstream associated with the block, which is supported by an intensified diabatic wave source upstream. This source is primarily driven by increased LH and LWCRE, with LWCRE providing positive feedback on LH (as indicated by the dashed arrow). The dashed arrow represents the indirect influence of LWCRE on LH via enhanced poleward moisture transport and potentially via increased surface evaporation. Together, the increased LH and LWCRE strengthen upward motion along the WCB, further amplifying LH. In the absence of LWCRE, this feedback mechanism weakens (see new **Fig. 9 in the revised manuscript** and **Fig. R9** below).

In the revised schematic, we also introduce a flowchart to clarify the key processes. While the vertical 3D structure of WCB is broadly similar to previous studies (e.g., Grams and Archambault, 2016; Quinting and Grams 2021; Steinfeld et al. 2022), our focus here is on the role of the diabatic wave activity source within the WCB, particularly in the presence of CREs, in modulating LWA.

The evidence for the “moist air injection” is now explicitly analyzed in **Fig. 7 and Fig. S10**. However, in the schematic, we refer to this process more generally as “moisture transport inflow (WCB)” for clarity.

Fig R9. Schematic diagram illustrating how CREs increase upstream diabatic heating and promote a block formation (see new Fig. 9 in the manuscript for details).

Minor comments:

1. Line 36: ‘there is no comprehensive dynamical theory’

Corrected (see L37).

2. Lines 35-36: ‘simulating and predicting blocking events ... are rare and localized’. This sentence could flow better. I recommend rephrasing it.

Rephrased (see L35-L37) “Despite its importance, simulating and predicting blocking events is notoriously challenging due to the lack of a comprehensive dynamical theory and because blocks are rare and localized”.

3. Line 10, 39: ‘notoriously underestimate’. I understand that the authors want to strongly motivate the study by pointing out the systematic model biases, but it sounds too strong. Also, in terms of the anomaly method for blocking detection, note that some regions show overestimating biases (Woollings et al. 2018). I recommend an expression like ‘mostly underestimate’.

Rephrased (see L41).

4. Line 46: ‘SST’. This acronym is not defined beforehand.

Corrected (see L49).

5. Line 53: ref. 34. Information of ref. 34 is missing in the reference.

Corrected.

6. Line 101: ‘climatology in winter (December to January)’. There seems a mismatch between this expression and the figure caption. I guess ‘(December to February)’?

Yes, winter DJF. Corrected. (L104)

7. Figure 1 caption: It would be helpful to expand all acronyms at their first usage in captions.

All acronyms (e.g., CTL, CLOCK, LWOFF) are defined in the main text when describing the figure. Owing to space constraints, they are not included in the caption.

8. Hatches in Fig. 2a: What does this black hatch mean?

Region where at least 10% of the models disagree. We remove this to avoid confusion.

9. Line 242: ‘stronger southerly flow ($V_e < 0$)’. Is this sign error? I think $V_e > 0$ represents southerly flow, not $V_e < 0$.

Corrected.

10. Line 244: ‘) in the upstream’ -> ‘in the upstream’

Corrected.

11. Lines 275-276: ‘Reduction in latent heat release ... evaporation’. Alternatively, this could be simply due to a weaker warm conveyor belt airstream and the associated moisture transport. See major comment 6.

Yes, this is mainly due to a weaker WCB, but potentially also due to enhanced surface evaporation. We have revised the discussion accordingly (see L309-332).

12. Line 300: ‘is minimal’ -> ‘are minimal’.

We remove this sentence.

13. Lines 308-311: This sentence states that there is a ‘partial weakening’ of the climatological ridge over Europe, but then it is followed by an expression ‘Despite no change in the mean state’. Based on the results in Fig. 7, further investigation seems necessary. See major comment 5.

We revise the sentence (L346-350). “*The CLOCK experiment exhibits a very weak poleward jet shift (Fig. 8a) and a partial weakening of the planetary-scale anticyclonic ridge associated with stationary waves over Europe (Fig. 8c). In contrast, LWOFF exhibits a more pronounced poleward jet shift (Fig. 8b) and a stronger weakening of the planetary-scale ridge over Europe.*”

14. Line 321: ‘the observed reduction ... over the region’. It sounds like the authors are describing observational results. I recommend rephrasing it.

Revised (L351-353) “...consistent with the reduction in blocking frequency in both experiments, with a substantially larger decrease in LWOFF than in CLOCK”.

15. Figure 7 caption: Is the unit of the streamfunction correct? It seems a factor of 10 is missing.

Corrected. It must be $\times 10^5$ m²/s.

16. Color of stippling: I strongly recommend changing the color of the stippling. It is very difficult to identify.

Yes, we have now changed it to black.

17. Lines 377-378: ‘the distinctive role of diabatic wave activity injection ... anticyclone’. I feel this sentence could flow better, and its structure also looks somewhat too similar to the original sentence in Neal et al. I recommend rephrasing it.

Revised (L412-414) “..the importance of diabatic wave activity generation within the WCB region of an upstream cyclone in developing a downstream blocking anticyclone”

18. Line 442: ‘a blocking-based anomalies index’. I think the authors meant ‘an anomaly-based blocking index’.

Corrected.

19. Line 480, Equation 13: Is total diabatic heating composed of only these four terms? I thought there is also a heating term of vertical diffusion (or sometimes referred to as diabatic heating due to turbulent mixing). Also, I guess longwave CRE (LWCRE) is derived from the difference between net longwave heating and clear-sky radiative heating, correct (same for SWCRE)? If so, please add this detail.

The contributions from turbulent mixing and surface friction are negligible compared to the four dominant processes and are therefore omitted. In fact, our condensation heating term (COND) includes heating from all moist microphysical processes, including turbulent heating and convective heating. So, for simplicity (as suggested by reviewer #2), we simply refer to it as 'latent heating (LH)' in the manuscript. The LW and SW cloud heating rates are defined as the difference between all-sky and clear-sky rates (see L497, L543-544).

Reviewer #2 (Comments to the Author):

This study investigates the radiative impact of clouds on the dynamics and frequency of midlatitude atmospheric blocking during winter. Using two modeling techniques, the authors assess the impact of cloud radiative effects (CREs) on the blocking and show that CREs significantly increase the frequency of blocking over the Euro-Atlantic sector. To gain a process-based understanding of the radiative impact of clouds on the blocking dynamics, a budget analysis of the local wave activity was performed for different simulations. The results indicate that CREs directly affect the onset of blocking by enhancing the diabatic source of wave activity upstream of the blocks. Furthermore, the results indicate that CREs can also indirectly affect the dynamics of blocking through changes in the mean atmospheric state, such as changes in the position of the jet stream and the eddy growth rate.

Overall, this is a well-structured study that highlights the importance of cloud-radiation-circulation coupling for the dynamics of atmospheric blockings that are associated with high impact weather extremes. Over the last decade, the importance of cloud-radiative effects on midlatitude dynamics, either on climate or weather time scales, has gained attention in climate studies, and with it the need to improve the simulation of CREs in models. A better understanding of blocking dynamics and a better simulation of CREs in models are both important aspects of modeling, and therefore I find this study interesting and useful as it provides new insights into midlatitude dynamics.

The methods used in the study are well established and documented in previous studies. The authors used both well-known cloud locking and COOKIE methods to assess the impact of CREs on blocking. The results were also validated by comparing the impact of CREs from other model simulations, and overall, the authors find a robust result regardless of the model used. In addition, the suitability of the diagnostics used was assessed using reanalysis data. Overall, I consider the study suitable for publication, although I have some remarks, questions and minor revisions.

Authors: We thank the reviewer for the positive feedback and thoughtful, detailed comments on improving the manuscript. A point-by-point response to each comment is provided below, highlighted in blue.

Remarks:

1- As mentioned in the preprint, many factors contribute to the formation of blocking. I am interested to know how the impact of CREs compares to the other factors. If possible, some information in the discussion can also help to put the impact of CREs into perspective.

Authors: We agree that atmospheric blocking arises from a complex interplay of dynamical and thermodynamical processes influenced by multiple factors (see L46-53). While our results provide clear evidence that CREs modulate key diabatic processes (such as latent heating and moisture transport) that influence blocking formation, quantifying their relative importance compared to other factors (e.g., SST forcing, sea-ice forcing, land forcing, etc.) remains an open question. Addressing this

would require a comprehensive suite of targeted experiments in which individual processes are selectively removed, which is beyond the scope of the current study.

Also, since CRE uncertainties are smaller than the CRE itself, can we expect to see a significant impact of CRE uncertainties on blocking frequency? For example, the study of Zhao et al., 2018 (<https://www.sciencedirect.com/science/article/pii/S0169809517309122>) shows that uncertainties in CREs could affect the mean circulation. This also makes me wonder about the time scale of the impact of CREs. It may be the case that uncertainties in the simulation of CREs affect the blocking frequency on longer climate time scales of years, but do we expect uncertainties in CREs to affect blocking events on time scales of weeks or during the blocking onset period? (This relates to the argument in lines 401-402: "minimizing CRE-related model biases, could therefore potentially enhance early-warning systems to mitigate damages caused by such events.").

Authors: While our results suggest that CREs influence blocking formation on synoptic timescales, we also find evidence for their longer-term impacts on blocking climatology (see **Fig. 1**). It is plausible that, although CREs modulate individual blocking events over short periods, their cumulative effect becomes more pronounced in shaping the mean state and the overall blocking frequency climatology (see **Fig. 1 and Fig. 8**).

That said, quantifying the influence of uncertainties in CREs on blocking frequency remains an open question. Addressing this would require targeted inter-model comparison studies using CRE-focused experiments, where differences in CRE representation could be explicitly linked to blocking behavior. Such an approach would help assess whether model uncertainty in CREs translates to meaningful differences in blocking statistics. We now briefly discuss this direction in the revised manuscript (see **L432-444**).

Finally, can the authors elaborate on which method is better to quantify the impact of CRE on blocking for medium-range weather forecasts?

Authors: A practical approach would be to perform sensitivity experiments, similar to those presented in our study, but using ensemble forecasts with and without CREs. These experiments could be complemented by data assimilation techniques to evaluate how CRE-induced processes influence the predictability and evolution of blocking events. Additionally, comparing forecast skills across operational models with differing CRE parameterizations could also offer valuable insights into the role of CREs in medium-range forecast skills. In summary, better representation of CREs, particularly within WCB, may improve the simulation and prediction of blocking in medium-range weather forecasts (see **L429-431**).

2- Considering that SST and sea ice are fixed in the simulations performed in this study (also preserving the mean state in cloud locking), it seems better to use the term cloud-radiative heating (CRH) rather than cloud-radiative effects (CRE)? CRE is more general and includes the impact of clouds on surface radiative effects. The term Atmospheric Cloud Radiative Effect (ACRE, (K/s)) has also been used

before (e.g., Voigt et al., 2020, <https://wires.onlinelibrary.wiley.com/doi/full/10.1002/wcc.694>), which separates the radiative effect of clouds in the atmosphere from that of the surface. Or perhaps it should be mentioned in the paper that the radiative heating of clouds in the atmosphere is actually being studied?

Authors: There is currently no strict consensus on the use of the terms "cloud-radiative heating" (CRH) or "cloud-radiative effects" (CRE), as they are often used interchangeably in the literature. In many cloud-locking studies (e.g., Grise et al., 2019; Medeiros et al., 2021; Lu et al., 2024), the term CRE is commonly used to refer to the atmospheric radiative heating associated with clouds. Even in Voigt et al. (2020), although the term ACRE is introduced to distinguish atmospheric from surface effects, the term CRE is still used when discussing cloud-locking experiments (e.g., **Section 3.2 and Fig. 2**). For consistency with the previous terminology in cloud-locking studies, we decide to use CRE throughout the manuscript.

Following on from this, I was wondering if the results might change if the interactive ocean was coupled to the model simulations rather than the SST being prescribed. Can the surface CRE also affect the blocking?

Authors: As described in the manuscript, our simulations are run with prescribed SSTs, which was aimed to isolate the role of CREs in blocking, with minimal interference from lower-boundary SST-related forcing. However, we agree that, in the real world, CREs may interact with SSTs, and they can jointly modulate blocking formation (e.g., Yamamoto et al., 2021; Matthews et al., 2024). For example, Matthews et al. (2024) demonstrate that suppressed surface moisture fluxes over the Gulf Stream can reduce atmospheric blocking frequency across the Northern Hemisphere. Understanding the coupled impact of CREs and SSTs on blocking would require targeted experiments using a fully coupled ocean-atmosphere model. We have now discussed this limitation of our study in the revised manuscript (see L438-444).

3- Are all temperature tendencies from the physical parameterization considered in the analysis of the diabatic source of LWA? In Fig. S6 the "Res" is attributed to dissipation processes, but in Eq. 13 only the heating rates from radiation and condensation are considered. What about heating from turbulence or convection? I am also confused by the changing notion of latent heating and condensation throughout the text. In your analysis, does condensation heating (COND) include heating from all microphysical processes and heating from convection? If so, I suggest calling this just latent heating.

Authors: The condensation heating term (COND) from our model includes all moist diabatic heating contributions from microphysical processes, including those associated with convection and turbulent heating. To avoid confusion and ensure consistency, we now refer to this term as latent heating (LH) throughout the manuscript, as suggested by the reviewer.

4- Regarding the repeated argument of LWCRE feedback on latent heating by enhanced evaporation. Can the authors elaborate on this? The preprint refers to studies that actually show that CRE warming

in the tropics is balanced by less latent heating. There are studies that indicate the mechanism of the impact of longwave radiation on clouds and latent heating. For example, Fu et al., 1995 showed that LW radiation enhance condensation and droplet growth. Or by affecting the eddies that can promote stronger vertical motion and hence latent heating (Li et al., 2015).

Authors: As also suggested by Reviewer #1 (see Pages 5-7), we have revisited the mechanisms by which LWCRE influences LH. Our revised analysis suggests that: (1) LWCRE enhances LH indirectly by strengthening large-scale ascent and poleward moisture transport, and (2) LWCRE enhances LH through evaporation feedback. These mechanisms are consistent with Li et al. (2015) and Keshtgar et al., (2023), who showed that LWCRE enhances vertical motion and LH, and Fu et al. (1995), who found that LW radiation promotes condensation (LH). We have also cited all these references in the revised manuscript. These new results and discussion have been added to the revised manuscript (see **L309-337, Figs. 6-7 and Figs. S7-S12**).

Minor revisions:

Line 15: ... (CREs significantly increase the formation of) - "CREs significantly enhance the formation of blocking or increase the frequency of blocking"

Corrected. "*CREs significantly increase the frequency of Euro-Atlantic blocking.*" (L15)

Line 46: ... sea surface temperature (SST)

Corrected.

Lines 43-44: ... (models mean state) - "model representation of mean atmospheric state and/or the waves at ..."

Corrected.

Line 53: reference 34 is missing.

Corrected.

Line 54: ... (blocking dynamics were mostly presumed to be predominantly dry) - "the dynamics of atmospheric blocking is predominantly driven by dry dynamics"

We use the word 'presumed' here because, a few decades ago, blocking dynamics were assumed to involve only dry physical processes, without moist dynamic contributions.

Line 68: ... (resulting from the presence of clouds) - "resulting from cloud-radiation interaction (absorb, emission, ...)"

Here, we refer to the conventional definition of CREs as the radiative heating due to clouds (the difference between all-sky and clear-sky rates).

Line 74: ... (over the extratropical ocean across models), I think reference 41 shows the impact of CRE

on tropical ocean rather than extratropical ocean, reference 37 talks about the impact of CRE on extratropical ocean.

Corrected “over the tropical and extratropical ocean across models” (L77).

Line 77: reference 42 do not directly show the impact of CRE on forecast error growth, rather the impact of all-sky radiation on error growth.

While the reference focuses on all-sky radiation, the authors state that “the stochastic convective scheme... represents explicitly resulting variability by **reshuffling of the individual clouds**” which inherently involves CREs (similar to our CLOCK). We therefore cite it to highlight the role of CRE-related variability in error growth.

Line 83: ... (greater clarity into), greater than? I believe compared to COOKIE methods? In this section I believe there is a need to shortly explain the difference between the two methods to help smoothly follow the story (even though it is explained later and in the method).

Thank you for the suggestion. We meant that “isolating the role of CRE in this way provides greater clarity into its impact on simulated variability.” We further explain the differences between these two methods in the experiment design section.

Line 84: ... "used cloud locking to study the impact of CRE on" Also, in reference 39 the cloud locking method is not used rather than just disabling CRH and applying COOKIE method.

Yes, Ref39 has been removed.

Line 87: ... (to longwave radiation), I understand that the impact of shortwave CRE is negligible but would be good to mention it.

Our LWOFF setup is similar to the COOKIE type 2 configuration, where clouds are made transparent to longwave radiation only. This approach minimizes climate drift caused by large surface energy balance changes that occur when both shortwave and longwave radiation are completely turned off (Webb et al., 2017).

Line 90: might be helpful to shortly mention here or in the discussion why CRE is biased in weather and climate models.

We prefer not to elaborate here, as this part highlights key results. The details are provided later in the discussion.

Figure S1: caption: shading (tendency) unit is m day⁻¹

Corrected.

Line 118: The reference to Fig. 1d with the result of 21% decrease in blocking frequency seemed at first confusing.

Corrected Figs. 1c, d.

Line 126: "Clouds On/Off Klimate Intercomparison Experiment (COOKIE)"
Corrected.

Line 129, as only the cloud radiative heating is disabled in these simulations, I suggest the term clouds-off LW or CLWOFF or something similar instead of LWOFF.

We use the term LWOFF to maintain consistency with COOKIE-2 terminology, which refers to the experimental setup in which clouds are transparent to longwave (LW) radiation only (cf. Medeiros et al., 2021; Webb et al., 2017).

Line 142: also, please refer to Table 1 here
Added.

Figure 2: caption: remove ((b) As in (a))?
Corrected.

Line 160: density-weighted vertical average, what is the interval of the vertical integral in this analysis? The vertical integral is computed over the full atmospheric column using MERRA-2 data. It has been clarified in the text:

$$\langle (\cdot) \rangle \equiv \frac{\int_0^{\infty} e^{-z/H} (\cdot) dz}{H}$$

The data is interpolated onto a regular pseudo-height (z) grid with $dz = 1\text{ km}$ before the computation of vertical integral. The upper limit of the integral is taken to be $z = 48\text{ km}$ ($p \sim 1\text{ hPa}$).

Line 162: remove (RHS), only used once
Done.

Line 169: Is A "wave activity" or local wave activity (LWA)?
Yes, LWA. Corrected.

Figure 3:

- (a-d) do these panels show the left-hand side term of Eq. 1 or sum of right-hand side terms?
As stated in the caption, these panels show both sides of Eq. 1.

- Caption: meridional eddy heat flux at the base of the atmosphere or at the lower boundary? And which level is that?

Following Huang and Nakamura (2017), the meridional eddy heat flux is calculated at the lower boundary using the two lowest model levels. The implementation is provided in the "Code Availability" section.

- Caption: ((m-p) sources and sinks of wave activity) - "(m-p) nonadiabatic sources and sinks of the wave activity"

Corrected.

Figure S3: ... blocking -in- the CTL ..., also change Lag to days, consistent with Fig 3

Corrected.

Line 209: OLR change to "outgoing longwave radiation", here would be good to also mention that minimum OLR and max water are associated with the position of WCB.

Revised (see L215-216).

Line 212-220: nonadvective or nonadiabatic?

The residual of LWA budget may include nonadiabatic sources of wave activity and nonadvective processes not represented in the budget diagnostics. To be consistent with previous works (Huang and Nakamura 2017; Nakamura and Huang 2018), we now refer to this as "**nonconservative (diabatic) sources and sinks of LWA**". We have corrected this throughout the manuscript.

Figure 4: is the "blocking high center" refer to the position of maximum geopotential anomaly?

Yes, the blocking high center refers to the local maximum of the 500-hPa geopotential height anomaly. This has been clarified in the figure caption.

Line 274: ... (eliminates the LWCRE to LWA) - "eliminates the impact of LWCRE on LWA"

Corrected.

Line 282: ... from LWCRE and COND, (introduced before)

We have removed this sentence and modified this paragraph (see L289-301).

Figure 6: caption: ... Euro-Atlantic blocking ((remove shading), unit: $m\ s^{-1}day^{-1}$), In (remove all panels (b-g)

Corrected (see new caption Fig. 6).

Lines 288-289: please refer to Fig. 6a and Fig. S6

Added.

Figure 7 caption: I suggest rewriting the caption, as using phrases such as 'as in (a-b)' does not imply what contours are shown in other panels.

We have improved the figure's caption (see Fig. 8).

Line 310: From Figure 7 I tend to say that the mean state is in fact changing in the CLOCK?

Yes, as also noted by Reviewer #1, we have revised this section to address the mean state change in CLOCK. Please see the updated text in Section 3.2.

Line 358: ... where clouds are fully transparent to longwave radiation.
This part has been revised (see L383-394)

Table 1: Clouds On/Off Klimate Intercomparison Experiment
Corrected.

Eq. 3: maybe better to bring equation 4 before 3 as "q" is not in Eq.2
We prefer to keep the current order, as q_e in Eq. 3 is defined based on q , which is introduced earlier.

Line 462: Qref change to QREF, consistent with other terms
Corrected.

Line 470: revise to "The density-weighted vertical average of LWA ("column LWA") tendency reads as"
We prefer to retain the term "column mean of LWA" to remain consistent with terminology used in previous studies.

Line 475: Non-advective or nonadiabatic?
Corrected. "Nonconservative process".

Ref 44: There is newer reference to Harrop et al., 2024 (<https://gmd.copernicus.org/articles/17/3111/2024/>): (Harrop, B. E., Lu, J., Leung, L. R., Lau, W. K. M., Kim, K.-M., Medeiros, B., Soden, B. J., Vecchi, G. A., Zhang, B., and Singh, B.: An overview of cloud–radiation denial experiments for the Energy Exascale Earth System Model version 1, Geosci. Model Dev., 17, 3111–3135, <https://doi.org/10.5194/gmd-17-3111-2024>, 2024.)
Updated.

Reviewer #3:

Review comments on Lubis et al.

In this study, the authors present compelling evidence on the critical role of the cloud radiative effects (CREs) in enhancing blocking frequency. By using a suite of idealized simulations and applying local wave activity diagnostic, they show that CREs amplify blocking frequency by strengthening upstream wave activity. Investigating the impact of CREs on blocking dynamics and frequency is novel, and their idealized simulations offer convincing support for the significant influence of CREs on wintertime blocking frequency, employing a solid dynamical framework.

However, the manuscript, in its current form, requires substantial refinement. I have several questions and concerns regarding the methodology of their idealized simulation, as well as the interpretation of their results. Particularly, the experimental setup contains inconsistencies that hinder reproducibility, and the interpretation of results lacks sufficient explanations of underlying physical mechanisms. My detailed comments are as follows.

Authors: We thank the reviewer for the positive feedback and thoughtful, detailed comments on improving the manuscript. A point-by-point response to each comment is provided below, highlighted in blue.

Major Comments:

1. Methodology:

- From my understanding, all the idealized simulations run in this work use the same SSTs and SICs computed from a 20-year monthly climatology from 1990 to 2010 (see my second minor comment below). With this choice of the boundary condition, I am concerned that the authors may be underestimating moisture sources from the ocean to the atmosphere, particularly over the Gulf Stream and North Atlantic Current. By averaging out mesoscale ocean eddies, which are prevalent in these regions, the simulations may suppress surface latent heat flux, potentially reducing moisture availability to the atmosphere. Previous studies (e.g., Yamamoto et al. (2021, WCD), Matthews et al. (2024, GRL)) have shown these regions to be critical moisture sources for latent heating associated with Euro-Atlantic blocking events. I recommend that the authors consider using non-averaged boundary conditions. If that is not feasible, please provide a clear justification for the choice of this SST/SIC boundary condition and acknowledge this potential limitation of this work in the main text.

Authors: We have included additional details in the model experiment section, particularly concerning the rationale for using a fixed SST lower boundary (see “Numerical Experiments”). As is now emphasized in the manuscript, prescribing SSTs as boundary conditions enabled a clearer attribution of the role of CREs in blocking formation, independent of confounding influences from

SST-related forcing. However, we agree that, in the real world, CREs may interact with SSTs, and they can jointly modulate blocking formation (e.g., Yamamoto et al., 2021; Matthews et al., 2024). For example, Matthews et al. (2024) demonstrate that suppressed surface moisture fluxes over the Gulf Stream can reduce atmospheric blocking frequency across the Northern Hemisphere. Understanding the coupled impact of CREs and SSTs on blocking would require targeted experiments using a fully coupled ocean-atmosphere model. We have now added to the discussion section to address the limitations of the results in light of this choice (see **L437-443**) and cited these related works (e.g., Yamamoto et al., 2021; Matthews et al., 2024).

- For CLOCK experiment, why do authors prescribe the cloud optical properties only from the first three years of the CTL simulation and not the entire 20-year period of the CTL simulation? The authors describe how in this manner they can ensure the similar CRE climatology to CTL, but I am not convinced. Furthermore, the authors write how they loop over the three-year cloud optical property files for the eleven-year cloud locking experiment, which sounds inconsistent with the description that the experiment is run for 20 years. Please rewrite the paragraph to clarify the above points if it is my mere misunderstanding. If not, I recommend that the authors undertake the simulation with the use of the CRE file with the entire 20-year period of the CTL simulation to truly ensure that both simulations have the same CRE climatology.

Authors: We prescribe cloud optical properties in CLOCK by looping over a representative 3-year period from the first part of the CTL run. This choice follows the common practice in cloud-locking experiments (e.g., Grise et al., 2019; Voigt et al., 2021) and is intended to retain a realistic and internally consistent CRE climatology while reducing computational burden. Prior studies have shown that even a single year of cloud properties can be sufficient for cloud-locking configurations (e.g., Middlemas et al., 2019), so we do not anticipate issues related to undersampling when using three years. Using a multi-year loop rather than a fixed year helps avoid unphysical repetition of cloud patterns while still capturing key variability and climatology of CREs (Harrop et al., 2024). For further details on our CRE model configuration and implementation, we refer the reader to Harrop et al. (2024, GMD; <https://doi.org/10.5194/gmd-17-3111-2024>). We have now made this point clearly in the manuscript (see **L465-470**).

- The usage of 1979-2020 ERA5 data is not quite consistent with the experimental design where the authors chose to use 20-year integration based on the 20-year climatological SSTs and SICs. In order to make a fair comparison with the model results, please use 1990-2010. Also, please add the results from ERA5 on Figs 4 and 5 as well.

Authors: We appreciate the reviewer's suggestion. However, we would like to clarify that our model simulations are not forced by transient SST patterns and do not include interannual SST variability. Instead, we use prescribed monthly climatological SSTs and sea ice concentrations (SICs) averaged over 1990-2010, which repeat annually, as described in Harrop et al. (2024). This is aimed to cleanly isolate the dynamical influence of cloud radiative effects on blocking,

independent of variations in lower-boundary SST forcing. Since the simulations are not intended to reproduce the transient SST forcing or year-to-year variability as observed in the real atmosphere, using the full 1979-2020 period of reanalysis enables a larger sampling of blocking events, thereby increasing the statistical robustness of the comparison.

In response to the reviewer's request, we have now added the reanalysis results to the updated **Figs. 4, 5, 6, and 7** for direct comparison with the model output.

- The decomposition of the total diabatic sources/sinks of wave activity is a great attempt, and I found the results quite interesting. I recommend that the authors use JRA55 reanalysis data, which I believe provides the necessary diabatic terms, to compare and validate their results.

Authors: We have now decided to use MERRA-2 instead of ERA5 as our observational-based reanalysis. MERRA2 provides comprehensive diabatic heating terms, including contributions from latent heating and cloud radiative effects, which are essential for our local wave activity budget analysis. These heating terms are derived from the assimilation of satellite observations within the GEOS-5 model framework, making MERRA-2 particularly well-suited for process-level diagnostics involving diabatic influences on wave activity (cf. Lu et al., 2024; Smith et al., 2024 who have also analyzed LWA and the associated diabatic wave sources using the same framework) (see L490-497).

2. Interpretation of the physical significance of CRE

From the model results that are presented in this manuscript, I agree with the authors that CRE does seem to significantly -- although secondarily to latent heating -- contribute to increase the wintertime Euro-Atlantic blocking frequency. However, the specific role of CRE in atmospheric blocking remain unclear. In the introduction, the authors review prior studies on the impact of CRE on large-scale atmospheric circulation, yet they do not discuss the physical processes directly relevant to blocking. Since CRE can induce both cooling and heating effects depending on cloud type, and blocking involves high-pressure systems (clear skies) alongside warm conveyor belts (precipitating clouds), the authors should clarify their hypothesis on how CRE may influence blocking. A more detailed physical explanation would strengthen their argument, especially regarding Figures S5 and 6(b).

Authors: As also suggested by Reviewer #1 (see Pages 5-7), we have revisited the mechanisms by which LWCRE influences LH. Our revised analysis suggests that: (1) LWCRE enhances LH by strengthening large-scale ascent and poleward moisture transport, and (2) LWCRE can also enhance LH through evaporation feedback due to a presence of high-clouds in the upstream region. These mechanisms are consistent with Li et al. (2015) and Keshtgar et al., (2023), who showed that LWCRE enhances vertical motion and LH, and Fu et al. (1995), who found that LW radiation promotes condensation (LH). We have included all these references in the revised

manuscript. These new results and discussion have been added to the revised manuscript (see L309-337, Figs. 6-7 and Figs. S7-S12).

Minor Comments:

1. The authors write that for CTL they use initial condition of the year 2000s, but which exact years do they use precisely? Or do they mean they used the atmospheric condition of year 2000?

Authors: The control run (CTL) uses the atmospheric forcing representative of the present-day climate conditions (using the initial condition of the year 2000 of the free-running E3SM v1 DECK AMIP simulation) (see L458-L459).

2. In 1.127 the authors write that they use SST from year 2000, while in 1.425 they describe that they use the same SSTs and SICs as CTL. Please amend this inconsistency accordingly.

Authors: We have revised this part (see L455 and L459-461).

3. Detailed description of the methodology of the COOKIE-2 simulations is completely missing, and the authors need describe it in detail in the Method section. Also, I found it very confusing that the authors use COOKIE-2 and CFMIP interchangeably.

Authors: We have added a brief description of the COOKIE-2 simulation in the Methods section and refer readers to Webb et al. (2017) for full details. To clarify, COOKIE refers to a type of model experiment designed to isolate cloud-radiative interactions, whereas CFMIP (Cloud Feedback Model Intercomparison Project) is the broader international modeling initiative under which COOKIE-2 simulations were coordinated. We have clarified this in the text (L485-487)

4. How many blocks are detected in each experiment and thus included in each composite maps?

Authors: Based on 41 years of data, MERRA-2 identifies 94 blocking events. In E3SM experiments based on 20 years of data, the CTL, CLOCK, and LWOFF simulations produce 58, 46, and 36 events, respectively. These event counts are now included in the updated figure 5 for reference.

5. The authors attribute the LWA sources almost solely to the diabatic heating term, which is not in line with the previous studies (e.g., Pfahl et al. 2015, Steinfeld and Pfahl 2019, and Yamamoto et al. 2021 all found ~50% associated with diabatic heating, and the rest adiabatic). How can you explain this discrepancy?

Authors: We appreciate the reviewer's comment. However, we would like to clarify that we do not attribute the LWA source solely to the diabatic heating term. As discussed in the manuscript,

we specifically conclude that the diabatic source of wave activity upstream of the block region is primarily associated with cloud latent heating and longwave cloud radiative effects (L390-398).

In the LWA budget equation (**Eq. 5**), the last term explicitly represents nonconservative (diabatic) processes and does not account for adiabatic contributions (e.g., see Huang and Nakamura 2017; Nakamura and Huang 2018). The first two terms on the right-hand side of **Eq. 5** can be considered adiabatic sources of wave activity, arising from (1) horizontal (primarily zonal) advection of LWA and (2) low-level source of wave activity associated with poleward meridional eddy heat flux. Depending on the region (e.g., upstream vs. downstream of the block), the sum of these adiabatic contributions can be much larger or comparable to the diabatic source term, as shown in **Fig. 5**.

Therefore, our results do not contradict previous studies (e.g., Pfahl et al. 2015; Steinfeld and Pfahl 2019; Yamamoto et al. 2021), but instead highlight the important role of cloud-related diabatic processes in modulating upstream wave activity within the LWA framework (see our discussion **L407-421**).

6. I suggest that the authors swap around Sections 3.1 and 3.2, to first show the mean states of the idealized simulations before discussing LWA results.

Authors: We appreciate the reviewer's suggestion. However, we believe the current structure, beginning with the LWA analysis in Section 3.1, is more appropriate for the scientific narrative we aim to present. Blocking formation is not solely governed by changes in the mean state, but also strongly influenced by the evolution of eddy (wave) activity and its interaction with the background flow during the lifecycle of the event (e.g., Nakamura and Wallace 1993; Shutts 1983; Hoskins et al. 1983).

Therefore, we first focus on the lifecycle of wave activity and the physical processes that contribute to blocking onset and development. This process-oriented approach allows us to identify where and how wave activity is generated and modulated. We then examine the background environmental conditions in **Section 3.2**, which provides context for how the large-scale mean state can either support or suppress eddy activity. This order facilitates a more logical progression in terms of physical mechanisms.

7. I would appreciate it if the authors can list limitations of the current study, including the point that I raised in my first comment, in the conclusion.

Authors: We have added a new paragraph in the discussion (L438-L444) to discuss the limitations of the current study, including the point raised in your first comment, and to provide recommendations for future research (L444-447). Thank you.

8. Fig 1b: interesting tail of negative signature extending all the way to off the east coast of North

America. If you extend the area to a broader area, would it extend to the proximity of Gulf Stream? (related to my first major comment)

Authors: A weak negative response is seen near the Gulf Stream region (see Figure R10 below for CLOCK); however, since SST gradients are held constant across all experiments, this signal cannot be attributed to changes in SST-driven forcing. Instead, it reflects differences in the impact of CREs on circulation and eddy activity as we discussed in our results. We have now added a new paragraph in the revised manuscript (L438-444) to highlight the limitation of using fixed SST boundary condition and to recommend future work to address this. Thank you.

Fig R10. Difference in wintertime (DJF) blocking frequency between the CLOCK and CTL (shading) and the climatology in the CTL run (contour lines).

Other small comments:

- L80 “climate feedback modes” -> “climate modes”?

Corrected.

- Fig S1: Is this a case study, or composite?

Composite. We have revised the caption to clarify this.

- L.130: Which major atmospheric heat source are the authors speaking about? Please be more specific.

In the LWOFF simulation (similar to COOKIE-type 2), the setup removes longwave cloud radiative effects by making clouds transparent to longwave radiation only. That means clouds no longer contribute to longwave radiative heating, which is a major atmospheric heat source, particularly in the mid-to-upper troposphere (see L132-L135).

- L.154-156: Again, please elaborate more, on what bases the authors come up with these two hypotheses, and what they exactly mean by them.

The two hypotheses are based on physical reasoning. First, diabatic heating can act as a direct source of wave activity, meaning that variations in heating (e.g., from clouds or latent processes) can influence blocking formation by enhancing or suppressing LWA. Second, mean cloud radiative effects (CRE) can modify the background mean state, such as the jet structure or temperature gradients, which in turn affects the environment in which blocks form. We have added several sentences to clarify this point (see Lines 153–157).

- Fig 2 caption: Please properly describe what the box-whisker plots in (b) and (c) show. Are they showing interannual variability, or ensemble spread? Please explain. They are clearly different from (a), so please remove “(b) As in (a)”. Also, the circles over the box-whisker plots are misleading, because they are normally used to indicate the presence of anomalies.

Caption corrected. Open and filled circles have been repositioned to the center to indicate statistically significant or insignificant differences between the two setups.

- L.174 “closely resembles”: I would lessen the tone, as there are visible differences, including that there is no clear low-pressure system on the west of blocking in reanalysis.

Revised. “.. *basically captures the observed blocking frequency distribution*” (L181).

- L.176 “stronger deceleration of the zonal flow”: Is this shown anywhere?

It is expected but not shown, as dA/dt is inversely proportional to the zonal wind (Nakamura and Huang 2018; Huang and Nakamura 2017).

- Fig S3: It reads lag-3 – Lag0 but are they also supposed to read “days -3” – “days 0”?

Corrected.

- L. 192 “congestion of a traffic jam”: Tautology. Please rewrite.

Revised (L197-198) “Specifically, *when the LWA zonal flux exceeds a capacity of the jet stream, blocking would manifest like a traffic congestion*”

- Fig S4: Please change the contours of all the panels to geopotential height anomalies, so it is easier to compare against Fig S2, to see what the authors are trying to show the readers in l.210.

We prefer to retain the current contours, as they are chosen to highlight specific aspects of the analysis. In the first column (**Figs. S4a-d**), our intent is to show OLR anomalies in the upstream region of enhanced LWA (contours) (**L216-218**). In the second column (**Figs. S4e-h**), the goal is to illustrate that enhanced column water vapor upstream of the blocking region lies within the WCB zone, located between low- and high-pressure systems, as described in the text (**L213-214**). Changing the contours to geopotential height anomalies would not support these specific points as effectively.

- Fig S5: “DTCOND” does not match with the description in the caption. Please make them consistent.

Corrected. It has now been replaced with LH.

- L.222 “the interactive CRE (CLOCK) and changes in the mean LWCRE (LWOFF)”: The descriptions of what you want to illustrate and the corresponding simulation designs are not quite matching, or more like the opposite. Please rewrite.

Thank you. It has been corrected.

- L.224: its budget “of all three simulations”

Corrected (L232).

- Figs 4, 5: Stippling is not quite visible. I suggest that the authors change the color range to something like [-5 to 5], [-20, 20] respectively.

Figures 4 and 5 have been updated. We adjusted the color bar ranges to improve visibility and increased the density of the stippling to make it more discernible.

- Fig 4: Please indicate the 10deg x 10deg area used to construct (d) on each map.

Corrected.

- Figs 4 & 5: please adequately explain what exactly the error bars indicate.

Corrected. The error bars represent ± 1 standard deviation of LWA budget term values within the defined polygon.

- Fig 5: Are the encircled regions also 10deg x 10deg?

The encircled regions indicate the upstream and downstream areas relative to the block. We have now added their coordinates in the caption of Fig. 5 for clarity.

- Fig S6: “ $\Delta\Sigma_{tot}$ ” does not match with the description in the caption.

Corrected.

- L.271 “cloud-radiative impact might operate”: please elaborate more on what you mean by this.

We removed this sentence.

- L.274: I don’t understand the sentence. Is it “from LWA”? Are you speaking about LWOFF simulation? Please clarify.

The sentence has been clarified (L292-295).

- L.276 “increased nighttime cooling”: Do you mean radiative cooling of the surface?

The sentence has been removed, and the paragraph has been revised based on new mechanisms (L309-L332).

- L.279 “discused” -> “discussed”

Corrected.

- L.284 “larger longwave radiative heating”: By which physical mechanisms? Please explain.

The sentence has been removed, and the paragraph has been revised based on new mechanisms (L309-L332).

- L.310 “no change in the mean state”: Just above this sentence, the authors described that there are some differences in the mean state.

As also suggested by the reviewer #1, we have revised this and tone down statements that overstate “the absence of significant change in the mean state for CLOCK” throughout the manuscript.

- L.312 “This is consistent with our previous analysis”: please specify which paper/results the authors are referring to.

We have revised this sentence (see L350-L351).

- L.317 Please explain how exactly turning off LWCRE change the meridional temperature gradient.

This effect has been demonstrated in previous studies (e.g., Li et al., 2015; Voigt et al., 2021) and in our recent paper (Lu et al., 2024). The zonal mean CRE difference between CLOCK minus CTL (in Lu et al., 2024) or CLOUD OFF minus CLOUD ON (in Li et al., 2015) is dominated by longwave (LW) radiation, featuring a positive heating in the upper troposphere and a cooling in the lower stratosphere, which, owing to the poleward sloping tropopause, can give rise to an enhanced differential heating between the tropics and the extratropics. We cite these works to avoid repeating the established mechanisms (see L350-355).

- Fig 7: Indicate what contours of (c)-(f) are, and also the contour intervals for all the panels

We have revised the caption see now in Fig. 8.

- L.323 “Eady’s growth rate” -> “Eady growth rate”

Corrected.

- L.442 “blocking-based anomalies index” -> “anomaly-based blocking index”

Corrected.

- L.463: the definition of q is already written

Deleted.

- L.465: H=7km is already written

Deleted.

- L.468: “the meridional displacement of the wavy potential vorticity contour relative to ϕ ”
We retain this phrasing, as it refers solely to the definition of the integral bounds, following the formulation in Nakamura and Huang (2018).

- L.485: reference for falwa is missing?
Corrected.

- L.488 “boundary conditions listed in Neal et al.”: please specify.
The treatment of the boundary conditions in Neal et al. (2021) is very detailed, and it would be impractical to outline all aspects here. For clarification, we have revised the sentence to refer to Eqs. 14-16 in the Supplementary Information of Neal et al. (2021).

- L.489 “Eddy Growth Rate” -> Eady Growth Rate
Corrected.

Reviewer #1 (Remarks to the Author):

I appreciate the authors' effort in addressing my previous comments and revising the manuscript accordingly, which I believe has substantially improved the manuscript. In their revision, the authors now propose two hypotheses to causally link CREs to dynamical processes of blocking formation: 1) increasing diabatic heating sources and 2) modifying the background mean state (L155-163). To test these hypotheses, partly in response to the reviewers' previous comments, they have included new results on the vertical structure of diabatic heating, cloud-related variables, and moisture transport. I find all of them relevant and supportive of their proposed mechanisms. The authors also newly streamlined these mechanisms at the end of the section (L308-330), which adds clarity to understanding how interactive CREs influence blocking frequency.

While I enjoyed reading the authors' detailed response to my previous comments, I have one follow-up comment, which is simply to share my thoughts, along with a minor suggestion on the updated results. I have listed a few other minor suggestions for clarifications.

Authors: We appreciate the reviewer's careful and insightful suggestions. Below, we provide a point-by-point response to each comment, highlighted in blue.

Minor Comments:

- Lines 308-330: The revised manuscript now suggests two (at least) possible pathways based on their original and new findings. They are clearly described and sound convincing for the most part. In particular, I find that the first mechanism provides a clear explanation dynamically linking LWCRE to changes in LH. Meanwhile, I would also like to point out that the possibility of the second mechanism, related to high cloud cover and evaporation feedback, has been explored only by MERRA2, not by model experiments. I assume the reason model results are not shown for the same diagnostics in the study is either that the variables relevant to the diagnostics are not available from the current model configuration, or that model results may differ substantially from the MERRA2 results, given that cloud fraction and surface properties are quite challenging to represent accurately in models. In any case, it is less clear whether the second mechanism similarly operates in the CTL simulation. The results from Figs. 7e and 7f also raise the possibility that the second mechanism may not be well captured by the model. Specifically, the positive LWCRE anomaly in MERRA2 is located vertically from 800 hPa to 400 hPa with its peak at 500 hPa, whereas that in CTL is found between 850-550hPa (L304) with its peak at ~800 hPa. This vertical difference may indicate that cloud cover fraction at different altitudes in the model deviates from MERRA2 quite noticeably. If this surface warming through high cloud cover anomaly is not explicitly shown by model experiments, then I believe the suggested causality from MERRA2 is not fully demonstrated as well (as surface warming itself can be induced by many other factors). With that said, I feel this would be a minor point relative to the key arguments of the study. The authors have already provided sufficient new and interesting results, and I do not think it is essential for this study to ascertain whether the proposed second mechanism holds in model experiments. Nevertheless, it would be appreciated if the authors could clarify that the second

‘possible’ pathway proposed in this study is based on reanalysis results, and that further investigation in a separate modeling study would be helpful.

Authors: We appreciate the reviewer’s thoughtful comment. Regarding the second proposed possible pathway, how LWCREs may feed back onto LH, we agree that further investigation in a separate modeling study is warranted. This is primarily because: **(1)** Our model did not output key surface diagnostic variables (such as dew point temperature, near-surface wind, etc.) needed to further investigate the driver of LHF changes linked to CRE from a thermodynamic perspective. **(2)** The model simulations were conducted with fixed SST, which limits this surface feedback. **(3)** As the reviewer mentioned above, there is a noticeable difference in the vertical structure of the LWCRE anomaly between MERRA2 and the model, which may reflect biases in high cloud fraction and associated surface warming.

Nonetheless, using reanalysis data, we further investigated the thermodynamic aspects of the second possible pathway by applying the bulk aerodynamic formula, as suggested by Reviewer #3 (see **Fig. R1 below**). Regions of positive latent heat flux (LHF) anomalies partially coincide with positive anomalies of LWCRE, latent heating, surface warming, and high cloud cover in the upstream region of the block (**Figs. R1a-f**). Positive LHF indicates increased moisture transfer from the surface to the atmosphere, supporting enhanced latent heating aloft. This increase in LHF (**Fig. R1e**) is primarily driven by stronger surface wind and enhanced near-surface vapor gradient ($q_s - q_a$) upstream of the block region (**Figs. R1g-h**). The latter can be associated with surface warming linked to CRE during the blocking onset (**Fig. R1f**), which increases the saturated specific humidity (q_s) and hence the moisture gradient (Fig. S11g). Removing this effect would suppress LHF and consequently weaken LH, consistent with previous studies (Keshtgar et al., 2023). While these reanalysis-based diagnostics support the plausibility of this pathway, they cannot fully isolate the role of LWCRE from other contributing processes. Currently our model simulations were conducted with prescribed SSTs, which limits this surface feedback, and the necessary surface diagnostic variables were not archived to directly evaluate this pathway. Therefore, future dedicated modeling studies, particularly those using interactive SSTs, are needed to robustly evaluate this mechanism.

We have revised the paragraph discussing this second “possible” pathway to present it as a plausible hypothesis rather than a firmly established mechanism (see **L332-350**).

- Magnitude of vectors: Information on the magnitude of vectors is missing throughout the manuscript. I recommend including reference vectors in the figures or providing their magnitude information in the figure captions.

Authors: Added.

- Lines 156-157 ‘changes in the large-scale mean state, ..., the strength of stationary waves, influence’-
> ‘and the strength of stationary waves’? I recommend revisiting this sentence.

Authors: Corrected (L161-L163). “Additionally, changes in the large-scale mean state, such as the location and structure of the jet and the strength of stationary waves, can influence the conditions that favor blocking development”.

- Lines 445-446 ‘Therefore, improving ... could therefore potentially ...’. I suggest removing one of ‘therefore’ in the sentence.

Authors: Corrected.

- Vectors in the second row of Fig. 8: I found the black vectors denoting the Plumb flux appear somewhat smaller compared to the vectors in other figures. Please consider thickening them or changing their color.

Authors: Corrected (see Fig. 8).

Fig R1. Blocking-relative composites of cloud fraction, radiation, and surface thermodynamic variables averaged from days -6 to -1 before the mature stage of the block from MERRA2. (see new Fig. S11 in the SI).

Reviewer #2 (Remarks to the Author):

Cloud-Radiative Effects Significantly Increase Wintertime Atmospheric Blocking in the Euro-Atlantic Sector (Lubis et al., 2025). This is my second review of the study. I appreciate the authors' efforts in addressing the previous comments, and the manuscript has really improved following the major revisions. I have only a few minor comments. Addressing these comments, I believe the paper is in good shape to proceed.

Authors: We gratefully acknowledge the reviewer's careful review. Below, we provide a point-by-point response to each comment, highlighted in blue.

Minor comments:

Lines 18-19 (also Lines 393-394): The sentence suggests that the main way in which CRE affects the diabatic source of wave activity is through direct impact ('primarily', CRE heating), while the indirect pathway through changes in latent heating is secondary. In fact, this is the opposite. It would be better to say, for example, that CRE affects the diabatic source of wave activity both directly and indirectly, with the indirect pathway playing a significant role.

Authors: The sentence has been revised (see **L18-19**). *“the interactive CREs increase LWA by strengthening the diabatic source of wave activity upstream of the block, both directly through longwave cloud radiative heating and indirectly via their feedback on latent heating, with the latter playing a dominant role.”*

Line 86: ‘... its impact on simulated variability,’

The term simulated variability here is unclear, -> changing to ‘its impact on blocking formation’

Authors: Corrected (see **L88**).

Line 107: Here, it would be better to refer to the limitations of the experiments discussed later in the method, as Fig. S1 illustrates that notably fewer blocking events are simulated with the E3SM CTL. Is this related to the missing lower-boundary SST-related forcing in E3SM simulations?

Authors: Yes, we have now discussed this potential limitation (see **L107-L109**): *“The relatively low blocking frequency in CTL may result from the absence of interactive lower-boundary SST forcing, which is known to modulate Euro-Atlantic blocking formation.”*

Figures 5, 6, and S6: as the diabatic terms shown in Fig. 6a do not add up to the total diabatic term shown in Fig. 5d, it seems a large contribution comes from clear-sky radiative heating (according to equation 13), right? And if yes, this term does not change a lot between CTL, LWOFF and CLOCK? In other words, the major changes in diabatic source wave activity are due to changes in diabatic heating and specifically the latent heating?

Authors: As shown in Eq. (13), **Fig. 5d** (\dot{A}) includes residual contributions (e.g., dissipation, surface damping, and budget errors) beyond the diagnosed terms in **Fig. 6a**. On the other hand, the

$\Delta\dot{\sigma}$ in Fig. S6a is primarily explained by the sum of terms in Fig. 6a, with latent heating providing the largest contribution. Yes, we found that the clear-sky radiative heating term changes little across CTL, LWOFF, and CLOCK, confirming that the main differences in the diabatic source of wave activity arise from changes in latent heating. This point is now clarified in the manuscript (e.g., see L18-19, L290-291, L414-416).

Line 279: From Fig. 7 (cf. panels a and b), it seems that the limitation mentioned is due to differences in latent heating between MERRA and CTL at lower levels rather than to low-cloud radiative processes. At least, the spatial variability of cloud radiative heating is comparable between MERRA and CTL (panels e and f), but not latent heating. This makes me wonder about latent heating at lower levels by MERRA that is absent in CTL. What could be the reason?

Authors: We have checked the low-level cloud (LCC) fraction during the blocking lifecycle in CTL and found that LCC is underestimated compared to MERRA, especially upstream and near the blocking center. This underestimation likely leads to reduced low-level condensation and is consistent with weaker latent heating in CTL (Fig. 7b). We have clarified this in the text (L286) “...which may stem from the model’s limitation in representing low-level clouds and the associated latent heating upstream during the blocking onset”.

Line 320: I believe that reference 41 fits better here, as it also shows that the absence of CRE cools the atmosphere and increases static stability within ascending regions of cyclones. However, it also highlights the significant role of LW cooling by low-level clouds in reducing static stability and weakening cyclones, similar to reference 38.

Authors: Yes, we have now included Ref. 41.

Line 328 References 37 and 54 show that CRE (warming) results in reduced latent heating (and precipitation) within the tropics. Reference 37 shows that CRE in the mid-latitudes increases precipitation (therefore LH), but this was attributed to enhanced eddies and vertical motion (similar to the first proposed pathway). From the ref 37: ‘the heating of the tropical atmosphere by ACRE (Fig. 4c) is primarily balanced by a reduction in the latent heating, consistent with the reduction in tropical precipitation evident in the clouds-on simulation (Fig. 8).’ Therefore, referencing these studies does not seem correct. A similar reference might be 40, which shows that CRE enhances cloud microphysical heating, although it does not show changes in surface evaporation.

Authors: We have now corrected the references.

Lines 414-416: I don’t think that the identified and proposed mechanism of CRE impact is ‘fundamentally’ different. Results show that CRE (directly and indirectly) affect upstream diabatic heating (already later in line 418, the importance of upstream diabatic heating for blocking is mentioned), and changes in LH will translate into changes in PV at upper-levels. (like ref 40 that shows changes in LH by CRE lead to changes in large-scale winds and PV at upper levels and dominate the changes in the tropopause wave that, as also said in the preprint here, is important for blocking

formation, e.g., ref 31). The results presented are additive and complementary to some established ideas. The point is that the indirect impact of CRE has a higher weight than the direct impact of CRE heating.

Authors: We thank the reviewer for the thoughtful comment. Rather than implying a fundamentally different mechanism, our intention is to emphasize how CREs modulate the upstream diabatic source of wave activity through both direct longwave radiative heating and, more significantly, through their feedback on LH. While the traditional PV tendency framework (refs. 31–35) explains blocking onset as the generation and amplification of negative PV anomalies aloft in response to diabatic heating, the LWA framework characterizes this process in terms of enhanced upstream diabatic LWA generation, driven by CRE and its feedback on latent heating, which results in stronger downstream convergence of horizontal LWA flux and increased LWA accumulation (thus to form a block see **Fig. R2 below**). Therefore, our findings offer a complementary perspective to earlier PV-tendency-based interpretations, while emphasizing the dominant role of the indirect CRE-LH pathway in amplifying the upstream diabatic source of wave activity critical for blocking onset. We have modified this discussion (**L433-L444**).

Line 480: Table S1 -> Table 1

Authors: Corrected.

Line 487: Table S2 -> Table 1

Authors: Corrected.

Line 496: cloud heating rates, all-sky rates -> cloud radiative heating, all-sky radiative heating, clear-sky radiative heating

Authors: Corrected.

Reviewer #3 (Remarks to the Author):

I appreciate the authors' extensive additional analyses and their point-by-point response to my previous comments. I think the manuscript has improved since the previous round, particularly with the inclusion of diabatic decomposition and cloud cover diagnostics. However, I find the physical interpretation of the diagnosed effects still lacks dynamical depth and clarity, and important diagnostics that support the proposed mechanisms remain underutilized. Below I outline my major concerns in detail:

Authors: We thank the reviewer for the thoughtful comments. Below, we provide a point-by-point response to each comment, highlighted in blue.

Major Comments:

1. Dynamical explanation

While the manuscript clearly establishes that CREs (especially LWCRE) enhance blocking by increasing upstream wave activity, I find the explanation of the physical mechanisms still remains largely descriptive. The text alludes to CRE-induced enhancements in latent heating and high cloud cover, but does not fully explain the dynamical pathway from cloud radiative forcing to blocking onset. The authors refer to Supplementary Figure S12 showing increased high cloud cover upstream of blocks. However, this key diagnostic is not meaningfully integrated into the main text or used to support the proposed feedback between LWCRE and latent heating. This weakens the causal argument. To improve clarity and support the proposed mechanism, I recommend:

- Bringing the high-cloud fraction results into the main text

Authors: We appreciate this thoughtful comment. In our manuscript, we have provided several explanations to causally link CREs to dynamical processes of blocking formation: **(A)** by increasing diabatic heating sources upstream and hence, enhancing the downstream convergence of LWA flux (**L316-L350**), and **(B)** by modifying the background mean state (Section 3.2, **L357-395**).

(A) Increasing diabatic heating sources upstream

Using the finite-amplitude LWA diagnostic, we showed clearly that during the onset period, in the upstream region where clouds dominate within ascending WCB airstreams, enhanced CREs strengthen the diabatic wave source upstream both directly through longwave CRE and indirectly via their feedback on latent heating, with the latter playing a dominant role. We further provide two possible pathways to explain **how CREs affect LH:**

(1) First pathway: CREs strengthen poleward and upward moisture transport in the WCB upstream of the block (see **L316-331**).

“The reduction of LH in CLOCK and LWOFF in the absence of CREs can be associated with the lack of LWCRE impacts on large-scale ascent and moisture transport. During the onset period, LWCRE significantly warms the mid-troposphere (850-450 hPa) in the upstream region in CTL (Fig. 7f), whereas this warming is markedly weaker in CLOCK (Fig. 7g) and absent in LWOFF (Fig. 7h). Without LWCRE-induced warming, cooler mid-tropospheric temperatures steepen the vertical temperature gradient, enhancing static stability and weakening large-scale ascent. This weaker ascent is consistent with reduced negative vertical velocity anomalies in CLOCK (Fig. 7i) and LWOFF (Fig. 7j) in regions of enhanced LH (see green box), indicating less efficient lifting of moist air into the mid-troposphere. This reduction in ascent is accompanied by weaker vertically integrated moisture transport (IVT) into the WCB (Figs. 7k-l and Fig. S10), confirming that less moisture is being advected poleward upstream of the block. The weakening of moisture transport in both CLOCK and LWOFF is consistent with the overall reduction in climatological moisture flux upstream (over the eastern North Atlantic) in these experiments (see later in Figs. 8g–h). As a result, condensation is suppressed, leading to a decline in LH. This is consistent with previous studies showing that LWCRE plays an important role in intensifying vertical motion and LH (Refs 37, 40, 41, 54).”

(2) Second “possible” pathway: LWCRE enhances LH through evaporation feedback (see **L332-350**).

Using observational-based MERRA2 data, we showed that regions of positive latent heat flux (LHF) anomalies partially coincide with positive anomalies of LWCRE, latent heating, surface warming, and **high cloud cover** in the upstream region of the block (**Figs. S11**). Positive LHF indicates increased moisture transport from the surface to the atmosphere, supporting enhanced latent heating aloft. While these MERRA2-based diagnostics support the plausibility of this pathway, they cannot fully isolate the role of LWCRE from other contributing processes. Currently our model simulations were conducted with prescribed SSTs, which limit this surface feedback, and the necessary surface diagnostic variables were not archived to directly evaluate this pathway. Therefore, further investigation in a separate modeling study, particularly those using interactive SSTs, is needed to robustly evaluate this pathway. For this reason, we have chosen not to include the high cloud fraction results in the main text but instead retain them in the Supplementary Materials as supporting evidence from observation (MERRA2) and leave the analysis of high cloud fraction for future investigation. **As suggested by Reviewer #1, we have revised the paragraph discussing this second “possible” pathway to present it as a plausible hypothesis rather than a firmly established mechanism (see L332-350).**

(B) By modifying the background mean state (L357-L395).

We clearly show that removing mean LWCRE causes a significant poleward shift of the jet, weakens stationary high-pressure ridges over Europe, reduces low-level baroclinicity, and mitigates poleward moisture transport upstream. These changes create unfavorable conditions for blocking formation, hence consistent with reduced block in CLOCK and LWOFF simulations. This is consistent with

previous studies (e.g., Barnes et al., 2010; Nakamura et al., 2018), which show that such background state conditions are unfavorable for blocking formation.

In summary, our physical mechanisms linking CREs to blocking dynamics are firmly supported by dynamical evidence, and the interpretation extends beyond a purely descriptive explanation.

- Interpreting results within the warm conveyor belt (WCB) framework, which provides a well-established basis for linking moist ascent, diabatic PV generation, and upper-level ridge formation (e.g., Grams et al., 2011; Methven, 2015).

Authors: We acknowledge the value of the WCB framework in linking moist ascent, diabatic PV generation, and upper-level ridge formation, as highlighted by the reviewer. However, in this study, we chose to analyze blocking onset using the local wave activity (LWA) diagnostic, following the theoretical formulation of Nakamura and Huang (2018) applicable for blocking dynamics/onset. This approach allows us to quantify the finite-amplitude and propagation of Rossby wave activity that causes a block formation and its modulation by diabatic processes. Our interpretation, therefore, centers on the role of enhanced diabatic wave activity generation, particularly within the WCB region of upstream, in driving downstream LWA flux convergence and triggering blocking anticyclones (cf. Neal et al., 2022). Our approach complements the PV-based approach (Grams et al. 2011; Methven 2015), which itself is quite involved. We have carefully discussed the results, emphasizing the connection between the CREs (and their feedback on LH) within the WCB region and the enhanced upstream diabatic source of wave activity, as well as its implication for downstream LWA flux convergence that favors block formation (see e.g., L219-L233, L258-263, L316-L331, L414-L420).

Given that we have clearly established the dynamical connections between CREs and the block formation using the LWA diagnostic, we refrain from adding another parallel analysis using the WCB framework. Although a comparison of the two complementary frameworks is of value, it distracts from the focus of this study, and it is more suitable for a more broadly framed methodology-focused work in the future.

In addition, the introduction (lines 70-82) would benefit from a more detailed discussion of prior work on how physically CREs influence large-scale dynamics, to better motivate the study and help interpret later figures, the former of which currently reads rather opportunistic.

Authors: We have expanded the introduction (L75-L84) to include the dynamical mechanism proposed by previous studies explaining how CREs influence large-scale circulation. Previous studies have shown that CREs can modulate baroclinicity by altering meridional temperature gradients and static stability, thereby affecting eddy activity and the large-scale flow (e.g., Li et al., 2015; Grise et al., 2019; Voigt et al., 2021; Lu et al., 2024). In addition, Baumgart et al. (2019) demonstrated that CREs influence forecast error growth in WCB regions and downstream Rossby wave activity by modifying the location, intensity, and vertical structure of latent heating -- key factors for diabatic PV

generation (L81- L82). Despite this, the role of CREs in the formation of atmospheric blocking remains unexplored. We discussed this dynamical mechanism in our manuscript (e.g., L372-374, L429-L449).

Finally, the discussion around lines 297–307 would be significantly strengthened by explicitly incorporating: 1. PV-based reasoning, how diabatic heating generates negative PV anomalies aloft and strengthens ridges (Haynes and McIntyre, 1987). 2. Low-PV air transport via WCBs, which links latent heating with anticyclonic upper-level anomalies (e.g., Pfahl et al., 2015; Steinfeld and Pfahl, 2019). Bringing these concepts into the discussion would ground the proposed mechanism in current dynamical theory and reinforce the CRE-blocking connection.

Authors: As we discussed above, we chose to analyze the dynamics of blocking onset using the local finite-amplitude wave activity (LWA) diagnostic, following the theoretical framework of Nakamura and Huang (2018). Although our approach is based on wave activity, it remains fundamentally rooted in potential vorticity (PV) dynamics. The PV-tendency-based and LWA-based frameworks are complementary to each other. The PV budget tracks the local generation and advection of PV anomalies, while the LWA framework quantifies the finite-amplitude displacement of PV contours from a zonally symmetric reference state, thereby capturing finite-amplitude Rossby wave activity and its modulation by adiabatic and diabatic forcing associated with blocking formation.

From a theoretical perspective (see the details in Huang and Nakamura 2016; Nakamura and Huang, 2018), the LWA budget provides a finite-amplitude reformulation of the PV conservation law, where wave activity is defined by the displacement of PV contours relative to a reference state, and the fluxes are derived from the advective transport of this displaced PV field (Huang and Nakamura, 2016). **Thus, while LWA budget does not track the sign of PV anomalies directly (e.g., negative PV anomalies generated by diabatic heating), it reflects their integrated effect on wave amplitude (see Fig. R2 below).** As shown in Fig. R2, the LWA anomalies clearly follow the development of the negative PV anomalies aloft and the strengthening of the upper-level ridge associated with a block. Thus, this diagnostic is as reliable as the traditional PV-tendency budget in explaining blocking onset (i.e., upper-level negative PV anomalies), but offers a different perspective, focusing on the development of wave activity aggregation rather than the evolution of PV itself.

Through the lens of the LWA, **the enhanced diabatic heating associated with CREs -- through its impact on PV-- projects onto the LWA framework as an enhanced diabatic wave activity generation/source within the WCB region of an upstream block, which in turn leads to stronger downstream convergence of horizontal LWA flux and, consequently, promotes blocking formation (see Fig R3 below, cf. Nakamura and Huang 2018).** This mechanism complements PV-based interpretations that emphasize diabatic generation of negative PV anomalies due to low-PV air transport within WCBs (e.g., Haynes and McIntyre, 1987; Pfahl et al., 2015; Steinfeld and Pfahl, 2019). However, it is framed here through the lens of finite-amplitude wave activity, which explains the strengthening of finite-amplitude Rossby wave activity leading to blocking formation via CRE-induced upstream diabatic wave generation that enhances downstream convergence of LWA flux. See our discussion section (L433-449, cf. Neal et al., 2021).

Fig R2. Blocking-relative composite lifecycle of (a-c) 300-hPa PV anomalies (shading) and (d-f) 300-hPa LWA anomalies (shading) from MERRA. Contours show Z500 anomalies. **The evolution of upper-level LWA anomalies closely follows the development of the blocking pattern and the associated upper-level negative PV anomalies, illustrating how the finite-amplitude LWA framework provides a complementary perspective on blocking onset.** Note that the column LWA and LWA budgets are dominated by upper-level features since the wave amplitude is largest in the upper troposphere (see Zhang et al. 2025 and Huang and Nakamura 2017).

Fig R3. Relationship between the upstream diabatic source of LWA due to LH and CRE prior to onset (days -3 to -1) and the downstream convergence of horizontal LWA flux at lag 0. This figure is based on values shown in Fig. 6a

and Fig. 5e of the manuscript. **The results highlight that enhanced diabatic wave activity generation within the WCB region of an upstream block leads to stronger downstream convergence of horizontal LWA flux that forms the block.**

2. Mischaracterization of LWA sources vs. redistribution

The manuscript attributes blocking formation primarily to the convergence of horizontal LWA flux. However, this interpretation risks conflating LWA redistribution with LWA generation. According to previous works (e.g., Nakamura and Huang, 2018; Yamamoto and Martineau, 2024), horizontal fluxes only redistribute LWA—they do not generate or destroy it. The true sources of wave activity are:

- The non-conservative term $\langle \dot{A} \rangle \cos \phi$, which includes diabatic heating, PV mixing, and numerical diffusion
- The low-level eddy heat flux, reflecting baroclinic generation.

Your Fig. 3 clearly shows both terms contributing positively upstream of the block, consistent with the findings of Yamamoto and Martineau (2024) for the Atlantic region. However, the manuscript text does not clearly distinguish these processes from flux convergence. I recommend reframing the interpretation to emphasize that CREs enhance wave activity generation via diabatic and baroclinic processes, which is then redistributed downstream by horizontal fluxes in lines 188-210.

Authors: We thank the reviewer for prompting us to more clearly distinguish between the generation and redistribution of LWA. In our manuscript, we distinguish these roles explicitly (see **L175-L178**). As stated in the manuscript, there are two sources of wave activity generation: (1) diabatic processes, represented by the non-conservative term $\langle \dot{A} \rangle \cos \phi$, and (2) low-level eddy heat flux, which reflects baroclinic (adiabatic) generation (see **L175-L176**). In contrast, the convergence of horizontal LWA flux reflects the redistribution of wave activity, not its generation (see **L178**).

As shown in Figs. 3(i-l) and Figs. 5(d-e), the contribution from the low-level eddy heat flux, representing baroclinic generation, is **relatively small and secondary** (see our **L198-L199, L236-L237**). This indicates that the dominant wave activity generation primarily arises from **diabatic** processes (consistent with previous studies e.g., Neal et al., 2022; Wang et al., 2021). This enhanced upstream diabatic wave generation, in turn, leads to stronger horizontal convergence of LWA flux, contributing to blocking formation. We have ensured that this interpretation, distinguishing generation from redistribution, is consistently conveyed throughout the manuscript (see **L234-237, L274-276, L351-L356, L414-420**).

3. Evaporation Feedback

In lines 327–331, the manuscript states that LWCRE-induced surface warming increases evaporation, which then amplifies latent heating. While plausible, I am afraid that this is oversimplified. According to the bulk formula, latent heat flux depends on the moisture gradient between surface and near-surface air, wind speed, and air density. Thus, an increase in surface temperature does not guarantee increased

evaporation if near-surface air also becomes more humid. I suggest the authors revise this explanation to clarify under what thermodynamic conditions the proposed feedback would hold.

Authors: We appreciate the reviewer’s thoughtful comment. Regarding the second proposed possible pathway, how LWCREs affect LH via evaporation feedback, we agree that further investigation in a separate modeling study is warranted. This is primarily because: **(1)** Our model did not output key surface diagnostic variables (such as dew point temperature, near-surface wind, etc.) needed to further investigate the driver of LHF changes linked to CRE from a thermodynamic perspective. **(2)** The model simulations were conducted with fixed SST, which limits this surface feedback. **(3)** There is a noticeable difference in the vertical structure of the LWCRE anomaly between MERRA2 and the model, which may reflect biases in high cloud fraction and associated surface warming.

Nonetheless, using reanalysis data, we further investigated the thermodynamic aspects of the second possible pathway by applying the bulk aerodynamic formula (see **Fig. R1 above**). Regions of positive latent heat flux (LHF) anomalies partially coincide with positive anomalies of surface evaporation, LWCRE, latent heating, surface warming, and high cloud cover in the upstream region of the block (**Figs. R1a-f above**). Positive LHF indicates increased moisture transfer from the surface to the atmosphere, supporting enhanced latent heating aloft. This increase in LHF (**Fig. R1 above**) is primarily driven by stronger surface wind and enhanced near-surface vapor gradient ($q_s - q_a$) upstream of the block region (**Figs. R1g-h above**). The latter can be associated with surface warming linked to CRE during the blocking onset (**Fig. R1f above**), which increases the saturated specific humidity (q_s) and hence the moisture gradient (Fig. S11g). Removing this effect would suppress LHF and consequently weaken LH, consistent with previous studies (Keshtgar et al., 2023). While these reanalysis-based diagnostics support the plausibility of this pathway, they cannot fully isolate the role of LWCRE from other contributing processes. Currently our model simulations were conducted with prescribed SSTs, which limits this surface feedback, and the necessary surface diagnostic variables were not archived to directly evaluate this pathway. Therefore, future dedicated modeling studies, particularly those using interactive SSTs, are needed to robustly evaluate this mechanism.

We have rephrased this discussion to present it as a plausible hypothesis rather than a firmly established mechanism (see **L332-350**), as suggested by reviewer #1.

Minor Comments:

- I have mentioned this for the first round, but I still think that the description of COOKIE/COOKIE2 are insufficient. For the sake of reproducibility, please clarify what they are in the main text.

Authors: We updated this CFMIP description (see **L506-513**).

“Finally, we also compare the LWOFF experiment to the COOKIE-2 simulations of the Cloud-Feedback Model Intercomparison Project (CFMIP) Tier 2 in two different configurations: standard Earth-like configurations (AMIP) with interactive CREs and AMIP with disabled LWCRE (AMIP–LWOFF). The COOKIE-2 experimental protocol specifies that only the LWCRE is deactivated, while

SSTs and sea-ice concentrations are prescribed and held identical between the control run and LWCRE-off simulations (Webb et al., 2017). We analyze daily Z500 fields from these simulations at a spatial resolution of 2.5°x 2.5°. These simulations have been conducted by four modeling groups so far and are listed in Table 1.”

- L.96 underscores -> underscore
Corrected.

Reviewer #3 (Remarks to the Author):

I appreciate the authors' effort in expanding the analysis and responding to earlier comments. The manuscript is technically improved, but two major issues of physical interpretation remain unresolved.

Authors: We thank the reviewer for the thoughtful comments. Below, we provide a point-by-point response to each comment, highlighted in blue.

This is the third time I raise this, and I will be more explicit here. I fully understand that this study uses the LWA framework to diagnose the impact of CREs on blocking. I am also aware of the formal dynamical basis of that framework, and its usefulness. My concern is not about methodology—but about the lack of physical explanation consistent with established dynamical understanding. The authors use LWA to quantify the contribution of diabatic processes to midlatitude wave activity and blocking. However, the manuscript never explains why those processes—particularly latent heating and LWCRE—lead to enhanced wave activity. For example, to me Figure 7 clearly shows the signature of WCB-type ascent—vertical and poleward transport of low-PV air—which amplifies the upper-level ridge. This is a well-established mechanism, yet the manuscript avoids acknowledging it. Similarly, the authors attribute increased waviness to LWCRE but offer no physical mechanism—only a diagnostic association. If the claim is that CREs dynamically influence blocking, then the physical pathway must be explained in terms of PV-based thinking (e.g., via diabatic PV generation, modification of the jet, ridge amplification, etc.). It is not sufficient to show the signal in the LWA budget without interpreting the underlying dynamics. The authors already offered some dynamical explanations in the response to my previous comment, so I am merely asking to add these in the main text where relevant. This omission is especially problematic given that the Introduction (1.75-84) frames the study as addressing the dynamical influence of LWCRE on the jet and storm track, but never follows through with a coherent explanation. To be absolutely clear, I am asking the authors to articulate the physical mechanism by which cloud radiative effects—particularly LWCRE—alter the atmospheric circulation, in terms of known dynamical processes such as diabatic PV generation, ascent in WCBs, or changes in jet structure. This includes explaining the role of cloud height and vertical heating profiles, since LWCRE is not a single uniform effect. Without this physical context, the claim that LWCRE enhances blocking lacks credibility, regardless of what appears in the LWA diagnostics. The LWA formulation, after all, is based on QGPV dynamics—you cannot claim it's a different framework and therefore exclude PV-based interpretation. At this point, I find it disappointing that these core interpretative issues remain unaddressed.

Authors:

We have added an explicit discussion that summarizes our proposed mechanisms based on LWA in terms of the common PV-based thinking and drawing an explicit analogy (see L444-457):

From a PV-based perspective, the enhanced upstream diabatic LWA source, associated with CREs and their feedback on LH, can be interpreted as enhanced diabatic PV source in the WCB region

upstream of the block, which in turn amplifies negative PV anomalies aloft (i.e., the ridge) in the upper troposphere (Nakamura and Huang2017, Neal et al., 2022). Such ridge amplification in our results can be explained by the enhanced downstream horizontal LWA convergence due to the CREs impacts on the background wind and quasi-stationary ridge (Fig. 8), which decreases the carrying capacity of the jet stream to carry wave activity (Fig. 5). This convergence, in turn, leads to a pile-up of LWA downstream, corresponding dynamically to the amplification of the upper-level ridge (negative PV anomaly) (see Fig. S13). Thus, this proposed mechanism based on the LWA perspective is consistent with PV-based thinking, while emphasizing the critical role of CREs and their feedback on LH in enhancing the upstream diabatic wave source essential for blocking formation (see L440-452).

We also revised some parts of the manuscript relevant to this process:

1. **WCB signature:** The signature of WCB-type ascent—vertical and poleward transport of low-PV air—has been included in the discussion of **Figs. 6 and 7** (see **L276-311**).
2. **Clean sequence of processes:** Non-conservative forcings (associated with CREs and their feedback on LH) and low-level eddy meridional heat flux generate LWA upstream of blocking events, which is then redistributed by zonal LWA convergence, favoring localized blocking formation (**L236-240, L355-360, L418-433, L446-457**).
3. **How LWCRE increases the upstream diabatic source (diabatic PV source)?** Two possible pathways were proposed: (1) Mid-tropospheric LW cloud heating amplifies large-scale vertical ascent and poleward moisture transport in the WCB region, thereby increasing LH; (2) LWCRE enhances LH through evaporation feedback, although this mechanism requires further investigation with targeted simulations (**L321-354**).
4. **CRE impacts on jet and quasi-stationary ridge:** We added a discussion that disabling CREs leads to a poleward jet shift (i.e., stronger PV gradient) and a weakened quasi-stationary ridge. These conditions increase the "carrying capacity" of the jet stream to carry wave activity, thereby limiting the pile-up of LWA that signifies blocking formation. This is consistent with reduced zonal LWA convergence and blocking in CLOCK and LWOFF (**L373-L379, L433-443, L449-452**).
5. **Linking CRE-induced jet/ridge modification to LWA in the WCB:** We emphasize that CRE-induced modifications to the jet and stationary waves increase the jet's carrying capacity to carry wave activity, consistent with weaker zonal LWA convergence in CLOCK and LWOFF compared to CTL (**L373-L379, L433-443, L449-452**).

We have also “explicitly” stated in the revised manuscript (**L172–177**) the reason for choosing the LWA budget instead of PV. Finally, we have also added a discussion acknowledging that fully explaining the jet, storm track, and stationary wave responses to cloud locking or the disabling of CREs remains a challenge in the community, as research on the diabatic dynamical effects of clouds on midlatitude circulation is still at an early stage (Voigt et al., 2021; Voigt et al., 2023; Lu et al., 2024}. A comprehensive explanation of the broader jet and stationary wave adjustments to CREs lies beyond the scope of the current manuscript, which underscores the importance of this line of research (see **L458-468**).

Second, the description of LWA sources and redistribution still remains unclear and fragmented, although I see much improvement compared to the previous version. I understand the authors do not want to label zonal LWA convergence as a “sink,” but the current wording does not help the reader clearly understand what is happening dynamically. The mechanism should be presented cleanly: non-conservative forcings (e.g., latent heating and CREs) and low-level eddy meridional heat flux generate LWA upstream of blocking events, which is then redistributed by zonal LWA convergence, forming localized blocking structures. This sequence is physically grounded and fully consistent with Yamamoto and Martineau (2024), which the authors should cite. Right now, the manuscript reads more like a descriptive catalog of LWA differences than a coherent dynamical explanation.

Authors:

We have revised the manuscript to “explicitly” state that the non-conservative forcings (e.g., LH and CREs) and low-level eddy meridional heat flux generate LWA upstream of the block, which is then redistributed by zonal LWA convergence, forming localized blocked structures. We have also emphasized that the upstream low-level eddy heat flux contributes to the upstream wave activity source, although its contribution is relatively small compared to the contribution from non-conservative diabatic forcings due to LH and CREs (see **Figs. 3(i-l) and 5(d-e)**), consistent with previous studies (e.g., Neal et al., 2022; Wang et al., 2021).

We have revised the manuscript to ensure that the sequence of processes is presented explicitly throughout (see **L194-L219, L236-L240, L277-L279, L355-L360, L418-L443, L446-456**). We have also cited Yamamoto and Martineau (2024) to highlight the role of low-level eddy heat flux (baroclinic generation) in an upstream adiabatic wave source (**L216**).

Reviewer #3 (Remarks to the Author):

I appreciate the authors' effort to address my previous concerns. The manuscript now provides much greater clarity regarding the dynamical context of CREs and diabatic heating. I have a few additional minor comments regarding the newly added paragraphs.

Authors: We are glad that the revised manuscript provides greater clarity. Below, we provide a point-by-point response to each comment, highlighted in blue.

1.211–1.216: There is some redundancy in the discussion of Figs. 3m–p. You might consider simplifying the whole thing to something like: “As shown in Figs. 3m–p, positive values of low-level sources and non-conservative (diabatic) processes are collocated with the emergence and growth of wave activity fluxes, acting as LWA sources, with non-conservative processes being of primary importance.”

Authors: Thank you for the suggestion. It has been revised (see **L207-209**).

1.375, 439, 451: What do you mean by “quasi-stationary wave ridge”? Is this different from the “large-scale stationary ridges” mentioned at 1.367? I found the expression confusing, since it is unclear whether you are referring to the climatological ridge or to a ridge associated with the blocking itself. Please clarify or rephrase.

Authors: By “large-scale stationary wave ridge,” we mean the climatological quasi-stationary ridge (Nakamura and Huang 2018 fig. 1; Barpanda and Nakamura 2025 fig.5). We have rephrased the text to clarify this (see **L360, L366, L392, L444**).

1.377: Did you mean “incipient” rather than “incident”?

Authors: By “incident,” we mean that the block is harder to form even when the incoming (upstream/approaching) transient wave-activity flux is at a given level (cf. Nakamura and Huang 2018; Barpanda and Nakamura 2025).

1.446–457: Here, the authors describe only the “indirect” diabatic impact (Hoskins et al. 1985; Steinfeld & Pfahl 2019). However, the “direct” diabatic impact — in which WCBs transport low-PV air from lower latitudes and altitudes into higher latitudes and the upper troposphere (e.g., Pfahl et al. 2015; Steinfeld & Pfahl 2019; Yamamoto et al. 2021) — should also be mentioned, as it likely plays a role.

Authors: Thank you for pointing this out. Since this paragraph is focused on discussing the LWA/PV-source perspective in relation to CRE-LH feedback (i.e., related to the indirect pathway), we have clarified that the proposed mechanism is consistent with the “indirect diabatic impact” from PV-based thinking. This keeps the main message clear without diluting it by introducing the direct impact of PV (see **L447-448**).